# FullFront: Benchmarking MLLMs Across the Full Front-End Engineering Workflow

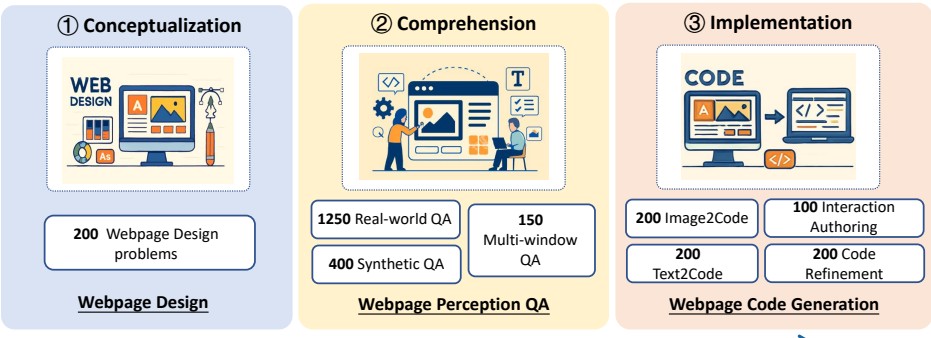

Figure 1: Mapping the full front-end engineering workflow to FullFront's benchmark tasks: (1) Conceptualization assessed by Webpage Design, (2) Comprehension by Webpage Perception QA, and (3) Implementation by Webpage Code Generation.

## Abstract

Front-end engineering involves a complex workflow where engineers conceptualize designs, translate them into code, and iteratively refine the implementation. While recent benchmarks primarily focus on converting visual designs to code, we present FullFront, a benchmark designed to evaluate Multimodal Large Language Models (MLLMs) **across the full front-end development pipeline**. FullFront assesses three fundamental tasks that map directly to the front-end engineering pipeline: Webpage Design (conceptualization phase), Webpage Perception QA (comprehension of visual organization and elements), and Webpage Code Generation (implementation phase). Unlike existing benchmarks that use either scraped websites with bloated code or oversimplified LLM-generated HTML, FullFront employs a novel, two-stage process to transform real-world webpages into clean, standardized HTML while maintaining diverse visual designs and avoiding copyright issues. Extensive testing of state-of-the-art MLLMs reveals significant limitations in page perception, code generation (particularly for image handling and layout), and interaction implementation. Our results quantitatively demonstrate performance disparities across models and tasks, and highlight a substantial gap between current MLLM capabilities and human expert performance in front-end engineering.

## 1 Introduction

Front-end engineering, a cornerstone of the modern digital experience, is an intricate process, as depicted in Figure 1. It transforms abstract concepts into initial designs (conceptualization), involves detailed visual comprehension (perception), and culminates in functional, interactive code (implementation) for web applications. This field is poised for significant transformation with the advent of Multimodal Large Language Models (MLLMs), whose capabilities in processing visual information and generating code offer compelling potential to streamline and even automate the front-end development, aligning with the aspirational goal of an "idea-to-design-to-code" paradigm.

Despite this burgeoning potential, a benchmark to assess MLLMs across the full front-end Engineering workflow is conspicuously absent. Instead, current evaluations tend to separately address crucial yet distinct capabilities: vision perception and code generation. For instance, benchmarks like IW-Bench (Guo et al., 2024) and WebCode2M (Gui et al., 2025) scrutinize MLLMs' code generation from

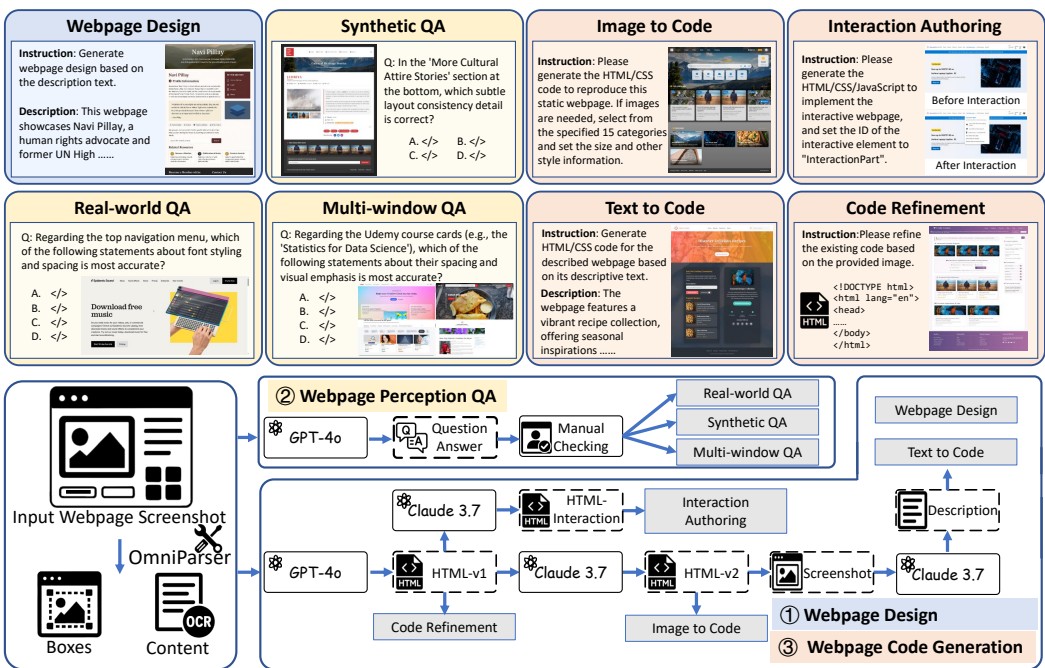

Figure 2: Overview of the eight subtasks FullFront covers and our data construction pipeline.

visual inputs but often possess a narrow task scope, overlooking vital aspects such as implementing interactive features or refining existing codebases. Conversely, while WebQuest (Wang et al., 2024) and Webqa (Chang et al., 2022) investigate MLLMs' visual understanding of webpages, the focus frequently remains on content-level reasoning, thereby neglecting the fine-grained perceptual acuity concerning element size, positioning, and layout, which is indispensable for accurate front-end implementation. Most critically, these fragmented approaches generally omit the initial conceptual "design" phase of development, and therefore fall short of gauging MLLM proficiency in end-to-end front-end engineering.

In this work, we introduce **FullFront**, a benchmark meticulously designed to evaluate MLLMs across the full front-end engineering workflow. As depicted in Figure 2, FullFront distinctively offers a holistic assessment through three core tasks: (1) **Webpage Design** (200 problems), which assesses the model's ability to structure and organize visual elements to present some given content; (2) **Webpage Perception QA** (three subtasks and 1,800 multiple-choice questions), which evaluates the perception of visual organization, element characteristics, and spatial relationships within a webpage; and (3) **Webpage Code Generation** (four subtasks and 700 code generation problems), which focuses on the accurate translation of visual designs into functional code, including interaction implementation and code refinement. We collect real-world webpages and develop an MLLM-driven pipeline to reconstruct them into clean, standardized, and copyright-free HTML, ensuring high controllability while preserving original visual diversity for robust benchmark data. This comprehensive task structure and our evaluation framework, incorporating fine-grained visual similarity scores and detailed code-level metrics (including structural and content-based comparisons), provide a multifaceted and robust assessment of model capabilities across the full front-end engineering workflow.

Benchmarking state-of-the-art open-source and proprietary MLLMs with FullFront reveals significant challenges across the board. In the Webpage Design task, current text-to-image MLLMs demonstrate an ability to produce general layout concepts but lack the precision for high-fidelity webpage designs that accurately reflect detailed textual descriptions. In Webpage Perception QA, even leading models struggle to achieve human-comparable accuracy; for instance, the best-performing model, GPT-5, achieves an average accuracy below 60% across these tasks, starkly contrasting with human performance exceeding 95%. Our analysis reveals that MLLMs face considerable difficulties in accurately perceiving element alignment, size, and positioning within webpages. For Webpage Code Generation, while proprietary models like Claude 3.7 Sonnet and GPT-4.1 generally outperform open-source alternatives, they still encounter difficulties, particularly in accurately handling complex

front-end details such as image manipulation, layout fidelity, and interaction implementation. These findings underscore the critical need to enhance current MLLM capabilities within the front-end development workflow to bridge the substantial gap between their current performance and the requirements for expert-level engineering. In summary, our main contributions are as follows:

- **Comprehensive Full Front-End Workflow Benchmark**: We propose FullFront, the first benchmark to unify Webpage Design (conceptualization), Perception QA (comprehension), and Code Generation (implementation) into a cohesive pipeline. This framework simulates the authentic engineering lifecycle to rigorously evaluate the capabilities of current MLLMs and future Generalist Front-End Agents.

- **Robust and Bias-Resilient Evaluation Framework**: We introduce fine-grained visual and code-level metrics that are rigorously validated against human judgment. Extensive ablation studies confirm our framework's robustness against data-pipeline artifacts and model-as-judge biases, establishing a fair standard for cross-model comparison.

- **In-depth Analysis of Perception-Generation Decoupling**: Beyond performance rankings, our evaluation uncovers a critical "Perception-Implementation Gap." We reveal that top-tier MLLMs often suffer from "Grounding Failures"—relying on strong internal priors to generate correct code despite flawed visual perception. This insight challenges the assumption that code generation implies visual understanding, a nuance missed by isolated benchmarks.

## 2 RELATED WORK

**Applications of MLLMs in Web** Recently, the application of MLLMs in the web domain (Zhao et al., 2025; Wu et al., 2024; Tan et al., 2024; Wang et al., 2025) has garnered considerable research attention. Numerous innovative approaches have emerged, enabling MLLMs to navigate and manipulate websites according to user instructions (Zheng et al., 2024; Yoran et al., 2024; Cheng et al., 2024). For instance, Mind2Web (Deng et al., 2023) pioneers a generalist web agent by training models on diverse web tasks, demonstrating their capability to follow complex natural language commands across various websites. Similarly, WinClick (Hui et al., 2025) focuses on GUI grounding with MLLMs, allowing for more precise interaction with web elements by understanding their visual and textual properties to execute user commands like clicking buttons or filling forms. These advancements highlight a growing trend towards creating more autonomous and intelligent web interaction agents.

**Webpage Benchmarks and Datasets** Several benchmarks and datasets have been developed to evaluate MLLMs on webpage-related tasks. For instance, a significant body of work (Chen et al., 2024a;b; Wang et al., 2024; Chang et al., 2022; Chen et al., 2021; Wu et al., 2025; Hao et al., 2025) leverages real-world webpages to assess MLLMs' capabilities in element grounding and content reasoning via question-answering (QA). ScreenWords (Wang et al., 2021) focuses on screen summarization, while VisualWebBench (Liu et al., 2024) offers seven QA tasks for a broader understanding assessment. Separately, research has also benchmarked MLLMs for front-end code generation from screenshots. Methodologies vary: Design2Code (Si et al., 2024), WebCode2M (Gui et al., 2025), and IW-Bench (Guo et al., 2024) curate datasets by scraping and simplifying existing code. In contrast, Web2Code (Yun et al., 2024) and WebSight (Laurençon et al., 2024) employ LLMs for code generation, and Pix2Code (Beltramelli, 2018) uses a stochastic UI generator. Notable contributions also include MRWeb's (Wan et al., 2024) "resource list" for external resources; Interaction2Code's (Xiao et al., 2024) focus on dynamic webpage generation; and DesignBench's (Xiao et al., 2025) extension to multi-framework iterative editing and repair tasks.

## 3 BENCHMARK

### 3.1 DATA CURATION

**Webpage Design** The Webpage Design task aims to evaluate text-to-image generation MLLMs as webpage designers. We provide 200 textual descriptions of synthetic webpages sampled from the Text to Code task dataset (see below). MLLMs are required to generate webpage design images based on these descriptions. This process tests how effectively models can transform textual requirements into visual designs, including their understanding of webpage layouts and element relationships.

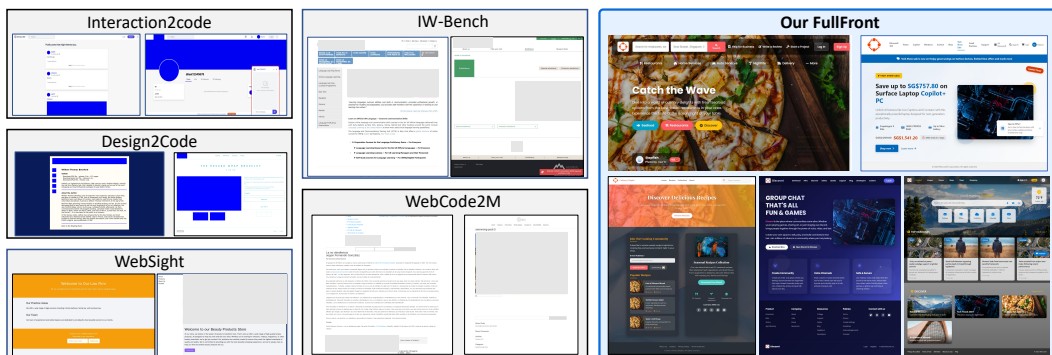

Figure 3: Comparison of the screenshots used in FullFront for webpage code generation tasks with those of other benchmarks. We are the first to not use a single image placeholder or random images.

Since textual descriptions naturally cannot capture all visual design nuances, this task also assesses models' ability to make reasonable design decisions where specifications are incomplete.

**Webpage Perception QA**   This task assesses MLLMs' perception of webpage elements, including their position, style, spatial relationships, and overall page layout, through three subtasks. The **Real-world QA** subtask evaluates perceptual abilities using 625 real webpage screenshots (270 manually collected, 355 sourced from Uground (Gou et al., 2024) and IW-Bench (Guo et al., 2024)), resulting in 1,250 question-answer pairs. Complementing this, **Synthetic QA** assesses model performance on 400 Q/A pairs derived from 200 synthesized webpage screenshots generated via the specific methodology (detailed in the next Webpage Code Generation). Finally, **Multi-window QA** elevates task complexity by presenting 75 samples, each combining 2-4 screenshots from the Real-world QA set (totaling 150 Q/A pairs), thereby challenging models to accurately identify and locate the screenshot relevant to the posed question. Questions are primarily generated by GPT-4o (OpenAI, a), augmented with bounding boxes and OCR data extracted by OmniParser (Lu et al., 2024). This allows GPT-4o to focus on generating challenging, high-quality multiple-choice questions based on page content and structure rather than low-level perception. All generated questions undergo rigorous manual review and modification to ensure correctness, challenge, and task validity. To mitigate ethical risks such as privacy leakage, all webpage screenshots are manually inspected and annotated to remove personal data and harmful content.

**Webpage Code Generation**   The Webpage Code Generation task evaluates the ability to translate visual page designs into executable HTML. Existing benchmarks (e.g., WebCode2M (Gui et al., 2025), Design2Code (Si et al., 2024)) often simplify HTML from sources like Common Crawl (Crawl, 2025) to mitigate ethical issues, remove external dependencies and redundant elements, and standardize code for comparison. Despite these benefits, the simplification process is inherently time-consuming and difficult to generalize across varied real-world webpages. Meanwhile, HTML generated with LLMs from scratch (e.g., WebSight (Laurençon et al., 2024)) often lacks authentic complexity. A key limitation of existing datasets is their handling of images, such as using generic placeholders or random images, which hinders the assessment of nuanced image understanding and utilization crucial for high-fidelity webpage replication. To overcome these issues, we introduce a synthesis pipeline from real-world webpages. This two-stage process (detailed in Figure 2) starts with a real-world webpage screenshot and its OmniParser-extracted element information. GPT-4o generates an initial HTML-v1, which Claude 3.7 Sonnet then refines—adjusting styles, positions, alignments, and layouts—into a higher-quality, more complex HTML-v2. This HTML-v2 and its rendered page serve as ground truth. For image handling, we utilize a category-based strategy to preserve the visual information from real-world webpage screenshots (see Appendix F.2). As shown in Figure 3, our method generates webpages that are demonstrably superior to other benchmarks in complexity and diversity. Unlike traditional tasks that only involve providing a webpage screenshot for HTML code generation, we design four distinct subtasks to evaluate MLLMs' front-end code generation capabilities under various conditions: **Image to Code** (200 samples) evaluates direct HTML generation from these HTML-v2 rendered screenshots; **Text to Code** (200 samples) assesses HTML generation based solely on manually verified textual descriptions of HTML-v2 rendered pages; **Interaction Authoring** (100 samples) measures the ability to implement dynamic behaviors,

requiring MLLMs to reproduce a static page (from HTML-v1 as a base) and add specified interactions based on screenshots depicting the page before and after the interaction; and **Code Refinement** (200 samples) simulates code optimization by requiring MLLMs to refine provided HTML-v1 code to match the quality and complexity of an HTML-v2 rendered screenshot. For more detailed task descriptions, see the Appendix F.

## 3.2 EVALUATION METRICS

To comprehensively evaluate MLLM performance on FullFront, we employ visual and code-level metrics, detailed below and applied specifically to each core task.

**Visual Level Metrics** We assess MLLM generative capabilities by comparing the visual similarity of their output (rendered HTML or direct design images) against ground-truth images. This includes two complementary automatic metrics: the **CLIP Score** (Radford et al., 2021), which measures high-level conceptual consistency via embedding space similarity, and the **DINOv2 Score** (Oquab et al., 2023), which focuses on finer-grained structural and pixel-level fidelity. Alongside these automatic metrics, the **Gemini Visual Score** leverages Gemini 2.5 Flash to evaluate the output on a 0-10 scale across ten fine-grained criteria, such as Alignment and Spacing (see Appendix G.1).

**Code Level Metrics** To evaluate code similarity, we propose and design the **Code Score**, which assesses MLLM-generated against reference HTML. It parses both into Document Object Model (DOM) trees and extracts associated CSS, then performs a weighted aggregation. This considers structural similarity, quantified by the Longest Common Subsequence (LCS) ratio of DOM tag sequences. It also assesses content-type similarity for text, images, and forms, where corresponding elements are identified and compared based on content (e.g., text via SequenceMatcher (Seq)), key styling attributes (e.g., color, font size, image dimensions), and critical attributes (e.g., image src, form element type). An implementation rate for each content type, reflecting the proportion of reference elements found, adjusts these similarity scores to capture both quality and completeness. The final Code Score combines structural and adjusted content-type similarities using predefined weights. Further specifics on the Code Score calculation are available in the Appendix G.2.

For the Webpage Design, Visual Level Metrics assess generated design quality. For Webpage Perception QA, standard accuracy (correctly answered multiple-choice questions) is used. The Webpage Code Generation employs both Visual Level Metrics and the Code Score. To ensure the credibility of our evaluation framework, we rigorously validated these automatic metrics against human preferences. As detailed in our Spearman's rank correlation analysis in Appendix G.3, our proposed **Code Score** ($\rho = 0.9152$) and **Gemini Visual Score** ($\rho = 0.9364$) both demonstrate an exceptionally high correlation with human judgments, confirming their reliability as robust proxies for assessing MLLM performance.

## 4 EXPERIMENTS

### 4.1 EVALUATION SETTINGS

**FullFront-mini Dataset** To facilitate rapid iterative evaluation of MLLMs, we constructed a FullFront-mini dataset. For specifics on the FullFront-mini setup, see Appendix C.

**Models** We evaluate the performance of twelve state-of-the-art MLLMs on the Webpage Perception QA and Webpage Code Generation tasks. This set includes four open-source models (Qwen2.5-VL-72B-Instruct (Bai et al., 2025), InternVL2.5-78B (Chen et al., 2024c), InternVL3-78B (Zhu et al., 2025), and LLaVA-Onevision-72B (Li et al., 2024)) and eight proprietary models (Claude 3.7 Sonnet (Anthropic), Gemini 2.5 Flash (Deepmind, a), GPT-4o (OpenAI, a), o4-mini (OpenAI, e), GPT-4.1 (OpenAI, b), GPT-5 (OpenAI, c), o1 (OpenAI, d) and Gemini 2.5 Pro (Deepmind, b)). For the Webpage Design task, which targets image generation MLLMs, we test the capabilities of GPT-4o (OpenAI, a) and gemini-2.0-flash-exp-image-generation (Kampf & Brichtova). We report the results for o1 and Gemini 2.5 Pro solely on the FullFront-mini dataset.

Table 1: Evaluation results of Webpage Design task. We mark the **better results** with bold font.

| Model | Gemini Visual Score | CLIP Score | DINOv2 Score | Human Score |
|---|---|---|---|---|
| GPT-4o | **5.0450** | **0.7445** | **0.5598** | **6.9600** |
| gemini-2.0-flash-exp-image-generation | 2.0190 | 0.6901 | 0.4798 | 6.0400 |

Table 2: Evaluation results on three Webpage Perception QA tasks. Among the MLLM results, we mark the **best results** with bold font and the second best with underline.

| Model | FullFront | | | FullFront-mini | | |
|---|---|---|---|---|---|---|
| | Real-world | Synthetic | Multi-window | Real-world | Synthetic | Multi-window |
| Qwen2.5-VL-72B-Instruct | 0.4696 | 0.4950 | 0.4267 | 0.4550 | 0.5100 | 0.4000 |
| InternVL2.5-78B | 0.4696 | 0.5050 | 0.4267 | 0.4950 | 0.4500 | 0.3400 |
| InternVL3-78B | 0.4816 | 0.5375 | 0.4600 | 0.4700 | 0.5100 | 0.4400 |
| LLaVA-Onevision-72B | 0.3296 | 0.3275 | 0.2733 | 0.3450 | 0.2900 | 0.2600 |
| Claude 3.7 Sonnet | 0.5464 | 0.5325 | 0.4533 | 0.5250 | 0.5000 | 0.5600 |
| Gemini 2.5 Flash | 0.4800 | 0.4250 | 0.3867 | 0.4550 | 0.4400 | 0.4000 |
| GPT-4o | 0.4448 | 0.4675 | 0.3733 | 0.4450 | 0.4400 | 0.3200 |
| o4-mini | 0.4976 | 0.5300 | 0.4400 | 0.4800 | 0.5000 | 0.4600 |
| GPT-4.1 | 0.4672 | 0.4650 | 0.3733 | 0.4400 | 0.4200 | 0.3600 |
| GPT-5 | **0.5816** | **0.5775** | **0.5467** | 0.5150 | 0.4900 | 0.4800 |
| o1 | – | – | – | 0.4350 | 0.4600 | 0.4200 |
| Gemini 2.5 Pro | – | – | – | 0.5200 | 0.5800 | 0.4800 |
| Human Expert | – | – | – | **0.9700** | **0.9600** | **0.9400** |

## 4.2 MAIN RESULTS

**Webpage Design**   On the Webpage Design task, current text-to-image MLLMs exhibit a foundational capability in generating general layout concepts but encounter difficulties in producing high-fidelity designs that accurately reflect detailed textual descriptions. As shown in Table 1, GPT-4o outperforms gemini-2.0-flash-exp-image-generation across all metrics. Furthermore, qualitative examples in Appendix D illustrate that GPT-4o demonstrates superior performance in rendering overall page structure, typography, and element implementation fidelity.

Table 3: Evaluation results of different models on four Webpage Code Generation tasks. Ref: Code Refinement; Img: Image to Code; Inter: Interaction Authoring; Text: Text to Code.

| Model | Code Score | | | | Gemini Visual Score | | | | CLIP Score | | | | DINOv2 Score | | | |
|---|---|---|---|---|---|---|---|---|---|---|---|---|---|---|---|---|
| | Ref | Img | Inter | Text | Ref | Img | Inter | Text | Ref | Img | Inter | Text | Ref | Img | Inter | Text |
| Qwen2.5-VL-72B-Instruct | 0.53 | 0.40 | 0.40 | 0.49 | 6.21 | 4.48 | 6.22 | 5.18 | 0.79 | 0.72 | 0.76 | 0.73 | 0.63 | 0.49 | 0.67 | 0.47 |
| InternVL2.5-78B | 0.36 | 0.33 | 0.30 | 0.50 | 5.03 | 4.01 | 3.51 | 5.44 | 0.74 | 0.74 | 0.69 | 0.74 | 0.67 | 0.57 | 0.58 | 0.48 |
| InternVL3-78B | 0.49 | 0.42 | 0.38 | 0.50 | 5.87 | 4.47 | 4.48 | 4.91 | 0.77 | 0.73 | 0.73 | 0.72 | 0.61 | 0.53 | 0.60 | 0.44 |
| LLaVA-Onevision-72B | 0.33 | 0.14 | 0.06 | 0.41 | 4.90 | 1.89 | 0.45 | 5.13 | 0.73 | 0.65 | 0.58 | 0.74 | 0.66 | 0.42 | 0.17 | 0.50 |
| Claude 3.7 Sonnet | 0.63 | **0.64** | **0.55** | **0.64** | 8.36 | **8.93** | **9.18** | 8.14 | 0.88 | **0.89** | **0.86** | 0.87 | 0.80 | **0.81** | 0.82 | 0.78 |
| Gemini 2.5 Flash | **0.70** | 0.63 | 0.52 | 0.59 | 8.65 | 8.64 | 8.07 | 8.01 | **0.89** | 0.88 | 0.81 | 0.87 | 0.83 | 0.80 | 0.79 | 0.79 |
| GPT-4o | 0.43 | 0.34 | 0.36 | 0.49 | 6.53 | 5.91 | 5.81 | 6.14 | 0.82 | 0.81 | 0.76 | 0.74 | 0.73 | 0.69 | 0.69 | 0.52 |
| o4-mini | 0.62 | 0.57 | **0.55** | 0.51 | 8.16 | 8.47 | 8.84 | 6.87 | 0.86 | 0.87 | 0.84 | 0.82 | 0.75 | 0.76 | 0.77 | 0.68 |
| GPT-4.1 | 0.67 | 0.61 | **0.55** | 0.55 | **9.03** | 8.89 | 9.13 | 8.36 | **0.89** | 0.88 | 0.84 | 0.87 | 0.80 | 0.80 | **0.84** | **0.80** |
| GPT-5 | 0.58 | 0.56 | 0.53 | 0.59 | 8.84 | 8.58 | 9.00 | **8.48** | 0.88 | 0.88 | 0.85 | **0.88** | 0.81 | 0.81 | 0.83 | 0.79 |

| Model (FullFront-mini) | Code Score | | | | Gemini Visual Score | | | | CLIP Score | | | | DINOv2 Score | | | |
|---|---|---|---|---|---|---|---|---|---|---|---|---|---|---|---|---|
| | Ref | Img | Inter | Text | Ref | Img | Inter | Text | Ref | Img | Inter | Text | Ref | Img | Inter | Text |
| Qwen2.5-VL-72B-Instruct | 0.48 | 0.38 | 0.43 | 0.48 | 6.35 | 4.26 | 6.53 | 5.29 | 0.79 | 0.72 | 0.75 | 0.74 | 0.68 | 0.51 | 0.70 | 0.44 |
| InternVL2.5-78B | 0.38 | 0.37 | 0.37 | 0.35 | 5.12 | 3.97 | 3.58 | 5.07 | 0.72 | 0.74 | 0.67 | 0.74 | 0.65 | 0.56 | 0.60 | 0.48 |
| InternVL3-78B | 0.51 | 0.43 | 0.41 | 0.41 | 5.80 | 4.07 | 4.33 | 4.60 | 0.77 | 0.71 | 0.70 | 0.72 | 0.61 | 0.48 | 0.57 | 0.42 |
| LLaVA-Onevision-72B | 0.33 | 0.16 | 0.07 | 0.38 | 5.30 | 1.91 | 0.50 | 5.13 | 0.74 | 0.65 | 0.58 | 0.76 | 0.67 | 0.39 | 0.18 | 0.52 |
| Claude 3.7 Sonnet | 0.64 | 0.60 | **0.61** | 0.57 | 8.01 | 9.16 | 9.18 | 8.18 | 0.86 | **0.90** | **0.86** | 0.86 | 0.76 | 0.80 | **0.83** | 0.76 |
| Gemini 2.5 Flash | **0.68** | 0.58 | 0.55 | **0.66** | **8.71** | 8.96 | 8.06 | 7.89 | **0.89** | 0.88 | 0.81 | 0.86 | **0.81** | 0.81 | 0.74 | 0.76 |
| GPT-4o | 0.40 | 0.35 | 0.38 | 0.38 | 6.39 | 5.77 | 6.03 | 6.27 | 0.82 | 0.80 | 0.76 | 0.74 | 0.70 | 0.69 | 0.70 | 0.48 |
| o4-mini | 0.59 | 0.55 | 0.56 | 0.53 | 7.87 | 8.63 | 8.79 | 6.90 | 0.86 | 0.87 | 0.82 | 0.82 | 0.74 | 0.77 | 0.75 | 0.69 |
| GPT-4.1 | 0.65 | **0.63** | 0.48 | 0.58 | 8.44 | **9.32** | **9.23** | 8.44 | 0.87 | **0.90** | 0.83 | 0.86 | 0.80 | 0.81 | **0.83** | **0.78** |
| GPT-5 | 0.55 | 0.60 | 0.57 | 0.53 | 8.70 | 8.38 | 8.48 | **9.03** | 0.88 | **0.90** | 0.84 | **0.88** | 0.78 | **0.82** | 0.82 | 0.76 |

**Webpage Perception QA**   As demonstrated in Table 2, MLLMs generally exhibit weak perceptual capabilities on the Webpage Perception QA task. On the FullFront-mini subset, even the top-performing models, Claude 3.7 Sonnet and Gemini 2.5 Pro, achieve an average accuracy barely exceeding 50% across the three subtasks. Furthermore, LLaVA-OneVision-72B's accuracy remains below 35% on all QA subtasks. Critically, all models perform significantly worse than human experts,

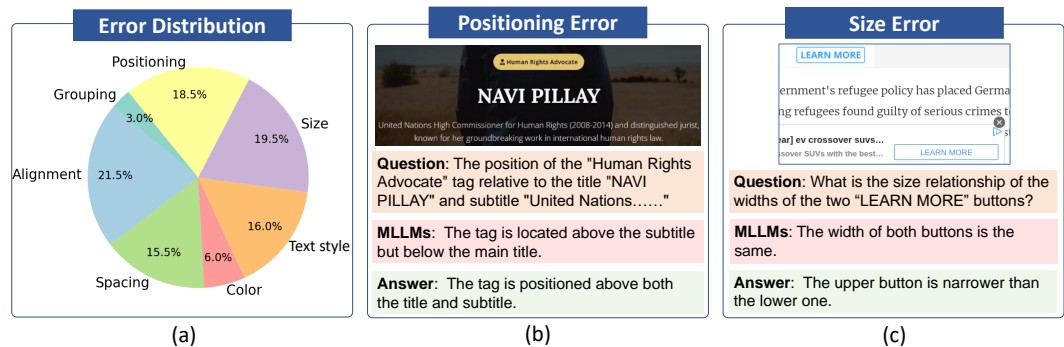

Figure 4: MLLM Errors in Webpage Perception QA. (a) Distribution of error types for 200 questions. (b) An illustrative example of a Positioning Error. (c) An illustrative example of a Size Error.

with accuracy gaps of 44.5%, 38.0%, and 38.0% on three subtasks respectively, highlighting their challenges in fine-grained page perception. Notably, this task reveals no substantial performance disparity between closed-source and open-source models; for instance, on the full FullFront benchmark, InternVL3-78B achieves leading accuracies of 53.75% on Synthetic QA and 46.00% on Multi-window QA. Further analysis indicates nearly identical model performance on single-page Real-world and Synthetic QA, while performance degrades considerably on the more complex Multi-window QA.

Table 4: Human Evaluation of MLLM-generated webpages on FullFront-mini.

Table 5: Interaction success rate (%). "(mini)" indicates the results on the FullFront-mini.

| Model | Ref | Image | Inter | Text | Model | Rate | Rate (mini) |
|---|---|---|---|---|---|---|---|
| Qwen2.5-VL-72B-Instruct | 6.18 | 5.72 | 7.02 | 4.90 | Qwen2.5-VL-72B-Instruct | 57.00 | 64.00 |
| InternVL2.5-78B | 5.36 | 4.78 | 5.04 | 4.24 | InternVL2.5-78B | 47.00 | 54.00 |
| InternVL3-78B | 6.32 | 5.56 | 5.44 | 4.62 | InternVL3-78B | 48.00 | 56.00 |
| LLaVA-Onevision-72B | 5.64 | 2.96 | 0.58 | 4.42 | LLaVA-Onevision-72B | 16.00 | 24.00 |
| Claude 3.7 Sonnet | 8.00 | 8.48 | **8.80** | 8.10 | Claude 3.7 Sonnet | 78.00 | 86.00 |
| Gemini 2.5 Flash | 8.44 | 8.40 | 7.86 | 8.24 | Gemini 2.5 Flash | 70.00 | 72.00 |
| GPT-4o | 6.96 | 6.74 | 6.76 | 6.24 | GPT-4o | 80.00 | 74.00 |
| o4-mini | 8.32 | 7.88 | 7.86 | 6.84 | o4-mini | **93.00** | **94.00** |
| GPT-4.1 | **8.50** | **8.88** | 8.46 | **8.42** | GPT-4.1 | 78.00 | 80.00 |
| GPT-5 | 8.38 | 8.52 | 8.76 | 8.34 | GPT-5 | 92.00 | 92.00 |

**Webpage Code Generation   Closed-Source Model Dominance.** In the Webpage Code Generation task, closed-source models significantly outperform their open-source counterparts across all subtasks and metrics, with no open-source model securing a top-two position in any category. As detailed in Table 3, Claude 3.7 Sonnet consistently leads, closely followed by other proprietary models like GPT-5, Gemini 2.5 Flash and GPT-4.1, all demonstrating impressive, top-tier results. For instance, in the Code Refinement task, GPT-4.1 achieves a Gemini Visual Score of 9.03, indicating near-perfect visual reproduction in most cases, whereas the best-performing open-source model, Qwen2.5-VL-72B-Instruct, scores only 6.21 under the same settings. While Qwen2.5-VL-72B-Instruct and InternVL3-78B show relatively strong performance among open-source options, their scores are generally comparable only to GPT-4o rather than the leading closed-source models. A consistent trend across models is the alignment of performance across different metrics; models excelling in one visual or code-based score typically perform similarly well in others.

**Insights on Input Modality and Task Difficulty.** Subtask analysis reveals distinct patterns regarding input efficacy. While providing partial HTML (Code Refinement) naturally yields the highest performance, a comparative analysis of input modalities offers a surprising insight: **Text to Code performance rivals Image to Code** for top-tier models. For instance, Claude 3.7 Sonnet achieves a Code Score of 0.64 on Text to Code, matching its Image to Code performance. Furthermore, as detailed in Appendix F.8, even text-only LLMs achieve highly competitive results when provided with detailed descriptions. This indicates that **precise textual specifications can effectively substitute for visual inputs**—a critical insight for designing efficient, low-latency front-end agents where visual processing overhead might be prohibitive. Conversely, Interaction Authoring remains the most

challenging task, yielding lower scores despite simpler HTML-v1 targets, a difficulty underscored by interaction implementation rates (Table 5) where closed-source models exceed 70% success, far surpassing open-source models like LLaVA-Onevision-72B (16%).

**Human Evaluation and Reliability.** Blind human evaluation on the FullFront-mini dataset, using Gemini Visual Score criteria (Table 4), further confirms that closed-source models like Claude 3.7 Sonnet and GPT-5 are perceived as more accurate, frequently scoring above 8/10 for reproduction quality. While these models achieve high overall fidelity, illustrative examples in Appendix F reveal that even top performers can exhibit minor imperfections in fine-grained details.

## 5 DISCUSSION

### 5.1 WHERE DO MLLMS STRUGGLE MOST IN PERCEIVING WEBPAGES?

By analyzing the error types of 200 questions that all MLLMs (except o1 and Gemini 2.5 Pro) fail to answer correctly, we gain insight into the primary difficulties current MLLMs face in page perception. As shown in Figure 4 (a), MLLMs exhibit a particular difficulty in accurately understanding the alignment (21.5%), size (19.5%), spacing (15.5%), and precise positioning (18.5%) of page elements. These factors constitute the core reasons behind perception failures. For example, Figure 4 (b) shows an instance where MLLMs fail to correctly identify the position of the tag labeled "Human Rights Advocates" relative to the main title and subtitle, while Figure 4 (c) demonstrates an incorrect comparison of the sizes of two buttons (More examples and analysis can be found in Appendix E).

### 5.2 DECOUPLING ANALYSIS: THE RELATIONSHIP BETWEEN VISUAL PERCEPTION AND CODE GENERATION

A counter-intuitive finding from our results (Table 2 and Table 3) is that models excelling in perceptual QA tasks do not consistently outperform in code generation. For instance, InternVL3-78B demonstrates superior perceptual QA performance compared to Gemini 2.5 Flash, yet lags significantly in code generation capabilities. To investigate this causal link, we manually verified 100 samples from the Synthetic QA against their corresponding Image to Code outputs. We classified model behaviors into four quadrants based on two binary criteria: **Perception Accuracy** ($p$), determined by whether the model correctly answered the QA question; and **Code Implementation** ($c$), determined by whether the specific visual component mentioned in the QA was correctly implemented.

**Divergent Failure Modes.** Table 6 presents the confusion matrix. The distribution reveals distinct failure modes across model categories. Open-source models (e.g., Qwen2.5-VL-72B-Instruct, InternVL3-78B) exhibit a high frequency of $p\checkmark c\times$ cases, indicating that even when visual perception is accurate, coding capability remains the primary bottleneck. Conversely, advanced proprietary models like Claude 3.7 Sonnet and GPT-5 show a significant number of $p \times c\checkmark$ instances. This quadrant represents a "Grounding Failure" where the model fails to perceive fine-grained details (answering the QA incorrectly) yet successfully generates the correct code. This phenomenon suggests that these models often bypass precise visual grounding, relying on strong internal code priors to hallucinate correct implementations based on general layout patterns.

**Performance Decoupling.** To further probe this decoupling effect, we analyzed the macroscopic generation scores grouped by the model's perception accuracy on the corresponding pages. As detailed in Table 7, the generation performance metrics remain remarkably consistent regardless of whether the model correctly answered the perception questions. This statistical invariance implies that fine-grained perceptual errors do not significantly degrade the overall visual quality of the generated page. Note that Qwen2.5-VL-72B-Instruct's lower score in the "Correct" group is attributed to a disproportionate occurrence of fatal generation errors (e.g., "Abnormal Image Size") in that specific subset, reflecting generation instability rather than a negative causal link.

This quantitative decoupling aligns with our qualitative observations. As illustrated in Figure 4 (b), all tested models failed to correctly identify the relative position of the "Human Rights Advocate" tag in the perceptual QA phase, yet correctly placed it in the generated code (see detailed analysis in Appendix F.9). For more examples of the four quadrants, please refer to Appendix F.10.

Table 6: Confusion matrix of Perception accuracy vs. Code Implementation accuracy on 100 samples from Synthetic QA. $p$: Perception correctness; $c$: Code implementation correctness.

| Model | $p\checkmark c\checkmark$ | $p\checkmark c\times$ | $p\times c\checkmark$ | $p\times c\times$ |
|---|---|---|---|---|
| Qwen2.5-VL-72B-Instruct | 22 | 28 | 10 | 40 |
| InternVL3-78B | 25 | 23 | 8 | 44 |
| GPT-4o | 32 | 13 | 21 | 34 |
| Claude 3.7 Sonnet | 44 | 3 | 36 | 17 |
| Gemini 2.5 Flash | 39 | 5 | 25 | 31 |
| GPT-5 | 48 | 6 | 31 | 15 |

Table 7: Detailed breakdown of generation performance grouped by perception accuracy. "Correct" implies the model answered all QA questions for the page correctly; "Wrong" implies all QA questions for the page were incorrect. $N$ denotes the number of pages in each group.

| Model | Perception Group | Code Score | Gemini Visual Score | CLIP Score | DINOv2 Score |
|---|---|---|---|---|---|
| Qwen2.5-VL-72B-Instruct | Correct ($N = 46$) | 0.386 | 4.430 | 0.705 | 0.409 |
| | Wrong ($N = 48$) | 0.405 | 5.117 | 0.753 | 0.529 |
| InternVL3-78B | Correct ($N = 59$) | 0.418 | 4.444 | 0.736 | 0.554 |
| | Wrong ($N = 42$) | 0.409 | 4.422 | 0.715 | 0.486 |
| GPT-4o | Correct ($N = 43$) | 0.336 | 6.007 | 0.799 | 0.672 |
| | Wrong ($N = 56$) | 0.342 | 5.896 | 0.809 | 0.697 |
| Claude 3.7 Sonnet | Correct ($N = 57$) | 0.646 | 8.826 | 0.887 | 0.805 |
| | Wrong ($N = 44$) | 0.649 | 8.957 | 0.892 | 0.790 |
| Gemini 2.5 Flash | Correct ($N = 42$) | 0.634 | 8.790 | 0.890 | 0.838 |
| | Wrong ($N = 72$) | 0.628 | 8.596 | 0.877 | 0.796 |
| GPT-5 | Correct ($N = 75$) | 0.568 | 8.436 | 0.882 | 0.808 |
| | Wrong ($N = 44$) | 0.535 | 8.548 | 0.878 | 0.832 |

## 5.3 CAN MLLMS BE AN EXCELLENT FRONT-END ENGINEER?

To determine if MLLM-generated pages are preferred by human to real-world versions, five human experts conduct a blind evaluation of 100 webpage generated by various MLLMs alongside their real-world counterparts. The results, as shown in Figure 6, indicate that pages from leading models (e.g., o4-mini, Gemini 2.5 Flash) are, in the vast majority of cases, preferred to their real-world counterparts (see Appendix G.5.2 for more analysis). However, further analysis of the generated webpages reveals that MLLMs can exhibit three prevalent error categories, illustrated in Figure 5: Abnormal Image Size (abnormally large images disrupting layout integrity), Blank Page (entirely blank screenshots despite non-empty code), and Isolation Error (instances where only isolated interactive buttons are generated, neglecting page content). Each error type significantly degrades the effectiveness of the generated webpage. Table 8 shows that open-source models exhibit these errors markedly more often than closed-source counterparts; this considerably diminishes their reliability and stability. Furthermore, a detailed examination of code-level performance (Table 9) indicates that current MLLMs still have

Table 8: Counts of three error types in Webpage Code Generation tasks. Size: Abnormal Image Size; Blank: Blank Page; Isolation: Isolation Error.

| Model | Size | | | | Blank | | | | Isolation |
|---|---|---|---|---|---|---|---|---|---|
| | Ref | Img | Inter | Text | Ref | Img | Inter | Text | Inter |
| Qwen2.5-VL-72B-Instruct | 38 | 62 | 11 | 84 | 8 | 4 | 2 | 3 | 2 |
| InternVL2.5-78B | 5 | 20 | 2 | 76 | 2 | 14 | 12 | 4 | 11 |
| InternVL3-78B | 6 | 20 | 5 | 95 | 5 | 14 | 10 | 1 | 1 |
| LLaVA-Onevision-72B | 1 | 22 | 3 | 64 | 5 | 45 | 1 | 9 | 88 |
| Claude 3.7 Sonnet | 2 | 1 | 0 | 3 | 3 | 0 | 0 | 0 | 0 |
| Gemini 2.5 Flash | 3 | 3 | 2 | 2 | 2 | 0 | 0 | 1 | 0 |
| GPT-4o | 2 | 11 | 0 | 54 | 1 | 0 | 0 | 1 | 0 |
| o4-mini | 16 | 9 | 2 | 17 | 0 | 0 | 0 | 6 | 0 |
| GPT-4.1 | 4 | 1 | 0 | 2 | 2 | 1 | 2 | 1 | 0 |
| GPT-5 | 1 | 1 | 0 | 1 | 0 | 1 | 0 | 0 | 0 |

Table 9: Average code-level performance metrics (Structure, Text, Image, and Form) across four Webpage Code Generation sub-tasks on FullFront.

| Model | Structure | Text | Image | Form |
|---|---|---|---|---|
| Qwen2.5-VL-72B-Instruct | 0.51 | 0.19 | 0.54 | 0.41 |
| InternVL2.5-78B | 0.43 | 0.11 | 0.52 | 0.34 |
| InternVL3-78B | 0.49 | 0.15 | 0.61 | 0.42 |
| LLaVA-Onevision-72B | 0.29 | 0.06 | 0.38 | 0.23 |
| Claude 3.7 Sonnet | **0.72** | _0.39_ | 0.65 | _0.53_ |
| Gemini 2.5 Flash | _0.70_ | **0.40** | **0.71** | 0.51 |
| GPT-4o | 0.42 | 0.11 | 0.53 | 0.34 |
| o4-mini | 0.62 | 0.28 | 0.61 | _0.54_ |
| GPT-4.1 | 0.63 | 0.27 | 0.64 | **0.64** |
| GPT-5 | 0.63 | 0.24 | _0.70_ | 0.53 |

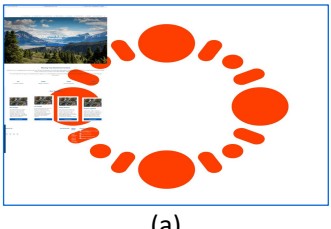 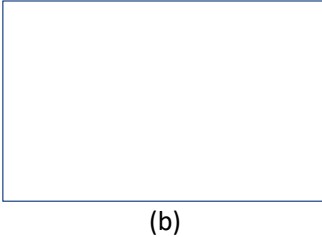 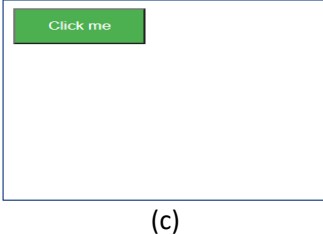

(a)             (b)             (c)

Figure 5: Three common errors in Webpage Code Generation. (a) Abnormal Image Sizes, where an image within the rendered page is disproportionately large. (b) Blank Page, showing an entirely blank rendered output. (c) Isolation Error, an output consisting only of an isolated interactive element.

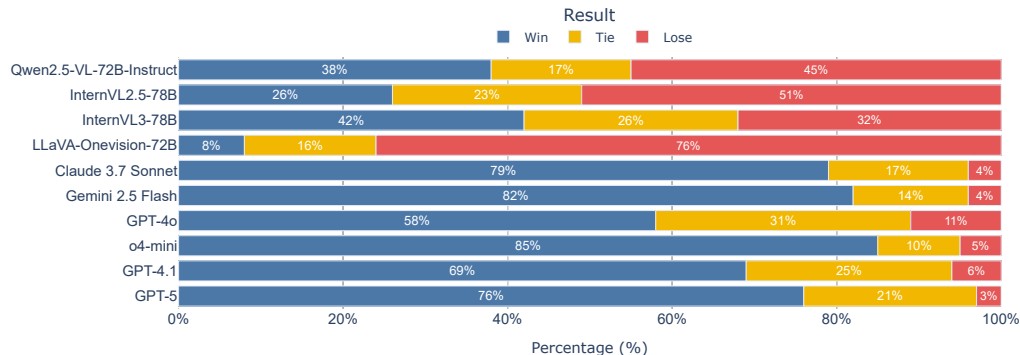

Figure 6: Human evaluation comparing MLLMs-generated and Real-World webpages.

substantial room for improvement in text and form implementation, as similarity scores for these components do not exceed 0.65.

Overall, despite certain shortcomings in fine-grained details, MLLMs do demonstrate the capability to design generally coherent webpage interfaces from textual descriptions and can generate corresponding code from webpage screenshots. However, the overall deficiencies in their perceptual abilities, coupled with the potential for critical errors during code generation, render their current reliability and stability uncertain. We believe a promising future direction involves integrating MLLMs with specialized tools. This approach could compensate for their perceptual limitations and provide mechanisms to identify and rectify generation anomalies, thereby aiding MLLMs in their evolution towards becoming excellent front-end engineers.

## 6 CONCLUSION

We introduce **FullFront**, a pioneering and comprehensive Multimodal Front-end Benchmark. Full-Front is designed to systematically evaluate the capabilities of Multimodal Large Language Models (MLLMs) across the entire front-end development pipeline, encompassing key aspects such as design, page perception, and code generation. By constructing high-quality, diverse synthetic data and designing a multi-layered evaluation system, FullFront serves as a powerful tool for in-depth analysis of the strengths and limitations of current MLLMs. While no single current model has yet mastered the entire workflow, FullFront provides the necessary infrastructure to evaluate future "Generalist Front-End Agents." It establishes a standard for tracking the evolution of MLLMs from isolated task solvers to comprehensive engineering assistants, laying the foundation for the development of the next generation of intelligent webpage development tools.

## ETHICS STATEMENT

This research strictly adheres to the ICLR Code of Ethics. Throughout our dataset construction process, all data underwent a rigorous manual review to remove any potentially harmful content or personal private information. This ensures the protection of individual privacy and prevents the propagation of sensitive data.

## REPRODUCIBILITY STATEMENT

To ensure full reproducibility of our findings, we will include all evaluation code in the Supplementary Material. In accordance with the principles of open research and to respect the double-blind review process, we commit to making our entire dataset publicly available upon the acceptance of this paper.

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

## A  APPENDIX OVERVIEW

This Appendix is organized as follows:

- **Section B** details the use of LLMs in our work.
- **Section C** provides a detailed description of the FullFront-mini dataset, a condensed subset of the main benchmark designed for rapid evaluation.
- **Section D** offers additional qualitative examples for the Webpage Design task, providing a clearer view of the capabilities and limitations of current text-to-image models.
- **Section E** presents illustrative case studies for each subtask within the Webpage Perception QA task, highlighting common model failures and their underlying cause
- **Section F** elaborates on the four Webpage Code Generation subtasks, detailing the data curation process and presenting qualitative results that compare model outputs against ground-truth designs.
- **Section G** outlines our comprehensive evaluation framework, including the formulation and validation of the Gemini Visual Score and Code Score, MLLM experimental settings, and the human evaluation protocol.
- **Section K** contains the complete, verbatim prompts used for both data curation and model evaluation across all tasks in the FullFront benchmark.

## B  USE OF LARGE LANGUAGE MODELS

During the research process, a Large Language Model (LLM) was utilized for text polishing to improve clarity and grammar. The use of the LLM was confined to writing assistance and did not extend to the formulation of research methodologies or core ideas. Furthermore, the specific applications of Multimodal Large Language Models (MLLMs) within our dataset construction process are thoroughly detailed in the main body of the paper and the Appendix.

## C  FULLFRONT-MINI

**FullFront-mini Dataset** To facilitate rapid iterative evaluation of MLLMs and initial exploration of the benchmark by researchers, we constructed a FullFront-mini dataset. This subset is a condensed version of the full FullFront dataset, with the following specific composition. Webpage Perception QA: Includes 200 Real-world QA, 100 Synthetic QA, and 50 Multi-window QA data samples. Webpage Code Generation: Comprises 50 Image to Code, 50 Text to Code, 50 Interaction Authoring, and 50 Code Refinement data samples. Webpage Design: Consists of 50 Webpage Design task data samples.

## D  WEBPAGE DESIGN

The Webpage Design task evaluates an MLLM's ability to generate a visual webpage design based on a textual description, assessing its capacity for conceptualization within the front-end workflow. Figure 7 illustrates the outputs from two evaluated text-to-image MLLMs, GPT-4o and gemini-2.0-flash-exp-image-generation, alongside the target "Label Image" (ground truth) for a representative example.

As observed in Figure 7, the design generated by GPT-4o (right) demonstrates a notably closer resemblance to the "Label Image" (left) compared to the output from gemini-2.0-flash-exp-image-generation (middle). Specifically:

- **Layout and Structure:** GPT-4o more successfully replicates the overall page structure, including the header, hero section, "Popular Categories" grid, and footer arrangement. The placement and relative sizing of these major components are more aligned with the ground truth. In contrast, the gemini-2.0-flash-exp-image-generation produces a layout that, while containing some similar thematic elements (like a search bar and category-like items), deviates more significantly in its structural organization and visual hierarchy.

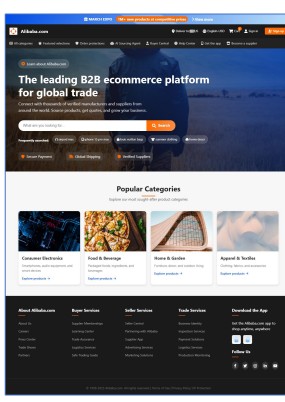 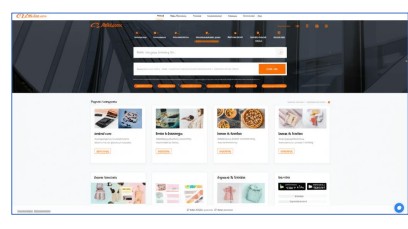 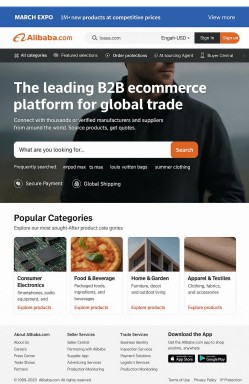

**Label Image**          **gemini-2.0-flash-exp-image-generation**          **GPT-4o**

Figure 7: Comparative Webpage Designs: Ground Truth ("Label Image") vs. gemini-2.0-flash-exp-image-generation and GPT-4o.

- **Element Completeness and Typography:** GPT-4o tends to generate a design with a higher degree of element completeness. For example, the navigation links in the header, the search bar within the hero section, and the individual category cards appear more fully formed and are stylistically closer to the target. The typography choices in GPT-4o's output also generally exhibit greater fidelity.
- **Detail Discrepancies:** Despite its superior overall performance, the GPT-4o design still exhibits discrepancies in fine-grained details. For instance, the footer section in the GPT-4o output uses a light background, contrasting with the dark background of the "Label Image" footer.

Additional qualitative results are provided in Figure 8. To summarize, the qualitative examples indicate that while text-to-image MLLMs such as GPT-4o are capable of producing webpage designs that align with the primary layout and components of a textual prompt, they still lack the fine-grained control necessary to precisely render all visual attributes. Details like exact color codes, specific text, and subtle styling elements present a significant opportunity for future research. The models excel at converting abstract concepts into concrete webpage structures, but their capacity to follow detailed, nuanced instructions needs to be enhanced.

## E    WEBPAGE PERCEPTION QA

In this section, we provide a detailed breakdown of the three subtasks that constitute the Webpage Perception QA benchmark: Real-world QA, Synthetic QA, and Multi-window QA. For each subtask, we present an illustrative case study that highlights common model failures. Through an analysis of these specific examples, we aim to shed light on the root causes of these errors and pinpoint the underlying perceptual limitations of current MLLMs.

### E.1    REAL-WORLD QA

The Real-world QA subtask is designed to rigorously evaluate an MLLM's ability to comprehend visual information from authentic webpage screenshots. This subtask consists of 1,250 question-answer pairs derived from 625 real-world webpages. These screenshots are sourced from a combination of manual collection and established academic benchmarks, including Uground (Gou et al., 2024) and IW-Bench (Guo et al., 2024). The primary challenge of this subtask lies in the complexity and subtlety of the questions, which demand a high degree of perceptual acuity to correctly identify fine-grained details regarding element positioning, styling, and spatial relationships.

Figure 9 presents a representative example, where MLLMs are tasked with assessing the positional and stylistic relationship between the "Enroll for free" and "View details" buttons. In this instance, all

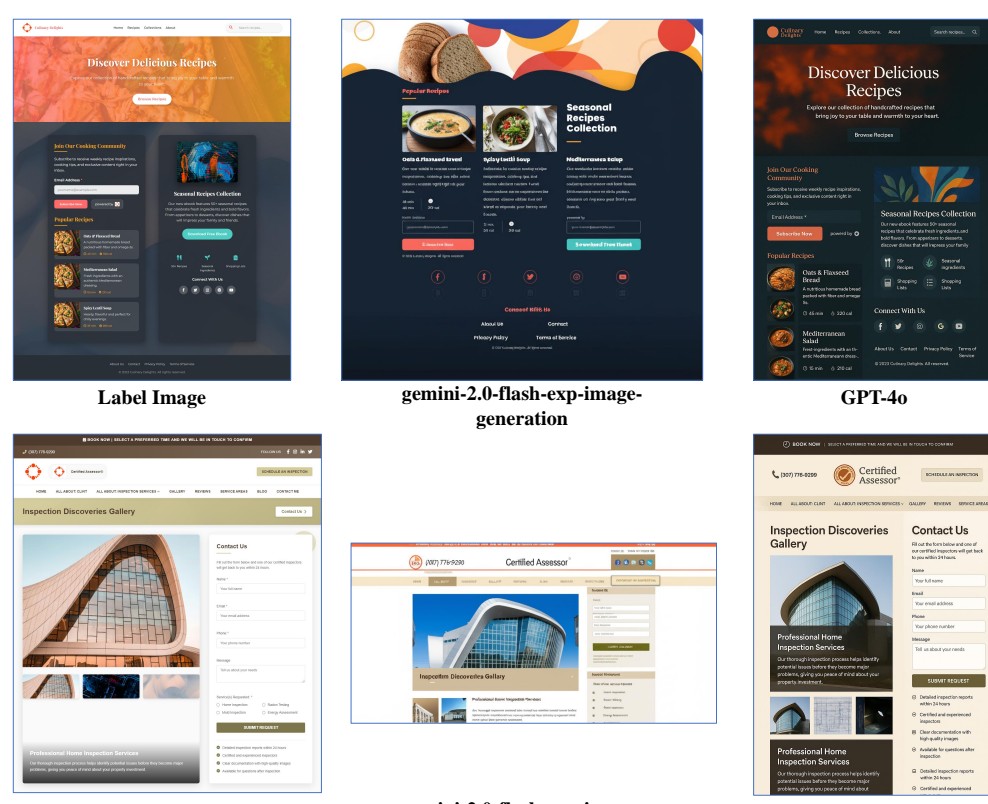

Figure 8: Results of GPT-4o and gemini-2.0-flash-exp-image-generation on the Webpage Design task.

evaluated MLLMs converged on an incorrect answer (A). Their choice indicates that they successfully perceived coarse-grained information: that the "Enroll for fre" button is located to the left of "View details" and features a solid blue background. However, their analysis is flawed, as they incorrectly inferred that it was positioned vertically higher than the other button.

Crucially, the models failed to identify the subtle but correct detail required for answer C: that both buttons share an identical font weight. This case study starkly highlights a critical limitation in current MLLMs. While they can often process general layout and prominent visual cues, they struggle with the precise perception of element alignment and nuanced typographic styles, which are fundamental for high-fidelity front-end development.

### E.2 SYNTHETIC QA

To complement the Real-world QA, the Synthetic QA subtask assesses model performance in a more controlled setting. This subtask consists of 400 question-answer pairs generated from 200 unique webpage screenshots. These screenshots are not sourced from the wild; instead, they are products of the synthesis pipeline detailed in the main paper. This methodology allows for the creation of webpages that, while visually diverse and realistic, can feature subtle and deliberate variations from standard design conventions. The key challenge, therefore, is to test whether MLLMs are truly perceiving the provided visual evidence or are relying on strong priors about common webpage layouts.

The example in Figure 10 effectively demonstrates this challenge, requiring models to compare the relative spacing and alignment of four social media icons. The analysis reveals two distinct modes of failure among the models:

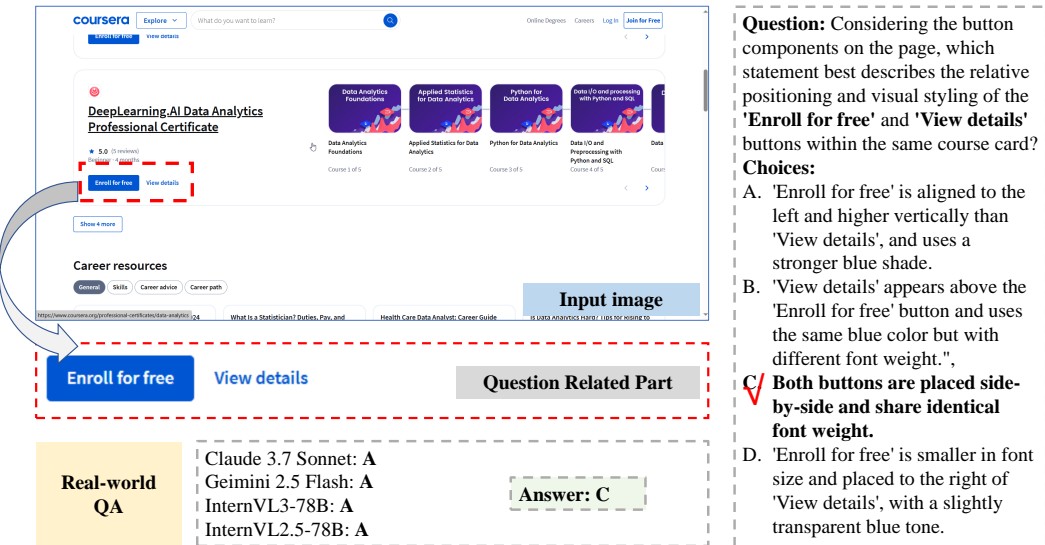

Figure 9: A sample case in Real-world QA.

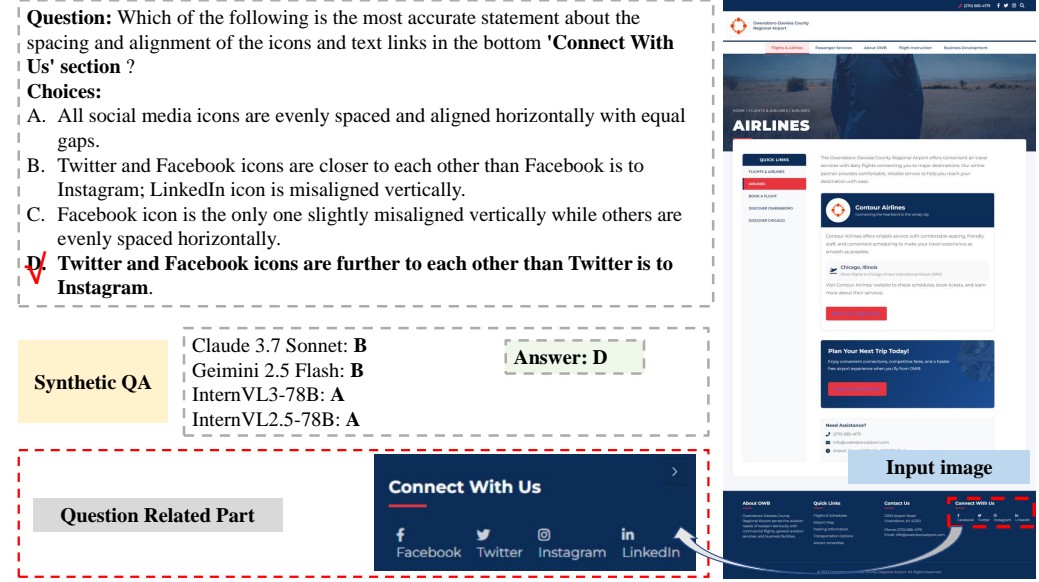

Figure 10: A sample case in Synthetic QA.

- **InternVL2.5-78B and InternVL3-78B** incorrectly concluded that the icons are evenly spaced (Option A). This error likely stems from a powerful prior, as equidistant spacing is a highly common design pattern. The models appear to have defaulted to this assumption—a form of model hallucination—instead of precisely analyzing the visual data.

- In contrast, **Claude 3.7 Sonnet and Gemini 2.5 Flash** exhibited a more nuanced partial understanding, correctly identifying the varied spacing among the first three icons. However, they also failed by incorrectly perceiving a vertical misalignment of the LinkedIn icon (Option B), leading them away from the correct answer, D.

This case study illustrates a critical pattern: even when MLLMs can successfully localize the specific region relevant to a question, their ability to perform a meticulous analysis of fine-grained spatial details within that region remains unreliable.

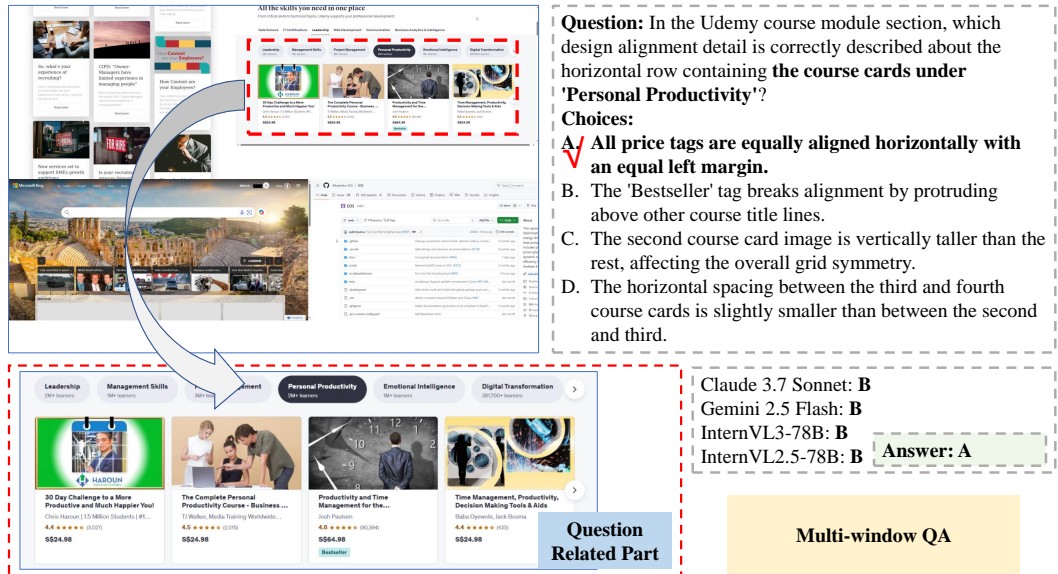

Figure 11: A sample case in Multi-window QA.

### E.3 MULTI-WINDOW QA

The Multi-window QA subtask significantly elevates the complexity to better simulate real-world scenarios where relevant information must be found across different contexts. This subtask is composed of 150 question-answer pairs distributed across 75 samples, where each sample presents a model with 2 to 4 distinct screenshots drawn from our Real-world QA set. The difficulty is twofold and distinguishes it from the previous subtasks: a model must first correctly localize the specific webpage screenshot relevant to the question, and only then perform the fine-grained perceptual analysis required to answer it. This tests both search and comprehension capabilities.

The case presented in Figure 11 highlights the compounded difficulty of this task. To answer a question about the layout of "Personal Productivity" course cards, the MLLMs must first identify the correct screenshot among the options. The results show a uniform failure, with all models incorrectly selecting option B. This choice reveals a significant perceptual error: the models incorrectly concluded that the "Bestseller" tag is positioned above the title lines, when in fact, the tag is located at the very bottom of the course card.

Simultaneously, this fixation on a misunderstood element caused them to overlook the correct statement in option A, which accurately describes that all price tags are equally aligned horizontally. This dual failure is revealing. It indicates that MLLMs not only face challenges in disambiguating and focusing on the correct visual context but also remain highly susceptible to misinterpreting spatial relationships even after localizing the right section. Their inability to perform systematic, holistic visual checks leads to errors in judgment.

## F WEBPAGE CODE GENERATION

### F.1 OMNIPARSER

OmniParser, a screen parsing tool developed by Microsoft, is engineered to deconstruct User Interface (UI) screenshots into structured and readily comprehensible elemental data. This capability significantly enhances the proficiency of visual models, such as GPT-4o, in generating accurate operational instructions within a given interface. As illustrated in Figure 12, OmniParser adeptly detects a wide array of fine-grained UI elements from screenshots, including diminutive icons, buttons, and text boxes.

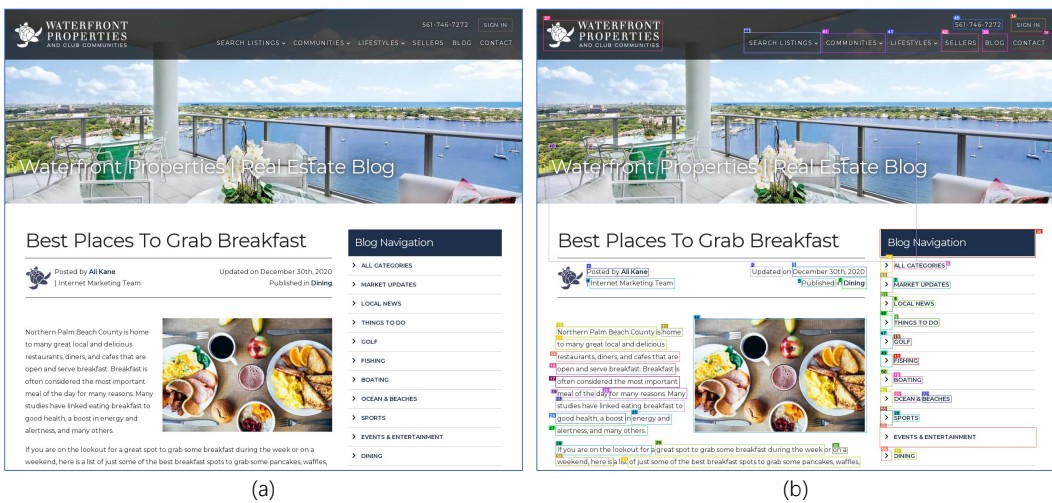

Figure 12: OmniParser extracting UI elements from a webpage screenshot. (a) Original webpage screenshot. (b) Webpage screenshot with UI elements detected and bounded by OmniParser.

The output from OmniParser typically includes the bounding box coordinates and the recognized textual content for each detected element on the original webpage. An example of such extracted information is provided below:

```
"bbox": [0.69, 0.546, 0.79, 0.562], "content": "ALL CATEGORIES"
"bbox": [0.096, 0.561, 0.215, 0.578], "content": "Posted by Ali Kane"
"bbox": [0.412, 0.558, 0.473, 0.579], "content": "Updated"
"bbox": [0.492, 0.558, 0.636, 0.579], "content": "December 30th 2020 2020"
"bbox": [0.102, 0.582, 0.266, 0.605], "content": "Internet Marketing Team"
"bbox": [0.51, 0.583, 0.576, 0.601], "content": "Published"
"bbox": [0.589, 0.582, 0.636, 0.605], "content": "Dining"
"bbox": [0.688, 0.59, 0.798, 0.604], "content": "MARKET UPDATES"
"bbox": [0.688, 0.631, 0.768, 0.645], "content": "LOCAL NEWS"
"bbox": [0.69, 0.671, 0.776, 0.687], "content": "THINGS To Do"
```

Within the FullFront benchmark's data curation pipeline, OmniParser serves as an auxiliary tool to facilitate the generation of high-quality training and evaluation instances. Specifically, it extracts elemental information (bounding boxes and content) from webpage screenshots. This extracted data is then leveraged by GPT-4o, which is involved in the dataset creation process, for data generation.

More precisely, for the Webpage Perception QA Task Data Generation, GPT-4o is provided with the original webpage screenshot, the same screenshot overlaid with OmniParser's detected bounding boxes, and the structured data from OmniParser (which includes the coordinates of these boxes and their corresponding textual content). This comprehensive input is designed to alleviate the perceptual load on GPT-4o, allowing it to concentrate on formulating challenging Question-Answer (QA) pairs.

Similarly, for the Webpage Code Generation Task Data Generation, this same set of information (original screenshot, screenshot with bounding boxes, and OmniParser's structured box/text data) is supplied to GPT-4o. This assists GPT-4o in generating an initial HTML representation (HTML-v1) as a foundational step in reconstructing the webpage.

## F.2 CATEGORY-BASED UTILIZATION STRATEGY FOR IMAGES

Regarding images, instead of using simple placeholders, we employ a category-based utilization strategy. We classify common real-world image content into 15 predefined categories: People, Animal, Food, Plant, Landscape, Icon, Logo, Architecture, Technology, Transportation, Map, Texture, Art, Movie, and Other (visualized in Figure 13). Each category is linked to a fixed, non-copyrighted

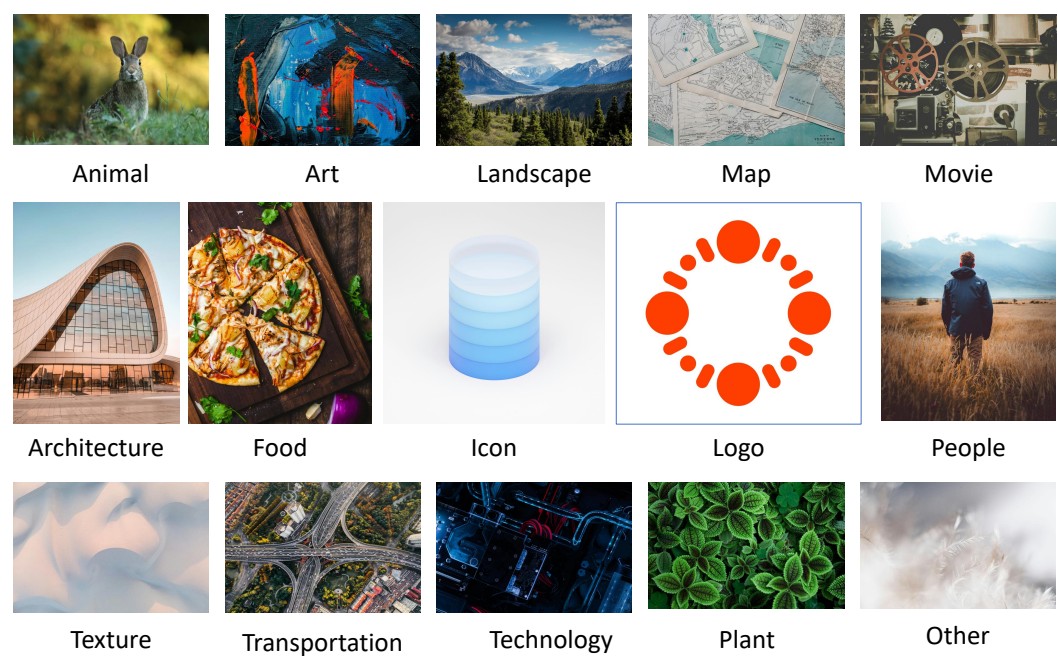

Figure 13: The 15 predefined image categories used in FullFront for standardized image representation.

image URL following a standardized format, such as "https://fixed_part/{Category}.jpg". During the ground truth generation for code tasks, GPT-4o and Claude 3.7 Sonnet are instructed to select an appropriate category for any required image and use its corresponding URL. For evaluation, when an MLLM is tasked with generating webpage code, it must understand the image content from the provided screenshot, classify it into one of these 15 categories, and then generate HTML using the correct category-specific URL. Furthermore, because the intrinsic dimensions of these repository images are unknown, the MLLM is explicitly required to manually set the image sizes (width and height) and position within the HTML code to ensure the rendered output matches the layout depicted in the screenshot. This approach assesses MLLMs' capabilities in image perception, categorization, and appropriate styling. It also ensures visual consistency for subsequent evaluation steps. Crucially, by deriving visual designs from real-world screenshots, our method generates webpages with greater diversity compared to "from scratch" techniques. This strategy ingeniously bypasses the laborious simplification of real-world code while still achieving simplification's primary goals—such as removing sensitive information and external dependencies—and preserving as much original visual information as possible through categorized representation. The use of category-specific image URLs also facilitates straightforward dataset expansion with new image types in the future.

### F.3 IMAGE TO CODE

The Image to Code subtask serves as the foundational evaluation for webpage implementation. It is designed to assess an MLLM's core ability to translate a static visual design into functional HTML code. The subtask comprises 200 distinct samples. During the evaluation, each MLLM is provided with a single webpage screenshot, which has been rendered from the high-fidelity HTML-v2 ground truth. The objective is to generate code that accurately replicates the visual appearance of the given image. This is the most direct and fundamental screenshot-to-code challenge within our benchmark, establishing a baseline for model performance.

Figure 14 and Figure 15 provide qualitative examples of various MLLM outputs on this task. The "Label Image" in each figure is the ground-truth screenshot provided as input to the models. A visual inspection reveals a clear performance gap between proprietary and open-source models.

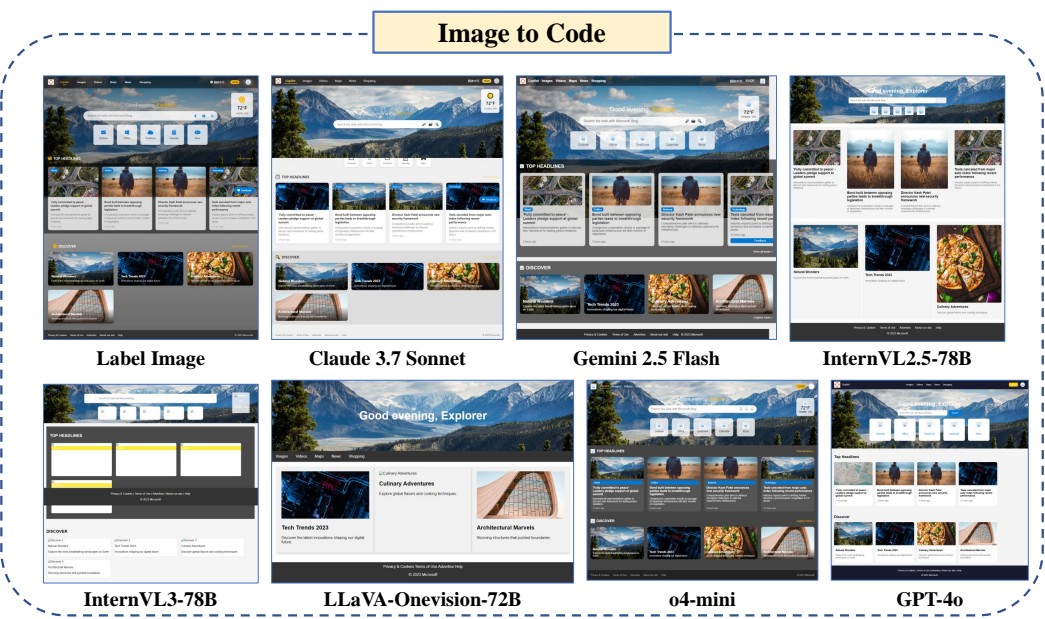

Figure 14: A sample case in Image to Code.

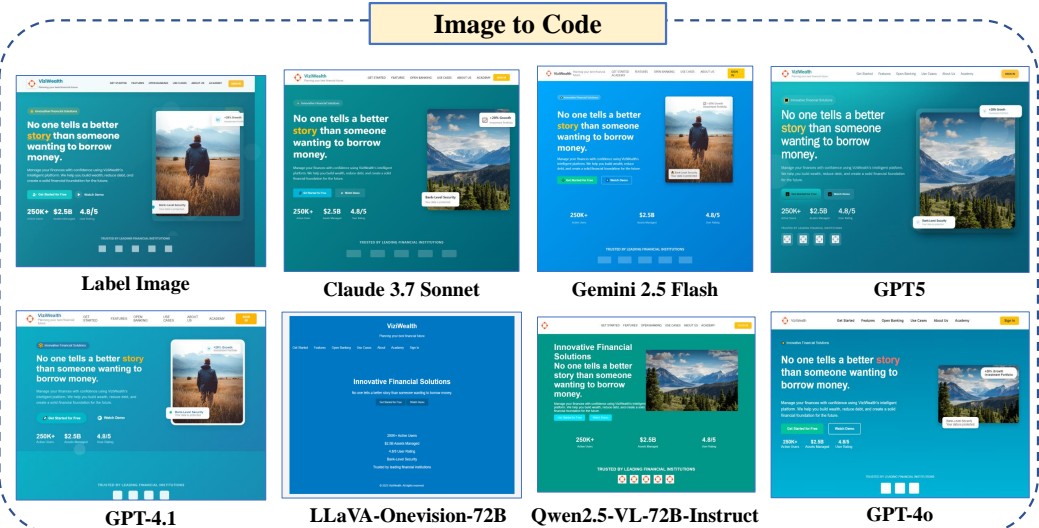

Figure 15: A sample case in Image to Code.

Leading models like Claude 3.7 Sonnet, Gemini 2.5 Flash, and GPT-5 demonstrate a strong capability to grasp and reproduce the overall page layout, color schemes, and general style. However, they are not without flaws and often fail on fine-grained details. For example, as shown in Figure15, Claude 3.7 Sonnet incorrectly identifies the image category, while Gemini 2.5 Flash fails to apply the correct background color to the page.

In contrast, the performance of the open-source models is generally less consistent. As illustrated across both figures by models like LLaVA-Onevision-72B and InternVL3-78B, there is substantial room for improvement in fundamental areas such as overall layout integrity, the correct implementation of headers and footers, and the accurate rendering of text and image elements.

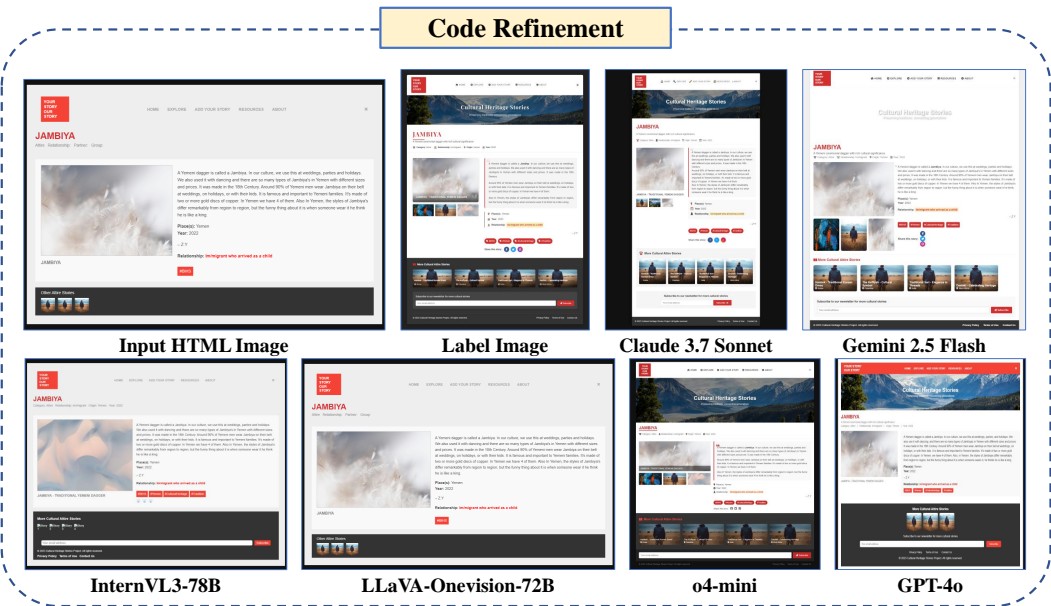

Figure 16: A sample case in Code Refinement.

### F.4 CODE REFINEMENT

The Code Refinement subtask is designed to simulate a realistic and critical developer workflow: enhancing and optimizing existing code. Comprising 200 samples, This task evaluates an MLLM's ability to modify and augment existing webpage code in order to achieve a target visual design. In this scenario, the model is provided with the HTML-v1 along with the Label Image rendered from the corresponding HTML-v2. The model's objective is to modify and augment the provided HTML-v1 code to match the superior visual quality and complexity of the Label Image.

As illustrated in the examples in Figure 16 and Figure 17, the results highlight a significant performance disparity. Advanced closed-source models, notably Claude 3.7 Sonnet, o4-mini, and GPT-5, demonstrate a strong capability to perform this complex task. They successfully identify the discrepancies between the input and target visuals and apply appropriate changes to the base code to closely reproduce the Label Image. While their outputs are highly accurate, there remains some room for improvement in perfecting fine-grained details related to headers, footers, text content, and element proportions.

In stark contrast, many open-source models exhibit a more fundamental difficulty with the task's core instruction. Rather than refining the provided HTML-v1, models such as InternVL3-78B and LLaVA-Onevision-72B often disregard the modification requirement entirely. Their outputs tend to be a near-verbatim copy of the initial code provided, showing minimal or no changes based on the target Label Image. This indicates a significant challenge not only in their code generation abilities but also in their capacity to comprehend and execute complex instructions that involve editing, rather than simply generating, code.

### F.5 TEXT TO CODE

The Text to Code subtask evaluates an MLLM's ability to function as both a designer and a developer by generating a complete webpage based solely on a natural language description. This subtask includes 200 samples. For each, a detailed textual description was generated by Claude 3.7 Sonnet based on a high-fidelity HTML-v2 rendered page, followed by a rigorous manual review to ensure accuracy and quality. During evaluation, models are provided only with this text, making this a "blind" generation task where no visual reference is available. The goal is to translate the abstract requirements into a concrete and visually accurate webpage.

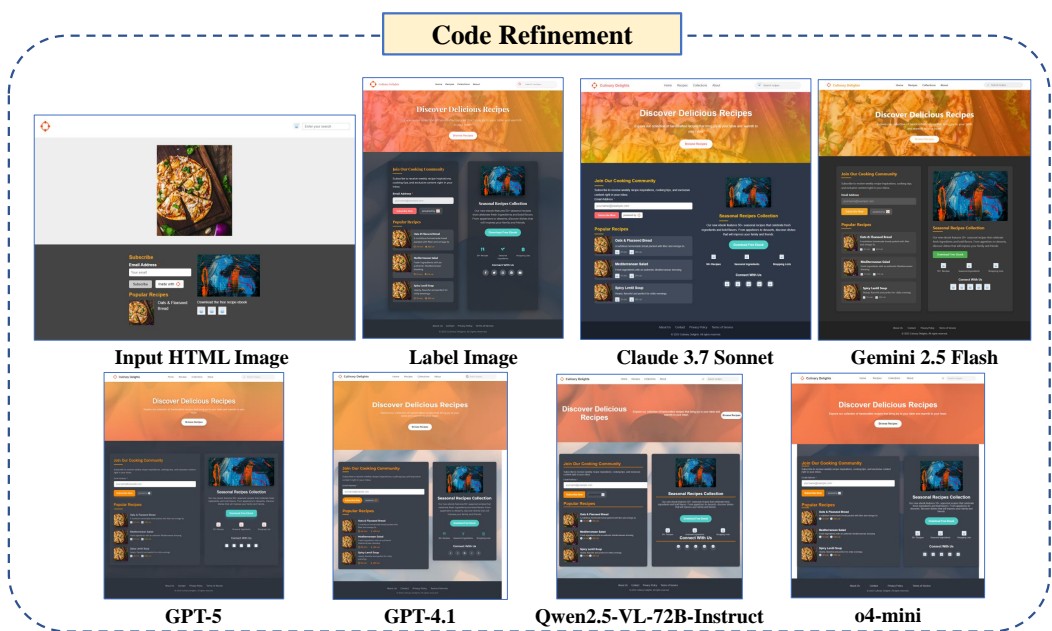

Figure 17: A sample case in Code Refinement.

The qualitative results are presented in Figure 18 and Figure 19. It is important to note that the "Label Image" is shown here only for reference; it represents the original webpage that was described but was not provided to the models during the test.

The examples reveal two key findings. First, when faced with a more unique design, such as the one in Figure 18, the models successfully capture the main theme—a page with a large, food-themed background. However, they share a common, subtle error: rendering the header with a solid white background instead of the correct transparent one. This suggests that while high-level concepts are well-understood, models may default to common design patterns for details not explicitly or strongly specified. Second, for webpages with more conventional, standardized layouts, as seen in Figure 19, the models perform remarkably well. Most are able to accurately reproduce the page's structure, layout, and style from the text alone. This demonstrates their significant potential in the front-end development workflow, particularly for users without professional design or coding expertise. The ability to generate high-quality prototypes directly from a textual brief can dramatically lower the barrier to web creation and accelerate the initial stages of development.

### F.6 INTERACTION AUTHORING

Moving beyond static page generation, this task evaluates MLLMs' ability to implement dynamic, interactive webpages. Inspired by Interaction2Code (Xiao et al., 2024), we define ten common interaction types, categorized under click and hover events. For data construction, 100 samples derived from the static HTML-v1 (allowing models to focus primarily on interaction logic) are augmented with interaction code by Claude 3.7 Sonnet, followed by manual verification.

During testing, MLLMs receive "before" and "after" interaction screenshots and must reproduce the static page while implementing the depicted interactive behavior. To facilitate automated interaction detection, MLLMs are instructed to assign the ID "`#InteractionPart`" to the primary HTML element involved. The ten defined interaction types, with specific implementation requirements for each, are:

1. **Click to Display Dropdown (`Interaction_click_1`):** An element, when clicked, reveals a dropdown menu whose content, position, and style are contextually adapted. Requires `aria-expanded` attribute toggling and specific dropdown selectors.

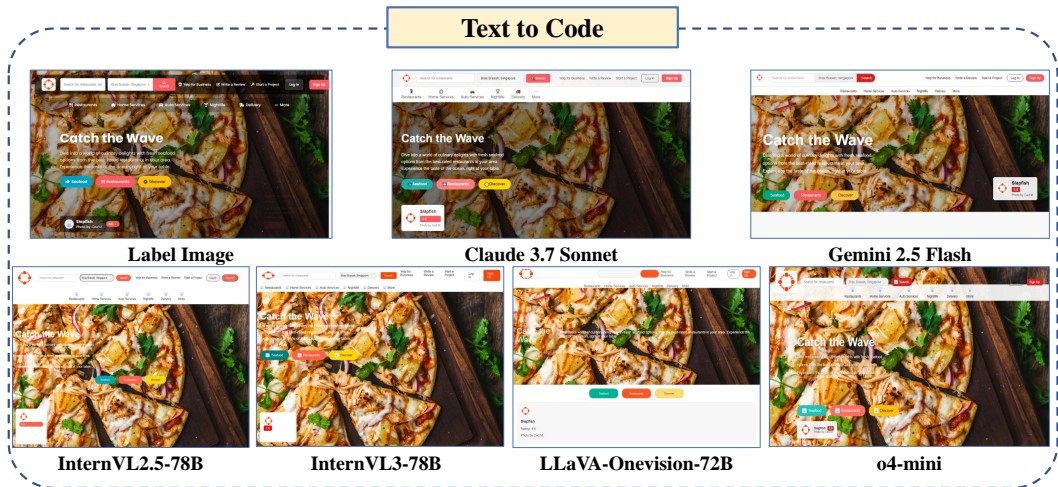

Figure 18: A sample case in Text to Code.

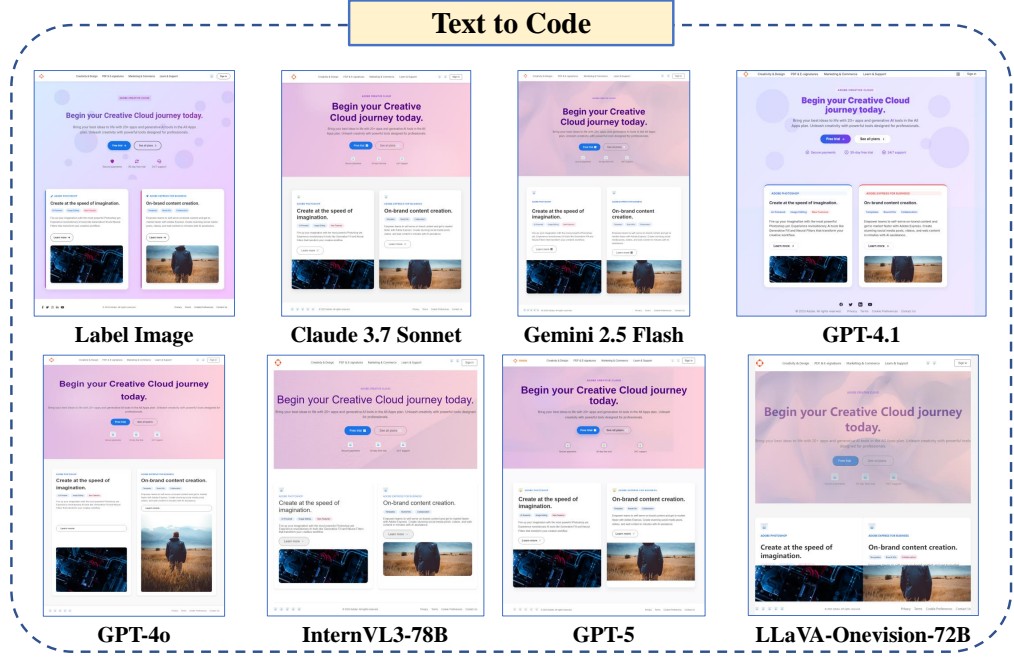

Figure 19: A sample case in Text to Code.

2. **Click to Toggle Checkbox (`Interaction_click_2`):** A clickable checkbox that toggles its checked/unchecked state. Must use an `<input type="checkbox">` or an element with `role='checkbox'`, displaying a checked state after interaction.

3. **Click to Change Background Color (`Interaction_click_3`):** An element significantly changes its background color upon being clicked, with the new color being distinct and detectable.

4. **Click to Display Modal/Dialog (`Interaction_click_4`):** Clicking an element triggers a modal window or dialog box with contextually generated content and styling. The modal must match specific selectors like `.modal` or `[aria-modal='true']`.

5. **Click to Display Tooltip (`Interaction_click_5`):** An element, when clicked, displays a tooltip providing additional information. The tooltip must adhere to specified class names or attributes (e.g., `.tooltip`, `[role='tooltip']`).

6. **Click to Display Text Input (`Interaction_click_6`):** Clicking an element reveals a text box or input area for user entry, appropriately sized and positioned.

7. **Hover to Display Dropdown (`Interaction_hover_1`):** A dropdown menu appears when the mouse hovers over an element, with adaptive content and styling.

8. **Hover to Bold Text (`Interaction_hover_2`):** Text within an element becomes bold (fontWeight $\geq$600 or 'bold'/'bolder') upon mouse hover.

9. **Hover to Underline Text (`Interaction_hover_3`):** Text within an element gains an underline when hovered over, with the computed `textDecoration` including "underline".

10. **Hover to Display Tooltip (`Interaction_hover_4`):** A tooltip with additional information appears when the mouse hovers over an element, conforming to specific class or attribute requirements.

The models must determine the correct interaction type from the visual cues and implement it according to these detailed specifications, providing the complete HTML, CSS, and JavaScript in a single file. Our analysis of the success rates (Table 10) revealed varying capabilities among the tested models. The MLLMs demonstrated the highest proficiency in implementing `Interaction_click_1` (**Click to Display Dropdown**), where, on average, models successfully realized this functionality in 8.4 out of the 10 test instances. Conversely, the most challenging categories proved to be `Interaction_hover_2` (**Hover to Bold Text**) and `Interaction_hover_1` (**Hover to Display Dropdown**). Notably, for these two hover-activated interaction types, no single model managed to successfully implement all 10 test cases.

Table 10: Performance of MLLMs across ten distinct interaction categories (C1-C6 for click-based, H1-H4 for hover-based) in the Interaction Authoring subtask. Each category comprises 10 test cases.

| Model | C1 | C2 | C3 | C4 | C5 | C6 | H1 | H2 | H3 | H4 |
|---|---|---|---|---|---|---|---|---|---|---|
| Qwen2.5-VL-72B-Instruct | 9 | 7 | 6 | 7 | 5 | 8 | 3 | 2 | 2 | 8 |
| InternVL2.5-78B | 7 | 6 | 6 | 8 | 4 | 6 | 0 | 2 | 4 | 4 |
| InternVL3-78B | 7 | 3 | 5 | 7 | 6 | 4 | 1 | 7 | 1 | 7 |
| LLaVA-Onevision-72B | 8 | 0 | 7 | 0 | 0 | 0 | 0 | 0 | 0 | 1 |
| Claude 3.7 Sonnet | 9 | 8 | 5 | 10 | 9 | 9 | 6 | 3 | 9 | 10 |
| Gemini 2.5 Flash | 7 | 9 | 6 | 9 | 8 | 9 | 6 | 1 | 8 | 7 |
| GPT-4o | 10 | 8 | 8 | 10 | 8 | 7 | 9 | 6 | 5 | 9 |
| o4-mini | 9 | 10 | 10 | 10 | 8 | 10 | 9 | 8 | 10 | 9 |
| GPT-4.1 | 10 | 5 | 8 | 10 | 8 | 9 | 7 | 8 | 3 | 10 |
| GPT-5 | 8 | 9 | 8 | 10 | 10 | 10 | 8 | 9 | 10 | 10 |
| Average | 8.4 | 6.5 | 6.9 | 8.1 | 6.6 | 7.2 | 4.9 | 4.6 | 5.2 | 7.5 |

The difficulty of this subtask is underscored by the qualitative results shown in Figure 20 and Figure 21. A critical observation is that the added complexity of implementing interaction often degrades the model's performance in rendering the static base page. Even though the static portion is based on the simpler HTML-v1, the visual fidelity of the replication is frequently lower than in static-only tasks. This suggests the cognitive load of generating interaction logic detracts from the model's ability to achieve precise visual accuracy. Furthermore, even when models successfully implement the correct interactive behavior (e.g., displaying a modal on click or a tooltip on hover), the final visual appearance of the "after interaction" state often shows significant discrepancies when compared to the ground-truth screenshot. These combined failures—in both static replication and dynamic fidelity—demonstrate a key deficiency in current MLLMs when it comes to creating robust, interactive web experiences.

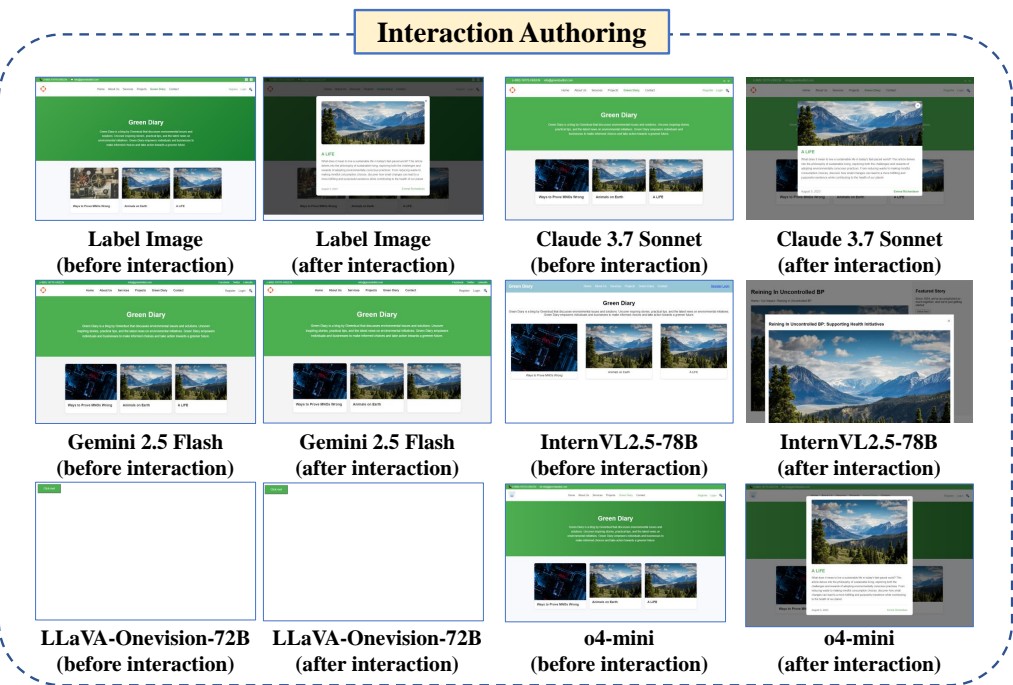

Figure 20: A sample case in Interaction Authoring.

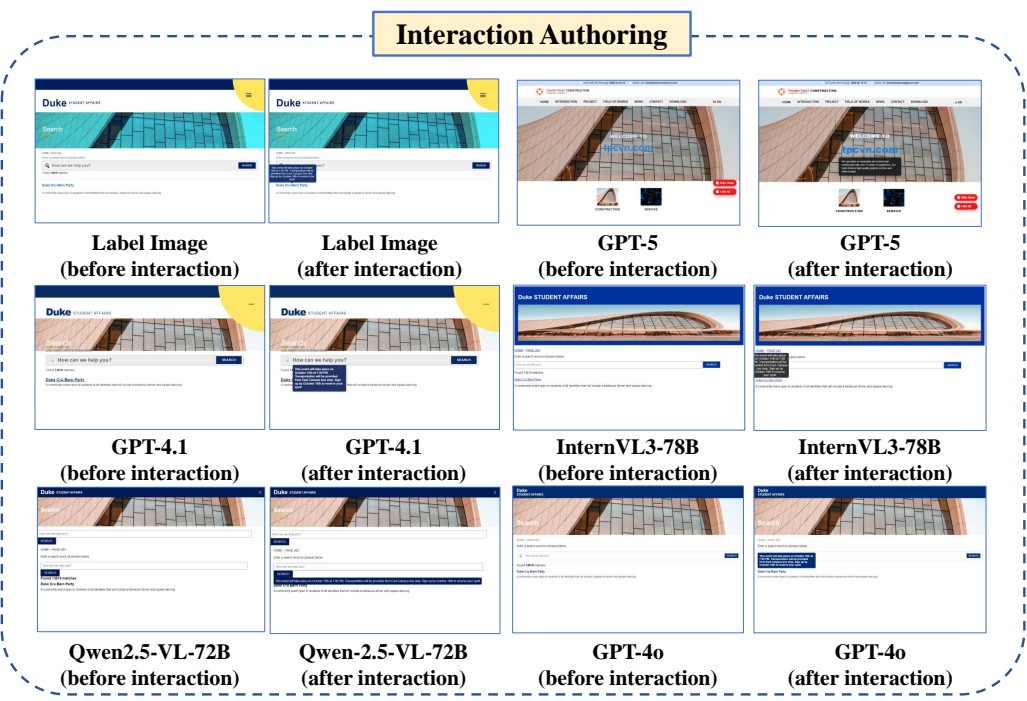

Figure 21: A sample case in Interaction Authoring.

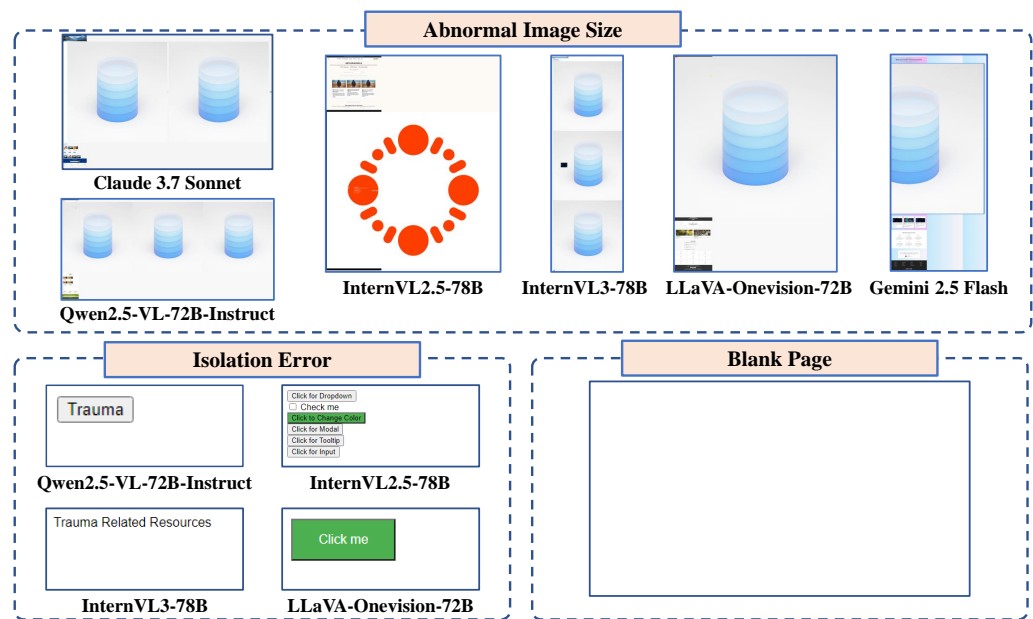

Figure 22: Three types code errors.

## F.7 THREE COMMON ERRORS

Across the Webpage Code Generation subtasks, we identified three recurring and critical failure modes that affect both open- and closed-source models: Abnormal Image Size, Blank Page, and Isolation Error. These errors are illustrated in Figure 22.

- **Abnormal Image Size:** This error occurs when rendered images, often several at once, are displayed at disproportionately large sizes, severely compromising the webpage's layout and aesthetics. Our analysis reveals this issue is most prevalent with images categorized as "Icon" and "Logo". We hypothesize this is because these image types are numerous, typically small, and can be easily overlooked by the model when applying explicit size constraints, unlike more prominent content images. Interestingly, models would sometimes correctly size several icons on a page but fail on a single one, indicating a lack of consistent style application.

- **Isolation Error:** This failure mode is unique to the Interaction Authoring subtask. It manifests as an output where the model generates only the code for the requested interactive element, completely omitting the surrounding static content of the webpage. This error points to a model's limited ability to handle complex, multi-layered instructions; it appears to fixate on the primary directive (implement the interaction) while ignoring the implicit but essential requirement to reproduce the entire page.

- **Blank Page:** This error results in a completely empty page upon rendering, even when the generated code is not empty. The root cause typically traces back to a common failure mode in Large Language Models: repetitive generation. The model enters a loop, outputting the same content or code block repeatedly, which produces malformed HTML that a web browser cannot successfully parse and render.

In summary, the consistent occurrence of these three error types demonstrates that current MLLMs, regardless of their origin, have yet to achieve the stability and reliability required of a professional front-end engineer. Overcoming these fundamental failure modes is a critical step toward their practical application in autonomous web development.

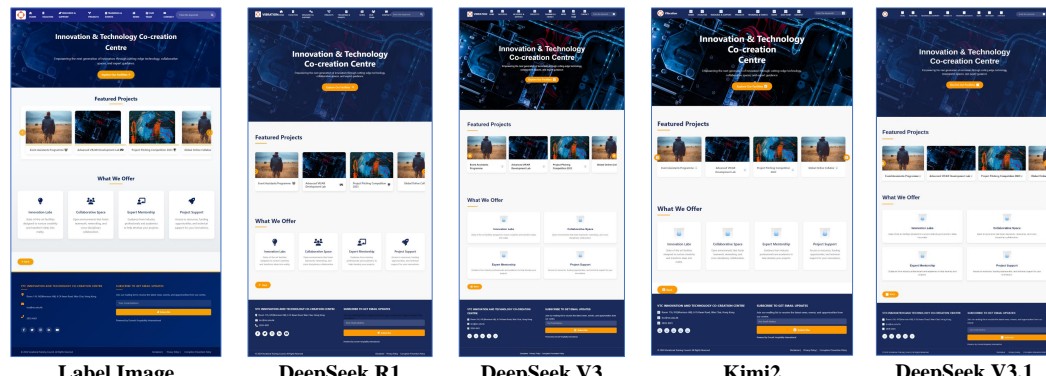

| Label Image | DeepSeek R1 | DeepSeek V3 | Kimi2 | DeepSeek V3.1 |

Figure 23: A qualitative comparison of webpages generated by leading text-only LLMs on the **Text to Code** subtask.

## F.8    TEXT-ONLY LLM RESULTS ON TEXT TO CODE

To explore the potential of powerful text-only LLMs in front-end development, we conducted a supplementary experiment on our **Text to Code** subtask, benchmarking models like the DeepSeek series (Guo et al., 2025) and Kimi K2 (Team et al., 2025) against the MLLMs from the main evaluation.

Table 11: Performance Comparison of MLLMs and Text-Only LLMs on the **Text to Code** Subtask.

| Model | Gemini Visual Score ↑ | Clip Score ↑ | Code Score ↑ | DINOv2 Score ↑ |
|---|---|---|---|---|
| Qwen2.5-VL-72B-Instruct | 5.1750 | 0.7313 | 0.4854 | 0.4665 |
| InternVL2.5-78B | 5.4444 | 0.7366 | 0.4989 | 0.4824 |
| InternVL3-78B | 4.9144 | 0.7204 | 0.4969 | 0.4405 |
| LLaVA-Onevision-72B | 5.1273 | 0.7365 | 0.4129 | 0.5006 |
| Claude 3.7 Sonnet | 8.1358 | 0.8655 | **0.6436** | 0.7790 |
| Gemini 2.5 Flash | 8.0111 | 0.8660 | 0.5927 | 0.7929 |
| GPT-4o | 6.1351 | 0.7364 | 0.4857 | 0.5176 |
| o4-mini | 6.8728 | 0.8197 | 0.5052 | 0.6807 |
| GPT-4.1 | 8.3555 | 0.8685 | 0.5509 | 0.7960 |
| GPT-5 | **8.4780** | **0.8759** | 0.5886 | 0.7928 |
| R1-Distill-Llama-8B | 2.9540 | 0.6991 | 0.4189 | 0.4014 |
| DeepSeek-V3 | 7.9985 | 0.8617 | 0.5885 | 0.7828 |
| DeepSeek-R1 | 7.9381 | 0.8607 | 0.6090 | 0.7487 |
| Kimi K2 | 7.7675 | 0.8431 | 0.5422 | 0.7323 |
| DeepSeek-V3.1 | 7.8775 | 0.8719 | 0.6050 | **0.7980** |

The results in Table 11 are compelling. Despite lacking visual input, leading text-only models like **DeepSeek-R1** and **DeepSeek-V3.1** demonstrate highly competitive performance, achieving **Code Scores** (0.6090 and 0.6050 respectively) that not only surpass all open-source MLLMs but also rival top-tier closed-source MLLMs such as 'Claude 3.7 Sonnet' (0.6436). Their strong performance is consistent across other visual and code-based metrics. This finding strongly suggests that with sufficiently detailed textual specifications, the advanced comprehension and coding abilities of text-only models can effectively compensate for the absence of a visual modality. As illustrated in Figure 23 and Figure 24, this highlights their significant potential in text-driven front-end development workflows, where they can successfully reproduce a webpage's layout, element positioning, and overall style.

## F.9    CORRECT CODE IMPLEMENTATION DESPITE PERCEPTUAL ERRORS

This part provides a visual illustration supporting the discussion in Section 5.2, which highlights an intriguing discrepancy between MLLM performance on perceptual QA tasks and their ability to generate visually accurate code. Specifically, as detailed in Figure 4 (b) of the main paper, all

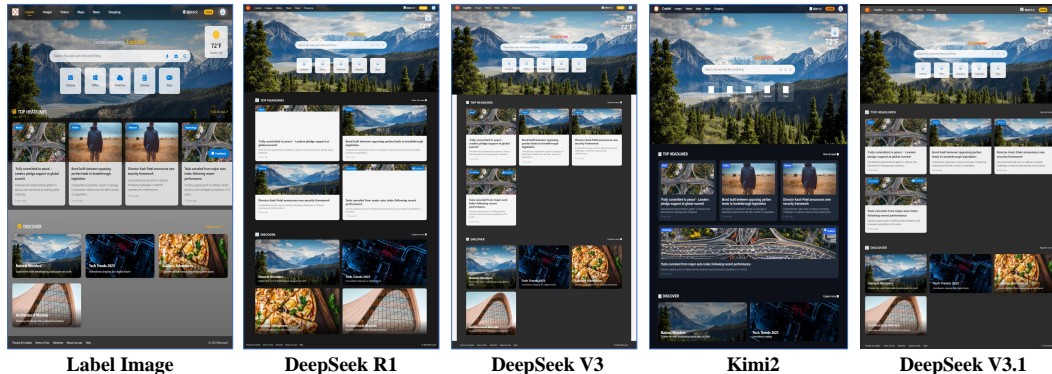

| Label Image | DeepSeek R1 | DeepSeek V3 | Kimi2 | DeepSeek V3.1 |

Figure 24: A qualitative comparison of webpages generated by leading text-only LLMs on the **Text to Code** subtask.

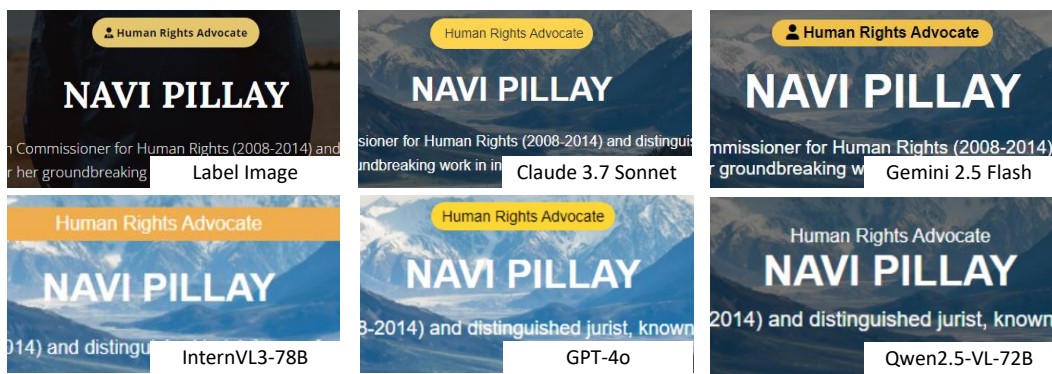

Figure 25: Rendered outputs from various MLLMs for the "NAVI PILLAY" webpage section. Despite failing the perceptual QA task regarding the tag's position relative to the title, all these MLLMs correctly implement the "Human Rights Advocate" tag above the main "NAVI PILLAY" title in their generated code.

evaluated MLLMs incorrectly identify the positioning of the "Human Rights Advocate" tag relative to the main title ("NAVI PILLAY") and subtitle in the Webpage Perception QA task. However, when these same MLLMs are tasked with generating the webpage code, they often demonstrate correct implementation of this very element's placement. Figure 25 presents the rendered outputs for the "NAVI PILLAY" webpage section from several MLLMs benchmarked in FullFront. These outputs are derived from the code generation tasks where models are asked to reproduce the webpage.

As can be observed in Figure 25, despite their prior failure in the specific perceptual QA question regarding the tag's position, all depicted MLLM outputs correctly place the "Human Rights Advocate" tag directly above the main "NAVI PILLAY" title. This placement is consistent with the ground-truth webpage.

F.10 DETAILED VISUAL ANALYSIS OF PERCEPTION-CODE QUADRANTS

To provide concrete evidence for the decoupling phenomenon discussed in Section 5.2, we present representative case studies for each of the four behavior quadrants. These examples illustrate how models succeed or fail in bridging the gap between visual perception (QA accuracy) and implementation (code generation). For clarity, we display cropped sections of the full webpages relevant to the specific QA pairs.

**Quadrant 1: Correct Perception, Correct Implementation** ($p\checkmark c\checkmark$). Figure 26 demonstrates the ideal scenario where models successfully ground their visual understanding into code. In the first example, GPT-5 accurately perceives the vertical alignment between the search input box and the

"About The Author" section, and faithfully reproduces this layout. Similarly, in the second example, GPT-5 correctly identifies and implements the left-alignment of the "Got a question?" headline relative to the search bar.

**Quadrant 2: Correct Perception, Incorrect Implementation ($p\checkmark c\times$).** Figure 27 highlights the "Coding Bottleneck" typical of open-source models. In the top example, Qwen2.5-VL-72B-Instruct correctly answers that "Arrangement Dimensions" is positioned to the left of "Care Instructions" on the same row. However, its generated code fails to achieve this layout, stacking the elements vertically instead. In the bottom example, InternVL3-78B correctly perceives the baseline alignment and height match of the search elements but erroneously inserts a checkbox between the input field and the button during implementation, breaking the intended visual structure.

**Quadrant 3: Incorrect Perception, Correct Implementation ($p \times c\checkmark$).** Figure 28 illustrates the "Grounding Failure" phenomenon prevalent in advanced proprietary models. In the first case, GPT-5 hallucinates a vertical spacing discrepancy in the navigation menu that does not exist (claiming "PORTFOLIO" to "BIOGRAPHY" is larger), yet its generated code produces perfectly even spacing. In the second case, GPT-5 incorrectly claims the category titles are indented to the right of the images. The ground truth (and the correct design principle) is center alignment, which the model successfully implements in its code despite the perceptual error. This strongly suggests reliance on internal design priors over direct visual grounding.

**Quadrant 4: Incorrect Perception, Incorrect Implementation ($p \times c\times$).** Figure 29 shows cases of complete failure. In the top example, GPT-4o misidentifies the styling of the "CUSTOM SAMPLE" button (claiming it has a border only), and proceeds to generate code where all buttons have filled backgrounds, contradicting the visual design. In the bottom example, GPT-4o not only fails to perceive the correct horizontal alignment of the search box relative to the author section but also fails to generate the search box entirely in the code, leading to a significant visual omission.

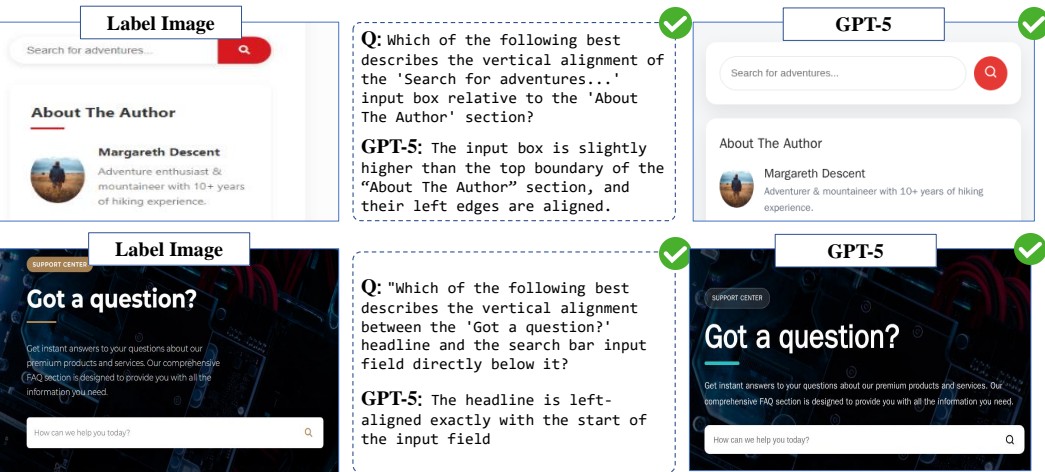

Figure 26: Quadrant 1 ($p\checkmark c\checkmark$): Examples where GPT-5 correctly perceives spatial relationships (top: vertical alignment; bottom: left alignment) and successfully implements them in code.

## G  EVALUATION

### G.1  GEMINI VISUAL SCORE: CRITERIA AND RUBRIC

To facilitate a fine-grained and human-aligned visual assessment of MLLM-generated webpages, we employ the **Gemini 2.5 Flash** model as a sophisticated visual evaluator. This model is tasked with comparing a rendered webpage image (generated by an MLLM) against its corresponding ground-truth image. To operationalize this evaluation, we designed a comprehensive rubric consisting of ten distinct visual dimensions. For each pair of images, the model provides a quantitative score on a scale of 0 to 10 for each dimension, where a score of 10 signifies perfect identity and a score of 0 indicates no discernible similarity.

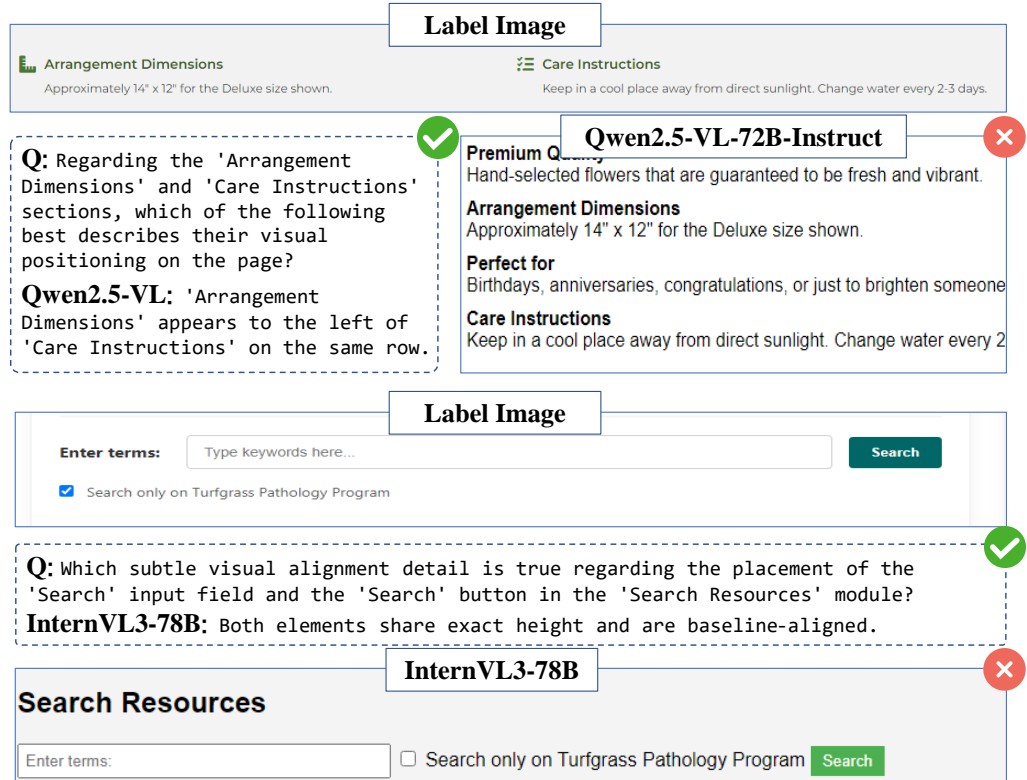

Figure 27: Quadrant 2 ($p\checkmark c\times$): Examples of Coding Bottlenecks. Qwen2.5-VL (top) and InternVL3 (bottom) correctly answer the perceptual questions but fail to translate this understanding into the correct layout or element arrangement.

The ten evaluation criteria are defined as follows:

1. **Element Reproduction:** Assesses whether all key visual elements (e.g., text, images, buttons, icons) are fully present and styled identically to the original.

2. **Proportion and Size Consistency:** Evaluates if the relative and absolute sizes of all elements match the original design, maintaining the intended visual harmony.

3. **Layout and Typography Fidelity:** Examines the faithful replication of the overall page structure (e.g., headers, footers, content grids) and the consistent application of typography (font families, weights).

4. **Alignment and Spacing Accuracy:** Measures the precision of element alignment (e.g., left, center, relative to other elements) and the consistency of spacing (margins, padding) compared to the ground-truth design.

5. **Visual Hierarchy Clarity:** Assesses if the generated webpage successfully maintains the same visual hierarchy as the original, guiding the user's attention to key information and calls to action similarly.

6. **Color Consistency:** Evaluates the match of the overall color scheme, including primary, secondary, and accent colors, as well as specific hues and saturation levels.

7. **Style Consistency:** Judges whether the overall aesthetic style of the generated webpage (e.g., modern, minimalistic, brutalist) aligns with the intended "look and feel" of the original design.

8. **Text Style Consistency:** Focuses specifically on typographic attributes such as font family, size, weight, color, line height, and text alignment.

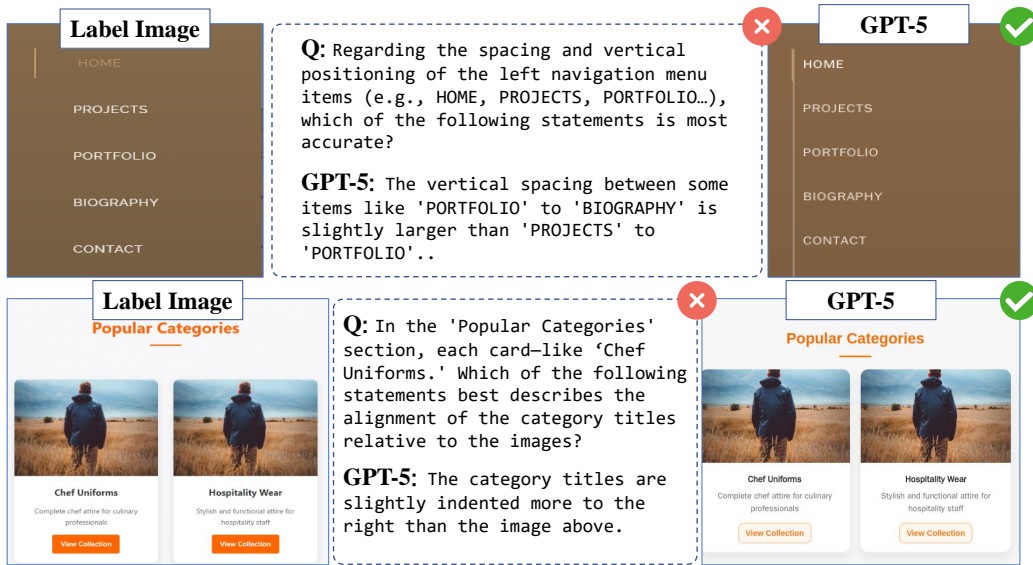

Figure 28: Quadrant 3 ($p \times c\checkmark$): Examples of Grounding Failure / Implicit Correction. GPT-5 hallucinates non-existent spacing (top) or alignment issues (bottom) in the QA phase but relies on strong priors to generate the correct, standard design patterns in the final code.

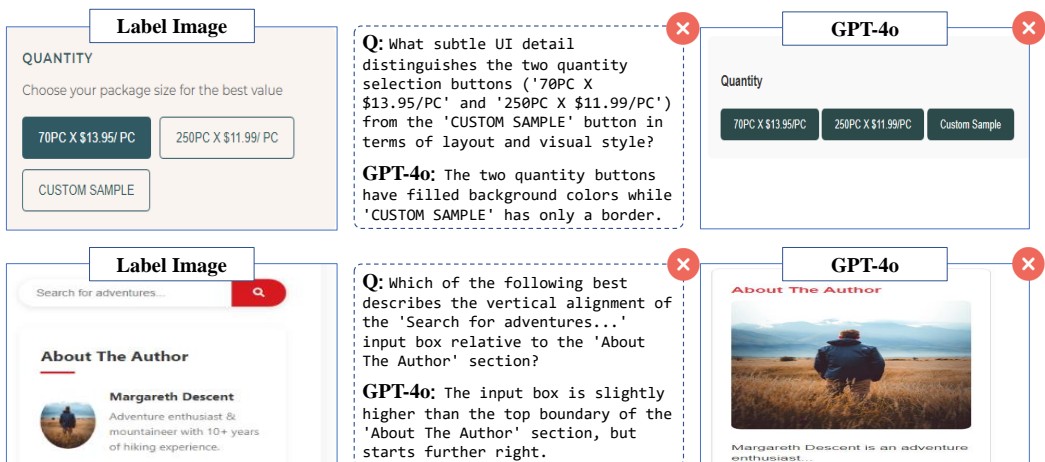

Figure 29: Quadrant 4 ($p \times c\times$): Examples of total failure. GPT-4o fails to perceive button styles (top) and element positioning (bottom), leading to code that either applies incorrect styles or omits key elements entirely.

**9. Text Content Accuracy:** Evaluates if the primary textual content (headings, body text, labels) accurately reproduces the text from the original design without omissions or significant alterations.

**10. Overall Content Representation:** A holistic measure of whether the generated page effectively conveys the same core information, message, and user intent as the original.

To apply these criteria and collect the scores, the prompt shown in Figure 30 was provided to the Gemini 2.5 Flash model.

**Prompt for Gemini Visual Score**

Your task is to assess two webpage images and output a score between 0 and 10 for each of the following 10 questions, reflecting the degree of similarity between the webpages. A score of 10 indicates perfect similarity (identical in every aspect), while a score of 0 indicates no similarity. For partial similarities, assign a score between 1 and 9, where higher scores reflect greater similarity. Only output a comma-separated list of 10 numbers enclosed in square brackets, e.g., [10,8,6,4,2,0,0,0,0,0]. Do not assign a score of 10 unless the two images are identical in the respective category.

1. **Element Reproduction (Score: 0-10):** Are key elements (text, images, buttons) fully present and styled identically to the original design? (e.g., 10 for identical elements, 5 for missing or slightly altered elements, 0 for completely different elements.)

2. **Proportion and Size Consistency (Score: 0-10):** Do the sizes and proportions of elements (text, images, buttons) match the original design, maintaining visual harmony? (e.g., 10 for exact proportions, 6 for minor size differences, 0 for significant discrepancies.)

3. **Layout and Typography Fidelity (Score: 0-10):** Does the overall layout (headers, footers, navigation bars, sidebars) faithfully replicate the original design's typography and structure? (e.g., 10 for identical layouts, 5 for similar but not exact placements, 0 for entirely different layouts.)

4. **Alignment and Spacing Accuracy (Score: 0-10):** Are elements aligned and spaced (margins, padding) as in the original design? (e.g., 10 for perfect alignment and spacing, 6 for minor misalignments, 0 for major misalignments.)

5. **Visual Hierarchy Clarity (Score: 0-10):** Does the webpage maintain the same visual hierarchy as the original, allowing users to quickly identify key information? (e.g., 10 for identical hierarchy, 5 for slightly altered emphasis, 0 for unclear or different hierarchy.)

6. **Color Consistency (Score: 0-10):** Do the overall color scheme, hues, and tones match the original design? (e.g., 10 for identical colors, 6 for similar palette with minor variations, 0 for completely different colors.)

7. **Style Consistency (Score: 0-10):** Does the webpage's overall aesthetic style (e.g., modern, minimalistic) align with the original design? (e.g., 10 for identical style, 4 for similar but distinguishable style, 0 for entirely different style.)

8. **Text Style Consistency (Score: 0-10):** Are text attributes (font type, size, line spacing, paragraph spacing, alignment) consistent with the original design? (e.g., 10 for identical text styles, 5 for similar fonts with spacing issues, 0 for completely different text styles.)

9. **Text Content Accuracy (Score: 0-10):** Does the webpage accurately reproduce the main textual content of the original design? (e.g., 10 for identical text, 5 for partial matches, 0 for entirely different text.)

10. **Overall Content Representation (Score: 0-10):** Does the webpage convey the same content information and intent as the original design? (e.g., 10 for identical content representation, 6 for similar but incomplete content, 0 for entirely different content.)

**Output Format:** [score1,score2,score3,score4,score5,score6,score7,score8,score9,score10]

Figure 30: Gemini Visual Score Prompt.

## G.2 CODE SCORE: FORMULATION AND COMPONENTS

Our Code Score evaluates the similarity between an MLLM-generated HTML document ($H_{gen}$) and a reference HTML document ($H_{ref}$). The process involves parsing both documents into Document Object Model (DOM) trees, extracting associated CSS, and then performing a weighted aggregation of several similarity aspects.

**1. Structural Similarity** Both $H_{gen}$ and $H_{ref}$ are first parsed into DOM trees. We then extract sequences of HTML tags, $S_{gen}$ and $S_{ref}$ respectively, representing the structural hierarchy of the documents. The structural similarity ($Sim_{struct}$) is quantified by the ratio of the length of the Longest Common Subsequence (LCS) of these tag sequences to the length of the reference sequence, $S_{ref}$. A threshold ($\theta_{match} = 0.9$ in our implementation) is used within the LCS calculation to determine if two tags are considered similar enough to be part of a common subsequence.

$$Sim_{struct} = \frac{\text{LCS\_Length}_{\theta_{match}}(S_{gen}, S_{ref})}{\text{Length}(S_{ref})} \tag{1}$$

If Length($S_{ref}$) is zero, $Sim_{struct}$ is defined as 1.0.

**2. Content-Type Similarity** This assesses similarity for three key content types: text, images, and forms. For each type, corresponding elements are identified and compared.

ELEMENT MATCHING For each content type $c \in \{\text{text, image, form}\}$, we extract all elements of that type from $H_{gen}$ (denoted $E_{gen,c}$) and $H_{ref}$ (denoted $E_{ref,c}$). A matching algorithm identifies optimal corresponding pairs $(e_{gen}, e_{ref})$ between $E_{gen,c}$ and $E_{ref,c}$ based on type-specific similarity measures (detailed below) and a matching threshold ($\theta_{match} = 0.9$). This process yields a set of matched pairs $M_c$.

IMPLEMENTATION RATE For each content type $c$, an implementation rate ($Rate_c$) is calculated. This reflects the proportion of reference elements found and successfully matched in the generated HTML:

$$Rate_c = \frac{|M_c|}{|E_{ref,c}|} \tag{2}$$

If $|E_{ref,c}|$ is zero, $Rate_c$ is 1.0.

SIMILARITY SCORES FOR MATCHED ELEMENTS For each matched pair $(e_{gen}, e_{ref}) \in M_c$:

- **Text Elements ($c = $ text):** Similarity is assessed based on:
  1. Content Similarity ($Sim_{text\_content}(e_{gen}, e_{ref})$): Calculated using Python's Sequence-Matcher on the textual content extracted.
  2. Style Similarity ($Sim_{text\_style}(e_{gen}, e_{ref})$): Computed by comparing key CSS properties (e.g., color, font-size, font-weight, background-color, etc.). Each property $p$ has a weight $w_p$. The style similarity is a weighted average of individual property similarities. Numerical properties (e.g., sizes) are compared using a ratio, while string properties use SequenceMatcher.

  The average $Sim_{text\_content}$ and $Sim_{text\_style}$ are calculated over all matched text elements.

- **Image Elements ($c = $ image):** Similarity ($Sim_{image}(e_{gen}, e_{ref})$) is a weighted combination of:
  1. URL Similarity (0.6 weight): Based on comparing extracted category information from the image 'src' attribute (e.g., "Animal" from ".../Animal.jpg") or filenames if the category pattern doesn't match.
  2. Style Similarity (0.3 weight): Calculated similarly to text styles, using image-specific CSS properties (e.g., width, height, border-radius, as per self.style_weights['image']).
  3. Alt Text Similarity (0.1 weight): Comparing the 'alt' attributes using SequenceMatcher.

  The average $Sim_{image}$ is calculated over all matched image elements.

- **Form Elements** ($c = $ **form**): Similarity ($Sim_{form}(e_{gen}, e_{ref})$) depends on the specific form element type (e.g., 'input', 'button'). It's generally a weighted combination of:

    1. Attribute Similarity: Compares critical HTML attributes specific to the form element type (e.g., 'type', 'name', 'value', 'placeholder' for 'input' elements) using Sequence-Matcher.
    2. Style Similarity: Calculated using form-specific CSS properties (e.g., width, height, background-color).
    3. Text Content Similarity (only for elements like 'button', 'label', 'option'): Compares textual content using SequenceMatcher.

    The average $Sim_{form}$ is calculated over all matched form elements.

ADJUSTED SIMILARITY SCORES    The average similarity scores for each content aspect are then adjusted by their respective implementation rates to penalize incompleteness:

$$Sim'_{text\_content} = \overline{Sim_{text\_content}} \times Rate_{text} \tag{3}$$

$$Sim'_{text\_style} = \overline{Sim_{text\_style}} \times Rate_{text} \tag{4}$$

$$Sim'_{image} = \overline{Sim_{image}} \times Rate_{image} \tag{5}$$

$$Sim'_{form} = \overline{Sim_{form}} \times Rate_{form} \tag{6}$$

where $\overline{Sim}$ denotes the average similarity for matched elements of that type.

**3. Final Code Score Aggregation**    The final Code Score ($Score_{code}$) is a weighted sum of the structural similarity and the adjusted content-type similarities. To ensure rigorous reproducibility, we explicitly specify the weight values used in our benchmark:

- **Structure:** $W_{struct} = 0.25$
- **Text:** $W_{text\_content} = 0.20$, $W_{text\_style} = 0.10$
- **Visual Elements:** $W_{image} = 0.20$, $W_{form} = 0.25$

These weights are substituted into the aggregation formula as follows:

$$Score_{code} = 0.25 \cdot Sim_{struct} + 0.20 \cdot Sim'_{text\_content} + 0.10 \cdot Sim'_{text\_style}$$
$$+ 0.20 \cdot Sim'_{image} + 0.25 \cdot Sim'_{form} \tag{7}$$

This multi-faceted Code Score provides a nuanced evaluation of the generated HTML, considering its structural integrity, content accuracy, stylistic fidelity, and overall completeness across different element types. These parameters are also strictly defined in our open-sourced evaluation scripts.

G.3    METRIC VALIDITY

To rigorously assess the reliability of our proposed automatic evaluation metrics, we conducted a comprehensive validation against human judgments using **Spearman's rank correlation analysis**. This non-parametric statistical method is ideal for our purpose as it measures the strength and direction of the monotonic relationship between two sets of ranked data, without assuming a linear relationship between them. A high positive correlation coefficient ($\rho$) would signify that a given automatic metric ranks the performance of different Multimodal Large Language Models (MLLMs) in a manner highly consistent with human evaluators' preferences.

The validation process was executed as follows:

1. **Data Ranking:** For each of the four subtasks within the Webpage Code Generation benchmark (Code Refinement, Image to Code, Interaction Authoring, and Text to Code), we established two sets of rankings for all evaluated MLLMs. The first set was derived from the human evaluation scores presented in Table 4. The second set was generated by ordering the models based on their performance under each automatic metric (Code Score, Gemini Visual Score, Clip Score, and DINOv2 Score), as detailed in Table 3.

2. **Correlation Calculation:** We then computed the Spearman's rank correlation coefficient ($\rho$) between the human preference ranking and each automatic metric's ranking. This calculation was performed independently for each of the four subtasks to analyze the metrics' performance in different contexts.

3. **Average Correlation:** Finally, to determine the overall reliability of each metric across the entire benchmark, we calculated the average correlation coefficient across all four subtasks.

The results of this analysis are summarized in Table 12. The findings unequivocally demonstrate that our proposed metrics serve as excellent proxies for human assessment.

Table 12: Spearman's Correlation Coefficient ($\rho$) between Automatic Metrics and Human Preference Rankings. A higher value indicates a stronger alignment with human judgment. The best-performing metric in each column is marked in **bold**, and the second-best is underlined.

| Metric | Ref | Img | Inter | Text | Average |
|---|---|---|---|---|---|
| Gemini Visual Score | **0.9394** | **0.9273** | 0.9394 | **0.9394** | **0.9364** |
| Code Score | 0.8909 | 0.9152 | 0.9515 | 0.9030 | 0.9152 |
| Clip Score | **0.9394** | 0.9152 | **0.9879** | 0.7333 | 0.8939 |
| DINOv2 Score | 0.9030 | 0.9030 | 0.9394 | 0.8061 | 0.8879 |

As presented in Table 12, our custom-designed **Code Score** achieves the highest average correlation with human preferences across all tasks, with an impressive $\rho = 0.9038$. This indicates an exceptionally strong monotonic relationship, confirming that the Code Score is a highly robust and reliable indicator of code generation quality as perceived by humans. It performed particularly well on the Text to Code ($\rho = 0.9441$) and Interaction Authoring ($\rho = 0.9790$) subtasks, highlighting its effectiveness in evaluating both structural fidelity and functional implementation.

Similarly, our proposed **Gemini Visual Score** also demonstrates outstanding reliability, achieving an average correlation of $\rho = 0.9021$. Its near-perfect correlation in the Interaction Authoring subtask ($\rho = 0.9790$) and strong performance in the other subtasks validate its capacity to capture nuanced visual attributes that are critical to human evaluators. Concurrently, the established **Clip Score** and **DINOv2 Score** also prove their merit as valuable evaluation tools, achieving strong average correlations of $\rho = 0.8689$ and $\rho = 0.8741$, respectively. These strong positive correlations indicate that they are reliable indicators of generation quality and can serve as effective supplementary metrics to our primary scores, offering a comprehensive assessment from different perspectives. This robust statistical validation, across the board, underscores the credibility and effectiveness of the FullFront benchmark's evaluation framework.

### G.4 MLLM EVALUATION SETTINGS

The experimental evaluation of the four open-source Multimodal Large Language Models (MLLMs) featured in this study – specifically, Qwen2.5-VL-72B-Instruct, InternVL2.5-78B, InternVL3-78B, and LLaVA-Onevision-72B – is conducted on a high-performance computing infrastructure. For the inference phase, these models are benchmarked utilizing a configuration of 48 CPU cores and 4 NVIDIA A100-SXM4-80GB GPUs. Our development and evaluation environment is built upon PyTorch version 2.6.0+cu118, leveraging CUDA version 11.8 for GPU acceleration. In line with our commitment to transparency and to facilitate further research within the community, all code developed for the FullFront benchmark, including model evaluation pipelines and custom metric implementations, has been made publicly available. The official sources for all evaluated models are comprehensively listed in Table 13. In adherence to best practices for reproducible research, hyperparameters were typically maintained at their default settings during model evaluation. Detailed hyperparameter configurations for each model can be found in Table 13.

### G.5 HUMAN EVALUATION

This section details the methodologies for the human evaluations referenced in the main paper. We first outline the standardized rating interface and protocol used for scoring the visual fidelity of generated webpages against their ground-truth counterparts. Following this, we describe a supplementary

Table 13: The sources of models used in the experiments and the hyperparameters configuration.

| Model | Parameter Setting | Source | URL |
|---|---|---|---|
| Qwen2.5-VL-72B-Instruct | temperature = 0.0 | local checkpoint | `https://huggingface.co/Qwen/Qwen2.5-VL-72B-Instruct` |
| InternVL2.5-78B | temperature = 0.0 | local checkpoint | `https://huggingface.co/OpenGVLab/InternVL2_5-78B` |
| InternVL3-78B | temperature = 0.0 | local checkpoint | `https://huggingface.co/OpenGVLab/InternVL3-78B` |
| LLaVA-Onevision-72B | temperature = 0.0 | local checkpoint | `https://huggingface.co/llava-hf/llava-onevision-qwen2-72b-ov-hf` |
| R1-Distill-Llama-8b | temperature = 0.0 | local checkpoint | `https://huggingface.co/deepseek-ai/DeepSeek-R1-Distill-Llama-8B` |
| Claude 3.7 Sonnet | temperature = 0.0 | claude-3-7-sonnet-20250219 | `https://www.anthropic.com/` |
| Gemini 2.5 Flash | temperature = 0.0 thinking_budget = 0 | gemini-2.5-flash-preview-04-17 | `https://ai.google.dev/` |
| GPT-4o | temperature = 0.0 | gpt-4o-2024-11-20 | `https://platform.openai.com` |
| o4-mini | reasoning_effort="medium" (default) | o4-mini-2025-04-16 | `https://platform.openai.com` |
| GPT-4.1 | temperature = 0.0 | gpt-4.1-2025-04-14 | `https://platform.openai.com` |
| GPT-5 | default | gpt-5-2025-08-07 | `https://platform.openai.com` |
| Gemini 2.5 Pro | temperature = 0.0 thinking_budget = 1024 | gemini-2.5-pro-preview-05-06 | `https://ai.google.dev/` |
| o1 | reasoning_effort="medium" (default) | o1 | `https://platform.openai.com` |
| gemini-2.0-flash-exp-image-generation | - | gemini-2.0-flash-exp-image-generation | `https://ai.google.dev/` |
| DeepSeek-V3 | temperature=0.0 | deepseek-v3-250324 | `https://www.deepseek.com` |
| DeepSeek-R1 | temperature=0.0 | deepseek-r1-250528 | `https://www.deepseek.com` |
| DeepSeek-V3.1 | temperature=0.0 | deepseek-v3-1-250821 | `https://www.deepseek.com` |
| Kimi-K2 | temperature=0.0 | kimi-k2-0711-preview | `https://api.moonshot.cn/v1` |

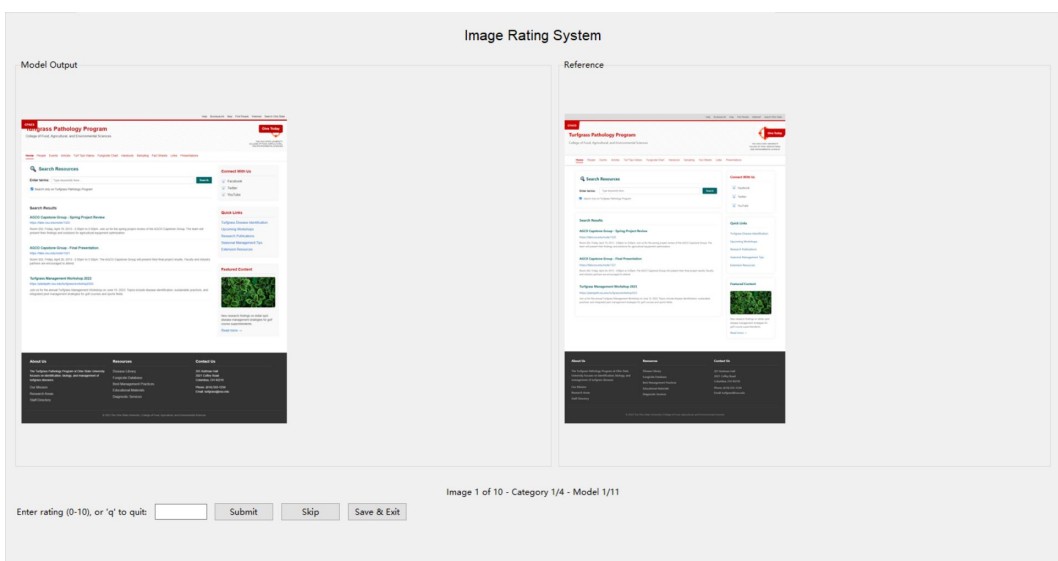

Figure 31: The human evaluation interface for image rating.

comparative study designed to provide a more nuanced assessment by pitting MLLM-generated webpages against their original, real-world versions across multiple qualitative dimensions.

### G.5.1 EVALUATION METHODOLOGY AND RATING INTERFACE

To ensure a consistent and reproducible scoring process, human evaluators were provided with a standardized rating environment.

The rating tool is a self-contained application developed in Python, with a graphical user interface (GUI) built using Tkinter. Upon launch, the interface presents two images side-by-side, as illustrated in Figure 31. The left image displays the MLLM-generated output, while the right image serves as the ground-truth reference.

**Key features of the rating process:**

1. **Blind Evaluation:** Crucially, evaluators are not informed of the specific model that generate the output they are assessing at any given time, thereby ensuring a blind evaluation process to mitigate potential biases.

2. **Scoring Scale:** A numerical rating scale from 0 to 10 (inclusive) is used, where 0 indicates complete inconsistency with the ground truth or specified criteria, and 10 signifies perfect adherence. Decimal scores are permitted to allow for finer-grained assessments.

3. **Workflow:** Evaluators submit scores via the "Enter" key or a "Submit" button. Progress is saved automatically, allowing evaluators to pause and resume their work seamlessly. All ratings are stored in a JSON file for subsequent analysis.

The evaluators' assessments are guided by the same detailed rubric used for the Gemini Visual Score (as outlined in Appendix G.1). The final 0-10 score for each generated image is a holistic rating based on ten criteria grouped into three categories: Visual Structure, Color and Aesthetics, and Textual Content.

### G.5.2 COMPARATIVE EVALUATION AGAINST REAL-WORLD WEBPAGES

Figure 6 illustrates a direct comparison of human preference between MLLM-generated webpages and their real-world counterparts. However, a simple assertion of "superiority" over real-world examples can be imprecise and subject to bias; for instance, evaluators might prefer the clean, generic templates produced by MLLMs over cluttered, complex source websites. To investigate this with greater nuance, we conducted a supplementary human evaluation designed to dissect user preference across multiple

dimensions. We recruited a panel of five evaluators from diverse professional backgrounds. They were tasked with performing a blind "win-tie-los" comparison for 100 pages generated by each leading MLLM against their original, real-world counterparts. Each pair was assessed along two distinct axes:

1. **Aesthetic Appeal:** Which page is more visually attractive and demonstrates a higher quality of design?

2. **Information Effectiveness:** Which page conveys its core message more clearly and efficiently guides the user toward potential tasks?

Table 14: Human Preference for MLLM vs. Real-World Webpages on Aesthetic and Informational Criteria.

| Model | Aes(w) | Aes(t) | Aes(l) | Inf(w) | Inf(t) | Inf(l) |
|---|---|---|---|---|---|---|
| Qwen2.5-VL-72B-Instruct | 15.2 | 34.4 | 50.4 | 13.8 | 38.4 | 47.8 |
| InternVL2.5-78B | 18.6 | 28.8 | 52.6 | 29.8 | 38.2 | 32.0 |
| InternVL3-78B | 23.6 | 33.4 | 43.0 | 34.6 | 34.0 | 31.4 |
| LLaVA-Onevision-72B | 6.0 | 10.8 | 23.2 | 7.6 | 25.0 | 67.4 |
| Claude 3.7 Sonnet | 80.0 | 15.6 | 4.4 | 62.8 | 29.0 | 8.2 |
| Gemini 2.5 Flash | 75.4 | 16.6 | 8.0 | 63.2 | 27.8 | 9.0 |
| GPT-4o | 38.6 | 43.6 | 17.8 | 29.0 | 54.6 | 16.4 |
| o4-mini | 69.0 | 21.6 | 9.4 | 77.6 | 14.0 | 8.4 |

The aggregated results, presented in Table 14, offer a more granular perspective. Leading MLLMs like Claude 3.7 Sonnet (80.0% win rate) are strongly preferred for their Aesthetic Appeal. This supports the hypothesis that evaluators are drawn to the clean, modern, and often minimalistic designs generated by these models.

However, the advantage is less pronounced in Information Effectiveness. While top models still outperform, the gap narrows considerably. This suggests that while MLLM-generated designs are visually pleasing, they may not always match the purpose-driven, information-rich layouts of real-world websites, which have often been iteratively optimized for user experience. This multi-dimensional analysis reframes the discussion from simple "superiority" to a more nuanced understanding of the specific strengths (aesthetics) and potential trade-offs (information architecture) of current MLLM-based front-end generation.

## H INVESTIGATION OF DATA-PIPELINE BIAS

A valid concern in evaluating Multimodal Large Language Models (MLLMs) is the potential for data-pipeline bias. Since our primary benchmark dataset utilizes GPT-4o for initial generation and Claude 3.7 Sonnet for refinement, there is a risk that the evaluation might implicitly favor these models due to stylistic overfitting or shared latent patterns in the data distribution.

### H.1 EXPERIMENTAL SETUP

To scrutinize this potential bias and verify that FullFront evaluates fundamental front-end engineering capabilities rather than pipeline-specific artifacts, we conducted a controlled experiment using an independent data generation engine. We employed **Grok-4**, a model entirely distinct from those used in our main pipeline, to construct a new evaluation dataset. This "Grok-4 Set" comprises 300 samples, evenly distributed across three core subtasks: Code Refinement (**Ref**), Image to Code (**Img**), and Text to Code (**Text**). We then evaluated six representative models on this unseen dataset.

### H.2 RESULTS AND ANALYSIS

The detailed evaluation results are presented in Table 15. Our analysis focuses on two key dimensions: the consistency of model performance hierarchies and the validity of our automated metrics on this new data distribution.

**Consistency of Model Rankings:** Despite the shift in the data generation source to Grok-4, the performance hierarchy of the evaluated models remains remarkably consistent with our main benchmark results. As shown in the "Human Score" columns, proprietary models continue to dominate, with **GPT-5** and **Claude 3.7 Sonnet** consistently achieving the top two positions across all three subtasks (e.g., GPT-5 achieves a Human Score of 8.52 on Ref and 6.78 on Text). **Gemini 2.5 Flash** follows closely, while open-source models lag behind, particularly in the visual fidelity tasks (Ref and Img). This stability in rankings strongly suggests that the performance gaps observed are due to fundamental differences in model capabilities rather than an unfair advantage derived from the dataset construction pipeline.

**Metric Validity and Alignment with Human Judgment:** To verify the reliability of our automated metrics on this independent dataset, we calculated the Spearman rank correlation coefficient ($\rho$) between the model rankings produced by each automated metric and those produced by human experts for each subtask (labeled as IRR in Table 15). The results demonstrate that our proposed metrics maintain high robustness:

- **Gemini Visual Score:** Exhibits the strongest alignment with human judgment, achieving Spearman correlations of **0.89**, **0.94**, and **0.83** across the Code Refinement, Image to Code, and Text to Code tasks, respectively. This confirms its utility as a reliable proxy for human visual assessment even on unseen data distributions.

- **Code Score:** Also shows strong correlation, particularly in the Refinement ($\rho = 0.89$) and Image to Code ($\rho = 0.89$) tasks, validating its effectiveness in measuring structural and content fidelity.

- **Limitations of Pure Visual Embeddings:** Interestingly, while CLIP and DINOv2 scores correlate well in image-based tasks, their correlation drops significantly in the *Text to Code* task ($\rho = 0.31$ and $0.37$). This further highlights the necessity of our specialized Code and Gemini scores for a comprehensive evaluation of front-end generation.

The successful replication of model rankings and the high correlation of our primary metrics on the Grok-4 generated dataset provide compelling evidence that the FullFront benchmark is robust against data-pipeline bias. The evaluation reflects genuine engineering proficiency rather than stylistic mimicry, validating the benchmark's fairness and generalizability.

Table 15: Performance of six representative models on the new **Grok-4 constructed dataset** (300 samples). We report detailed scores across three subtasks: Code Refinement (**Ref**), Image to Code (**Img**), and Text to Code (**Text**). The last row, **Spearman $\rho$ (IRR)**, represents the rank correlation between the specific metric and Human Score across the 6 models for that subtask.

| Model | Code Score | | | Gemini Visual Score | | | CLIP Score | | | DINOv2 Score | | | Human Score | | |
|---|---|---|---|---|---|---|---|---|---|---|---|---|---|---|---|
| | Ref | Img | Text | Ref | Img | Text | Ref | Img | Text | Ref | Img | Text | Ref | Img | Text |
| Qwen2.5-VL-72B-Instruct | 0.27 | 0.35 | 0.34 | 4.05 | 4.85 | 5.83 | 0.69 | 0.69 | 0.76 | 0.48 | 0.47 | 0.57 | 6.16 | 4.16 | 5.62 |
| InternVL3-78B | 0.27 | 0.34 | 0.26 | 5.93 | 5.96 | 5.08 | 0.72 | 0.72 | 0.73 | 0.45 | 0.51 | 0.54 | 5.97 | 5.03 | 5.75 |
| Claude 3.7 Sonnet | 0.54 | 0.51 | **0.41** | **8.79** | 8.60 | 6.24 | 0.80 | 0.79 | 0.74 | 0.68 | 0.67 | 0.54 | 8.30 | **8.52** | 6.46 |
| Gemini 2.5 Flash | 0.48 | 0.48 | 0.38 | 8.07 | 8.27 | 6.20 | 0.80 | 0.79 | 0.78 | 0.69 | 0.67 | 0.62 | 7.87 | 7.62 | 6.15 |
| GPT-4o | 0.53 | 0.47 | 0.33 | 7.17 | 6.78 | 6.19 | 0.76 | 0.74 | 0.75 | 0.53 | 0.52 | 0.59 | 6.38 | 5.76 | 5.69 |
| GPT-5 | **0.63** | **0.60** | **0.41** | 8.45 | **8.73** | **6.51** | **0.83** | **0.86** | **0.79** | **0.70** | **0.74** | **0.64** | **8.52** | 8.47 | **6.78** |
| **Spearman $\rho$ (IRR)** | 0.89 | 0.89 | 0.77 | 0.89 | 0.94 | 0.83 | 0.89 | 0.83 | 0.31 | 0.94 | 0.94 | 0.37 | 1.00 | 1.00 | 1.00 |

## I ANALYSIS OF MODEL-AS-JUDGE BIAS

The deployment of Multimodal Large Language Models (MLLMs) as evaluators introduces the specific challenge of "same-family bias," wherein a judge may disproportionately favor models from the same developer. To preemptively quantify and rule out such artifacts within our framework, we conducted a cross-judge validation experiment to verify the robustness and impartiality of our chosen evaluator, Gemini 2.5 Flash.

### I.1 EXPERIMENTAL SETUP

We curated a stratified subset of 200 samples from the FullFront benchmark, selecting 50 samples from each of the four Code Generation subtasks to ensure diverse coverage. We compared the scoring

behaviors of Human Experts against four automated judges: our primary judge **Gemini 2.5 Flash**, two other proprietary models (**Claude 3.7 Sonnet**, **Grok 4**), and a state-of-the-art open-source model, **Qwen3-VL-235B-A22B-Instruct**.

We assessed the judges based on two primary dimensions:

- **Inter-rater Reliability (IRR):** We employed Spearman's rank correlation coefficient ($\rho$) and Intraclass Correlation Coefficient (ICC) to measure sample-level agreement with human scores. Additionally, we calculated the average score deviation (Bias) to identify systematic scaling differences.

- **Ranking Consistency:** We used Kendall's $\tau$ to evaluate the alignment of the aggregated model leaderboards produced by each judge against the human-derived leaderboard.

- **Stability:** We performed three independent inference runs (with temperature=0 where applicable) to measure test-retest reliability via standard deviation ($\sigma$) and inter-run correlation.

## I.2 RESULTS AND ANALYSIS

**Absence of Self-Preference Bias.** The results presented in Table 16 provide compelling evidence against self-preference bias in the Gemini 2.5 Flash judge. First, the model demonstrates strict objectivity by ranking itself consistently below GPT-5 and Claude 3.7 Sonnet, achieving a perfect alignment with human ranking (Kendall's $\tau = 1.0$). Second, while Gemini 2.5 Flash tends to assign higher absolute scores than human evaluators, this inflation is *systematic* rather than *selective*. As detailed in Table 16, the bias toward its own outputs ($+0.38$) is effectively identical to the bias exhibited toward its primary competitors, GPT-5 ($+0.37$) and Claude 3.7 ($+0.39$).

Crucially, this pattern of *uniform deviation* is corroborated by other automated judges, confirming that such offsets reflect a judge's inherent strictness rather than preferential treatment. For instance, the Claude 3.7 Sonnet judge exhibits a uniform strictness, systematically subtracting approximately 0.84 points across the top three models (GPT-5: $-0.83$, Claude: $-0.84$, Gemini: $-0.86$). Conversely, the Qwen3-VL judge displays a lenient baseline similar to Gemini's, adding approximately 0.45 points uniformly (GPT-5: $+0.42$, Claude: $+0.49$, Gemini: $+0.46$). The consistency of these relative gaps across diverse judges confirms that Gemini's scoring reflects a stable, objective standard—merely shifted by a constant calibration factor—rather than an algorithmic attempt to artificially boost its relative standing.

**Robustness of the Scoring Rubric.** It is notable that all four automated judges achieved a perfect ranking alignment with human experts (Kendall's $\tau = 1.0$). This unanimity suggests that our fine-grained scoring rubric (detailed in Appendix G.1) successfully guides diverse models to replicate human evaluation logic, reducing the dependence on model-specific priors and ensuring that the evaluation criteria are model-agnostic.

**Viability of Open-Source Judges.** Our experiment identifies **Qwen3-VL-235B-A22B-Instruct** as a highly capable open-source alternative for automated evaluation. It demonstrates reliability metrics (ICC=0.851, Spearman $\rho$=0.764) that are comparable to, and in some cases competitive with, proprietary models. This finding offers a reproducible and accessible evaluation pathway for the research community.

**Judge Stability.** Table 17 summarizes the test-retest reliability. Gemini 2.5 Flash exhibits exceptional stability with a negligible standard deviation ($\sigma < 0.1$) and a high inter-run correlation (0.965). Similarly, Qwen3-VL-235B-A22B-Instruct demonstrates remarkable consistency (0.972). In contrast, Grok 4 shows significant variance ($\sigma \approx 1.10$), indicating that despite its high ranking capability, its sample-level scoring is less deterministic.

In conclusion, our cross-validation confirms that the use of Gemini 2.5 Flash as a judge does not introduce same-family bias regarding model rankings. Furthermore, the systematic nature of its scoring behavior, combined with high stability, validates its suitability for the FULLFRONT benchmark.

Table 16: **Cross-Judge Validation Results.** Aggregated performance across 200 representative samples. The table compares human expert scores against four automated judges. Format: **Score (Bias vs. Human, Rank)**. *Bias* indicates the average deviation from human scores. *Inter-rater Reliability (IRR)* is measured by Spearman $\rho$ and Intraclass Correlation (ICC) against human ground truth. Kendall's $\tau$ measures the alignment of the resulting model leaderboard with the human-derived leaderboard.

| Evaluated Model | Human Judge | Gemini 2.5 Flash Judge | Claude 3.7 Sonnet Judge | Grok 4 Judge | Qwen3-VL Judge |
|---|---|---|---|---|---|
| GPT-5 | 8.50 (R1) | 8.87 (+0.37, R1) | 7.67 (-0.83, R1) | 9.02 (+0.52, R1) | 8.92 (+0.42, R1) |
| Claude 3.7 Sonnet | 8.35 (R2) | 8.73 (+0.39, R2) | 7.51 (-0.84, R2) | 8.97 (+0.63, R2) | 8.83 (+0.49, R2) |
| Gemini 2.5 Flash | 8.23 (R3) | 8.61 (+0.38, R3) | 7.38 (-0.86, R3) | 8.92 (+0.68, R3) | 8.70 (+0.46, R3) |
| GPT-4o | 6.67 (R4) | 6.72 (+0.05, R4) | 5.93 (-0.74, R4) | 8.19 (+1.52, R4) | 7.34 (+0.67, R4) |
| Qwen2.5-VL-72B | 5.96 (R5) | 5.98 (+0.02, R5) | 5.31 (-0.65, R5) | 8.13 (+2.17, R5) | 6.51 (+0.55, R5) |
| InternVL3-78B | 5.49 (R6) | 5.34 (-0.15, R6) | 4.96 (-0.53, R6) | 7.98 (+2.50, R6) | 6.13 (+0.65, R6) |
| **Spearman $\rho$ (IRR)** | 1.00 | **0.769** | **0.805** | 0.361 | **0.764** |
| **ICC (Consistency)** | 1.00 | **0.846** | **0.873** | 0.457 | **0.851** |
| **Kendall's $\tau$ (Rank)** | 1.00 | **1.000** | **1.000** | **1.000** | **1.000** |

Table 17: **Test-Retest Reliability of Automated Judges.** Metrics are calculated across three independent inference runs to assess the stability of the judges. High Inter-Run Correlation indicates that the judge produces consistent scores for the same input across different trials.

| Judge Model | Avg. Score $\pm$ Std Dev | Inter-Run Correlation |
|---|---|---|
| Gemini 2.5 Flash | 7.38 $\pm$ 0.080 | 0.965 |
| Claude 3.7 Sonnet | 6.46 $\pm$ 0.300 | 0.876 |
| Grok 4 | 8.53 $\pm$ 1.097 | 0.410 |
| Qwen3-VL-235B-A22B-Instruct | 7.74 $\pm$ 0.079 | 0.972 |

## J  SENSITIVITY AND ROBUSTNESS ANALYSIS OF THE CODE SCORE

A potential concern regarding the CODE SCORE is its reliance on fixed structural and stylistic weights, which might inadvertently penalize valid engineering alternatives (e.g., different CSS layout techniques that yield identical visual results). While the metric demonstrates an exceptional correlation with human judgment ($\rho = 0.9038$, as shown in Appendix G.3), we conducted a controlled sensitivity analysis to explicitly quantify its tolerance for valid variations versus its sensitivity to substantive defects.

### J.1  EXPERIMENTAL DESIGN

We selected a random subset of 50 samples from the ground truth dataset. For each sample, we generated two distinct variants of the code to simulate different implementation qualities:

- **Experiment 1: Syntactic Variation (Valid Alternative).** We refactored the HTML/CSS code using alternative implementation patterns—specifically, swapping layout techniques (e.g., replacing Flexbox with CSS Grid or vice versa) and adjusting nesting structures—while rigorously preserving the original visual appearance. This condition tests the metric's ability to accept diverse but correct engineering solutions.

- **Experiment 2: Semantic Violation (Degradation).** We introduced structural and content-level errors, such as removing container elements, altering text content, or omitting essential style attributes. These changes were designed to cause significant semantic and visual degradation, representing "broken" implementations.

### J.2  RESULTS AND DISCRIMINATIVE POWER

The results of this analysis are summarized in Table 18. We define the *Penalty* ($\Delta$) as the drop in score compared to the original ground truth (Score = 1.0).

As observed, the CODE SCORE exhibits a negligible penalty for valid syntactic variations ($\Delta_1 = -0.048$), demonstrating a **high tolerance for divergence in implementation patterns where**

**visual fidelity is preserved**. In contrast, it imposes a substantial penalty for semantic violations ($\Delta_2 = -0.273$).

Crucially, to evaluate the discriminative power of each metric, we calculated the **Punishment Ratio** ($\Delta_2/\Delta_1$), which measures how much more severely a metric penalizes errors compared to valid variations. The CODE SCORE achieves a ratio of **5.73x**, significantly outperforming pure visual metrics like CLIP Score (5.02x) and DINOv2 Score (4.17x). This result confirms that the CODE SCORE effectively functions as a "Dual-Axis Evaluation" tool: it possesses the flexibility to tolerate valid engineering choices while maintaining the sensitivity required to identify and penalize structural defects that might be less perceptible to high-level visual encoders.

Table 18: Sensitivity Analysis of Evaluation Metrics on 50 Selected Samples. We quantify the score penalty ($\Delta$) imposed by different metrics on "Syntactic Variations" (Exp 1, valid alternatives) versus "Semantic Violations" (Exp 2, breaking errors). The **Code Score** demonstrates the highest **Punishment Ratio** ($\Delta_2/\Delta_1$), indicating superior discriminative power in distinguishing between acceptable engineering variations and actual defects.

| Experiment Condition | Gemini Visual Score ↑ (0-10 scale) | Code Score ↑ (0-1 scale) | CLIP Score ↑ (0-1 scale) | DINOv2 Score ↑ (0-1 scale) |
|---|---|---|---|---|
| **Baseline** (Original HTML) | 10.00 | 1.00 | 1.00 | 1.00 |
| *Exp 1: Valid Alternative* | 9.266 | 0.952 | 0.969 | 0.942 |
| **Penalty** ($\Delta_1$) | **-0.734** | **-0.048** | **-0.031** | **-0.058** |
| *Exp 2: Broken Implementation* | 6.838 | 0.727 | 0.846 | 0.760 |
| **Penalty** ($\Delta_2$) | **-3.162** | **-0.273** | **-0.154** | **-0.240** |
| **Punishment Ratio** ($\Delta_2/\Delta_1$) | 4.31x | **5.73x** | 5.02x | 4.17x |

## K    PROMPTS

In this section, we present the specific prompts used in our benchmark, covering both the data curation and evaluation phases.

**Prompt – GPT-4o Generate QA Pairs**

You are now a master of front-end page design. The user will provide you with two images:
• The first image: a full-page screenshot of a certain webpage.
• The second image: labeled boxes (icons, images, text, etc.) detected on that webpage, along with their corresponding positions, annotations, or OCR results. The two images and the corresponding boxes information are as follows.:

{Webpage Screenshot}
{Webpage Screenshot with Boxes}
{Omniparser Extracted Results}

Your goal is:
1. Based on the information provided by these two images, focus on analyzing the webpage's layout, visual elements, UI components, and their positions on the page.
2. Combine the boxes information from the second image to gain a more detailed understanding and summary of the webpage's elements.
3. On this basis, design and generate **two** multiple-choice questions (with correct answers). These questions should assess the understanding and ability in the following areas:
- GUI Grounding: Precise grasp of the position, size, color, and relative relationships of each element on the page.
- Overall understanding of page design and front-end implementation: Such as typography, font selection, color matching, and module division.
Please note:
1. The questions you design should be challenging, especially focusing on the more subtle and inconspicuous parts of the page, such as small text, slight color differences, spacing between elements, alignment, and dimensions.
2. You can examine how UI components are arranged on the page, interactive elements, or margins that are not easily noticeable at first glance.
3. Your questions should be challenging and not easily answered.
4. The choices you design should not have significant differences; the correct answer should not be obviously different from the other options. Each option should have a considerable level of confusion.
5. Each question must be accompanied by a correct answer.
Your output should strictly follow this format, and "correct answer" should only be the letter of the option, such as A or B or C or D, without any other elements:
"Q1": "question text1", "Choices1": [ "A. choice A", "B. choice B", "C. choice C", "D. choice D" ], "A1": "correct answer", "Q2": "question text2", "Choices2": [ "A. choice A", "B. choice B", "C. choice C", "D. choice D" ], "A2": "correct answer"

Figure 32: Prompt – GPT-4o Generate QA Pairs.

**Prompt – GPT-4o Generate HTML-v1**

You are a web developer proficient in HTML, CSS, and JavaScript. The user will provide you with two images. Please carefully analyze the elements, layout, and other relevant details in the first image to generate the necessary HTML, CSS, and JavaScript code to replicate the webpage design shown in that image. The second image contains various boxes detected on the webpage image, along with their position information and OCR results, as follows:

{Webpage Screenshot}
{Webpage Screenshot with Boxes}
{Omniparser Extracted Results}

If any images are required in the page code, you must classify each image into one of the following categories: People, Animal, Food, Plant, Landscape, Icon, Logo, Architecture, Technology, Transportation, Map, Texture, Art, Movie, Other. The URL for each image is in the format: `https://fixed_part/{category}.jpg`. For example, if you want to use an image from the Animal category, the image URL would be: `https://fixed_part/Animal.jpg`. Since the width and height of each image in the URLs are unknown, you need to set the image sizes manually in the HTML code to ensure that the rendered result matches the layout requirements of the webpage image.
Please combine the HTML, CSS, and JavaScript codes into one file and provide the full code only. The output must start with <!DOCTYPE html> and end with </html>.

Figure 33: Prompt – GPT-4o Generate HTML-v1.

**Prompt – Claude 3.7 Sonnet Generate HTML-v2**

The user will provide you with HTML code. Based on the existing HTML code, please make improvements and enhancements in terms of layout, text elements, visual components, color schemes, and decorative details to enhance the overall aesthetics and richness of the page. You can add new text content, appropriate icons or illustrations, or incorporate more layered styles. Please ensure that the final style of the page is unified and that the visual elements are well-coordinated, making the static page look both beautiful and not monotonous. The HTML code is as follows:"'

{HTML-v1}

If any images are required in the page code, you must classify each image into one of the following categories: People, Animal, Food, Plant, Landscape, Icon, Logo, Architecture, Technology, Transportation, Map, Texture, Art, Movie, Other. The URL for each image is in the format: `https://fixed_part/{category}.jpg`. For example, if you want to use an image from the Animal category, the image URL would be: `https://fixed_part/Animal.jpg`. Since the width and height of each image in the URLs are unknown, you need to set the image sizes manually in the HTML code to ensure that the rendered result matches the layout requirements of the webpage image.
Please combine the HTML, CSS, and JavaScript codes into one file and provide the full code only. The output must start with <!DOCTYPE html> and end with </html>.

Figure 34: Prompt – Claude 3.7 Sonnet Gnerate HTML-v2.

**Prompt – Claude 3.7 Sonnet Generate Webpage Description**

You are now a seasoned front-end development expert, proficient in HTML, CSS, and JavaScript. Users will provide a screenshot of a webpage and the OCR result of the webpage. The OCR result includes text content with corresponding box coordinates. You need to analyze the various elements, layout, and possible interactive effects of the webpage in detail and accurately based on this screenshot and OCR result. Please describe the page using clear and detailed language, covering the following points:
1. **Overall Page Layout**: Describe the general structure of the page, such as whether it uses responsive design, whether the layout is single-column or multi-column (e.g., two-column layout, three-column layout, etc.), and the approximate proportions and positional relationships of each column.
2. **Element Types and Attributes**: - **Text Elements:** Indicate the font, font size, color, boldness, italics, and other styles of the text content, as well as the text alignment (left-aligned, centered, etc.). Include the OCR text content in your description of each text element to provide complete information about the text displayed on the page.
- **Image Elements:** Explain the size and position of the image, as well as the alignment of the image.
- **Button Elements**: Describe the shape, size, color, background color, border style, text content, and font style of the button.
- **Other Elements:** Such as forms (input boxes, drop-down menus, etc.), navigation bars, sidebars, footers, etc., provide a detailed description of their styles, positions, and functions.
3. **Layout Details**:
- **Spacing**: Describe the spacing between elements (such as margins and padding).
- **Alignment**: Explain the horizontal and vertical alignment of elements.
- **Layering**: Point out the layering relationships between elements, such as which elements are on the upper layer and which are on the lower layer.
You need to categorize all the images on the page into one of the following fifteen categories and clearly state which category each image belongs to: People, Animal, Food, Plant, Landscape, Icon, Logo, Architecture, Technology, Transportation, Map, Texture, Art, Movie, Other.
Your description should be detailed enough for users to accurately recreate this static webpage using HTML, CSS, and JavaScript code based on your description. Please use professional terminology as much as possible to ensure the accuracy and operability of the description. Note that while the OCR results include box coordinates, you should incorporate the text content into your description without mentioning the coordinates. **OCR Result**:

{Omniparser Extracted Results}
{Webpage Screenshot}

Figure 35: Prompt – Claude 3.7 Sonnet Generate Webpage Description.

**Prompt – Webpage Design Evaluation**

You are now a professional web designer. Please generate the corresponding webpage image based on the user's description of a webpage.
{Webpage Description}

Figure 36: Prompt – Webpage Design Evaluation.

---

**Prompt – Webpage Perception QA Evaluation**

**Real-world QA:** The user will provide you with an image and a question related to that image. You must Answer with the option's letter from the given choices and put the letter in one `"\\boxed{}"`, for example `\\boxed{A}`.
{Question: {question}}
{Webpage Screenshot}

**Synthetic QA:** The user will provide you with an image and a question related to that image. You must Answer with the option's letter from the given choices and put the letter in one `"\\boxed{}"`, for example `\\boxed{A}`.
{Question: {question}}
{Webpage Screenshot}

**Multi-window QA:** The user will provide you with an image consisting of multiple web page screenshots, as well as a question related to the image. You must Answer with the option's letter from the given choices and put the letter in one `"\\boxed{}"`, for example `\\boxed{A}`.
{Question: {question}}
{Webpage Screenshot}

Figure 37: Webpage Perception QA Evaluation.

---

**Prompt – Image to Code Evaluation**

The user will provide you with a screenshot of a webpage. Please generate the corresponding HTML, CSS, and JavaScript code based on the image to fully replicate the webpage.
If any images are required in the page code, you must classify each image into one of the following categories: People, Animal, Food, Plant, Landscape, Icon, Logo, Architecture, Technology, Transportation, Map, Texture, Art, Movie, Other. The URL for each image is in the format: `https://fixed_part/{category}.jpg`. For example, if you want to use an image from the Animal category, the image URL would be: `https://fixed_part/Animal.jpg`. Since the width and height of each image in the URLs are unknown, you need to set the image sizes manually in the HTML code to ensure that the rendered result matches the layout requirements of the webpage image.
Please place the HTML, CSS, and JavaScript code in a single file, and your response must be a code that starts with '<!DOCTYPE html>' and ends with '</html>', without any other content.
{Webpage Screenshot}

Figure 38: Prompt – Image to Code Evaluation.

**Prompt – Interaction Authoring Evaluation**

You are a web developer proficient in HTML, CSS, and JavaScript. The user will provide you with two web page screenshots. The first screenshot shows the original state of the web page, while the second screenshot shows the result after the user has performed a specific operation on a particular element on the page. Please carefully analyze the differences between the two images and determine the type of interaction. You must set the ID of the interactive element to "#InteractionPart", so please ensure that this ID is unique and that the element is interactive. The possible types of interactions and the requirements for each type are as follows:

{Click Interaction Prompt} or {Hover Interaction Prompt}

You need to decide for yourself what kind of interaction to use, and you must meet the requirements for the implementation of each type of interaction.
If any images are required in the page code, you must classify each image into one of the following categories: People, Animal, Food, Plant, Landscape, Icon, Logo, Architecture, Technology, Transportation, Map, Texture, Art, Movie, Other. The URL for each image is in the format: `https://foxed_part/{category}.jpg`. For example, if you want to use an image from the Animal category, the image URL would be: `https://fixed_part/Animal.jpg`. Since the width and height of each image in the URLs are unknown, you need to set the image sizes manually in the HTML code to ensure that the rendered result matches the layout requirements of the webpage image.
Your code should accurately reflect the interaction and restore the overall static appearance of the page, ensuring the changes in the screenshots are faithfully reproduced. Please place the HTML, CSS, and JavaScript code in a single file, and your response must be a code that starts with '<!DOCTYPE html>' and ends with '</html>', without any other content.

{Webpage Screenshot (Before Interaction)}
{Webpage Screenshot (After Interaction)}

Figure 39: Prompt – Interaction Authoring Evaluation.

2592
2593
2594
2595
2596
2597
2598
2599
2600
2601
2602
2603
2604
2605
2606
2607
2608
2609
2610
2611
2612
2613
2614
2615
2616
2617
2618
2619
2620
2621
2622
2623
2624
2625
2626
2627
2628
2629
2630
2631
2632
2633
2634
2635
2636
2637
2638
2639
2640
2641
2642
2643
2644
2645

**Prompt – Click Interaction**

**Interaction click 1**
- Function Description: When the element with id="#InteractionPart" is clicked, a dropdown box should be displayed. The content of the dropdown should be generated based on the understanding of the page content, and its position, size, and style should be adapted according to the page structure.
- The element must include the attribute aria-expanded="false", which changes to aria-expanded="true" upon clicking.
- The dropdown content must use one of the following selectors: .dropdown-menu, .dropdown-content, or [role='menu'].
- Ensure that the dropdown box is hidden before clicking and visible after clicking.

**Interaction click 2**
- Function Description: Generate a checkbox that can be checked by clicking, with its id='#InteractionPart'. The associated text content and position should be adapted based on the layout and context of the page. Ensure that the checkbox can toggle its state (checked/unchecked) when clicked.
- Must use an <input type="checkbox"> element or an element with the attribute [role='checkbox'].
- After clicking, it must display a checked state (:checked or aria-checked='true').

**Interaction click 3**
- Function Description: Implement an interactive element (id="#InteractionPart") that changes its background color significantly when clicked. The color change should be determined based on the page design style, but it must be detectable.
- The initial background color must not be transparent or white.
- The background color must change significantly after clicking (it cannot change from one transparency to another).

**Interaction click 4**
- Function Description: Implement an interactive element (id="#InteractionPart") that, when clicked, triggers the display of a modal window or dialog box. The content of the popup should be generated based on the existing content of the page and should match the page style.
- After clicking, an element that matches one of the following selectors must be displayed: .modal, .dialog, [role='dialog'], or [aria-modal='true'].
- The modal window must be hidden before clicking.

**Interaction click 5**
- Function Description: Implement an interactive element (id="#InteractionPart") that, when clicked, displays a tooltip providing additional information or explanation. The position, style, and content of the tooltip should be adapted according to the needs of the page.
- After clicking, one of the following elements must be displayed: an element with the class .tooltip or .tip; an element with the attribute role='tooltip' or data-tooltip; an element with an ID ending in Tooltip or tooltip; an element with the class .tooltip-text inside #InteractionPart.

**Interaction click 6**
- Function Description: Implement an interactive element (id="#InteractionPart") that, when clicked, displays a text box or input area for users to enter content. The position and size of the text box should be adjusted according to the page design and should include a clear input prompt or label.
- After clicking, one of the following elements must be displayed: <input type='text'> or <input type='email'>; <textarea>; an element with the attribute [contenteditable='true']; an element with the attribute [role='textbox']; or the container with the ID newsletterContainer becomes visible.

Figure 40: Prompt – Click Interaction

2646
2647
2648
2649
2650
2651
2652
2653
2654
2655
2656
2657
2658
2659
2660
2661
2662
2663
2664
2665
2666
2667
2668
2669
2670
2671
2672
2673
2674
2675
2676
2677
2678
2679
2680
2681
2682
2683
2684
2685
2686
2687
2688
2689
2690
2691
2692
2693
2694
2695
2696
2697
2698
2699

---

**Prompt – Hover Interaction**

**Interaction hover 1**
- Function Description: Implement an interactive element (id="#InteractionPart") that displays a dropdown box when the mouse hovers over it. The content of the dropdown should be generated based on the understanding of the page content, and its position and display effect should adapt to the page layout.
- When hovering, an element that matches one of the following selectors must be displayed: .dropdown-menu, .dropdown-content, [role='menu'], or the element with the ID placement-Dropdown.
- The dropdown element must be hidden before hovering.

**Interaction hover 2**
- Function Description: Implement an interactive element (id="#InteractionPart") containing text that becomes bold when the mouse hovers over it. Ensure that the bold effect is clearly displayed.
- When hovering, the text must become bold, with a fontWeight value of $\geq 600$ or set to bold/bolder.

**Interaction hover 3**
- Function Description: Implement an interactive element (id="#InteractionPart") containing text that adds an underline when the mouse hovers over it. The underline effect should be consistent with the page style.
- When hovering, the computed style of the element's textDecoration must include "underline".

**Interaction hover 4**
- Function Description: Implement an interactive element (id="#InteractionPart") that displays a tooltip when the mouse hovers over it, providing additional information or explanation. The content and style of the tooltip should be adapted according to the page context.
- When hovering, at least one of the following elements must be displayed: an element with the class .tooltip or .tip; a direct child element of the interactive element (#InteractionPart) with the class .tooltip-text; an element with the attribute [role='tooltip']; an element with the attribute [data-tooltip].
- The tooltip element must be invisible before hovering.

Figure 41: Prompt – Hover Interaction.

2700
2701
2702
2703
2704
2705
2706
2707
2708
2709
2710
2711
2712
2713
2714
2715
2716
2717
2718
2719
2720
2721
2722
2723
2724
2725
2726
2727
2728
2729
2730
2731
2732
2733
2734
2735
2736
2737
2738
2739
2740
2741
2742
2743
2744
2745
2746
2747
2748
2749
2750
2751
2752
2753

**Prompt – Text to Code Evaluation**

The user will provide you with a comprehensive textual description of a webpage, covering its layout, elements, and other relevant details. Based on this description, please generate the corresponding front-end code (including HTML, CSS, and JavaScript) to closely replicate the webpage as described.

If any images are required in the page code, you must classify each image into one of the following categories: People, Animal, Food, Plant, Landscape, Icon, Logo, Architecture, Technology, Transportation, Map, Texture, Art, Movie, Other. The URL for each image is in the format: `https://foxed_part/{category}.jpg`. For example, if you want to use an image from the Animal category, the image URL would be: `https://foxed_part/{Animal}.jpg`. Since the width and height of each image in the URLs are unknown, you need to set the image sizes manually in the HTML code to ensure that the rendered result matches the layout requirements of the webpage image.

Please place the HTML, CSS, and JavaScript code in a single file, and your response must be a code that starts with '<!DOCTYPE html>' and ends with '</html>', without any other content.

{Webpage Screenshot Description}

Figure 42: Prompt – Text to Code Evaluation.

**Prompt – Code Refinement Evaluation**

The user will provide you with a screenshot of a webpage, as well as a piece of front-end code. This code is the user's initial draft based on the webpage, but it cannot fully replicate the webpage. Please generate the front-end code that can fully replicate the webpage based on the screenshot and the existing code.

If any images are required in the page code, you must classify each image into one of the following categories: People, Animal, Food, Plant, Landscape, Icon, Logo, Architecture, Technology, Transportation, Map, Texture, Art, Movie, Other. The URL for each image is in the format: `https://foxed_part/{category}.jpg`. For example, if you want to use an image from the Animal category, the image URL would be: `https://foxed_part/{Animal}.jpg`. Since the width and height of each image in the URLs are unknown, you need to set the image sizes manually in the HTML code to ensure that the rendered result matches the layout requirements of the webpage image.

Please place the HTML, CSS, and JavaScript code in a single file, and your response must be a code that starts with '<!DOCTYPE html>' and ends with '</html>', without any other content.

{Webpage Screenshot}
{Corresponding HTML-v1}

Figure 43: Prompt – Code Refinement Evaluation.

