# OpenReview forum: "FullFront: Benchmarking MLLMs Across the Full Front-End Engineering Workflow"
_ICLR.cc/2026/Conference — Submitted to ICLR 2026_

### Official Review · Reviewer_1Ttx · 2025-10-30

**Soundness:** 3
**Presentation:** 2
**Contribution:** 3
**Rating:** 4
**Confidence:** 4

**Summary:**

Proposes a full-pipeline benchmark for front-end work with three tasks: Webpage Design, Perception QA, and Code Generation. Datasets span hundreds to thousands of items across these stages.

**Strengths:**

1. Evaluates the full front-end pipeline from concept, perception, and implementation (3 tasks, 8 sub-tasks).
2. Combines visual similarity with a Code Score to evaluate fine-grained elements.
3. Benchmarks a wide set of open-source and proprietary models and analyzes task-level differences.

**Weaknesses:**

Minor weaknesses:
1. Typos and labeling inconsistency.
2. Numerical inconsistencies in reported results and category counts.

Major issues:
1. Incomplete specification of the Code Score.
2. Same-family bias from using a model as the judge while evaluating related models.
3. Data-pipeline bias due to using certain models to construct the dataset.

**Questions:**

1. Typo: In Fig.6, "InterenVL3-78B" should be "InternVL3-78B" (Line 435); In the caption of Table 6, "Blacnk" and "Isonlation" (Line447-448). In Table 1, "DiINOv2" (Line272). "outperforme" in Line 107.
2. Unreasonable numbers: In Table 3, the GPT-5 just got 0.58 on Img of Gemini Visual Score (Line 307); The authors mentioned "15 categories", but claimed "twelve categories" in Figure 31.
3. Missing weight values: the actual weights of Code Score are missing. I am concerned about the reproducibility.
4. Model-as-judge bias: using Gemini 2.5 Flash as the visual judge while also evaluating Gemini models risks same-family bias.
5. Data source with bias: HTML-v1 built by GPT-4o and refined to HTML-v2 by Claude 3.7 Sonnet, then those models are evaluated. The evaluation metrics may favor Claude and GPT-4o.

---

> ### Author Response · Authors · 2025-11-22
> **Response to Reviewer iTtx (1/2)**
>
> > **Q1: Small Typos**
>
> A1: We sincerely thank you for your meticulous review. We have corrected all the noted typos (e.g., in Fig. 6 and Table 6) and resolved the numerical inconsistencies in the final manuscript. Specifically, we corrected the GPT-5 Gemini Visual Score in Table 3 (a transcription error, updated to 8.58) and rectified the category count in Figure 31 to "fifteen" to match our actual experimental setup.
>
> We will conduct a rigorous proofreading of the entire paper to ensure no similar errors remain.
>
> > **Q2: Specification of the Code Score**
>
> A2: We agree that precise weight specifications are essential for reproducibility. We will explicitly list the weights used in our experiments in the Appendix of the final revision. The specific values are:
>
> - Structure: $W_{struct} = 0.25$
>
> - Text: $W_{text\_content} = 0.20$, $W_{text\_style} = 0.10$
>
> - Visual Elements: $W_{image} = 0.20$, $W_{form} = 0.25$
>
> These weight details are also strictly defined in our provided evaluation scripts (included in the supplementary material) to ensure exact reproducibility for future research.
>
> > **Q3: Model-as-judge bias**
>
> A3: We appreciate your rigorous focus on evaluation fairness. We acknowledge that using a Gemini-based judge to evaluate Gemini model carries a risk of "same-family bias."
>
> To explicitly quantify and mitigate this risk, we performed **cross-judge validation** on 200 samples (50 per Code Generation subtask), comparing Human experts against three proprietary models and Qwen3-VL-235B-A22B-Instruct (**as an open-source alternative**).
>
> **Table R1: Inter-rater Reliability (IRR) and Ranking Consistency.** Format: Avg Score (Bias vs. Human, Rank). Metrics: Spearman $\rho$ and ICC measure sample-level consistency ($N$=1200); Kendall's $\tau$ measures aggregated model ranking alignment ($N$=6).
> |Model|Human|Gemini 2.5 Flash|Claude 3.7 Sonnet|Grok 4|Qwen3-VL-235B-A22B-Instruct|
> |:---|:---:|:---:|:---:|:---:|:---:|
> |GPT-5|8.50 (R1)|8.87 (+0.37, R1)|7.67 (-0.83, R1)|9.02 (+0.52, R1)|8.92 (+0.42, R1)|
> |Claude 3.7 Sonnet|8.35 (R2)|8.73 (+0.39, R2)|7.51 (-0.84, R2)|8.97 (+0.63, R2)|8.83 (+0.49, R2)|
> |Gemini 2.5 Flash|8.23 (R3)|8.61 (+0.38, R3)|7.38 (-0.86, R3)|8.92 (+0.68, R3)|8.70 (+0.46, R3)|
> |GPT-4o|6.67 (R4)|6.72 (+0.05, R4)|5.93 (-0.74, R4)|8.19 (+1.52, R4)|7.34 (+0.67, R4)|
> |Qwen2.5-VL-72B-Instruct|5.96 (R5)|5.98 (+0.02, R5)|5.31 (-0.65, R5)|8.13 (+2.17, R5)|6.51 (+0.55, R5)|
> |InternVL3-78B|5.49 (R6)|5.34 (-0.15, R6)|4.96 (-0.53, R6)|7.98 (+2.50, R6)|6.13 (+0.65, R6)|
> |**Spearman ρ**|1.00|0.769|0.805|0.361|0.764|
> |**ICC**|1.00|0.846|0.873|0.457|0.851|
> |**Kendall's τ**|1.0|1.0|1.0|1.0|1.0|
>
> As shown in Table R1, we observe no evidence of self-preference bias in the Gemini 2.5 Flash judge.
> - **No Self-Preference:** Gemini 2.5 Flash demonstrates strict objectivity. It ranks itself consistently below GPT-5 and Claude 3.7, perfectly matching human ranking (τ = 1.0). Crucially, its score inflation is **systematic** rather than self-serving: the bias toward itself (+0.38) is effectively identical to the bias toward its top competitors, GPT-5 (+0.37) and Claude 3.7 (+0.39). This pattern of uniform bias is corroborated by other judges (e.g., Claude 3.7 Judge subtracts ≈ 0.84 and Qwen3-VL adds ≈ 0.45 across these three models), confirming that Gemini’s scoring reflects a shared, objective standard rather than a specific adjustment to boost its relative standing.
>
> - **Robust Rubric:** All judges achieved **τ=1.0**, demonstrating that our detailed scoring rubric effectively guides diverse models to replicate human ranking logic, reducing reliance on specific model priors.
>
> - **Open-Source Alternative:** We identified Qwen3-VL-235B-A22B-Instruct as an excellent **drop-in open-source judge**, offering reliability (ICC=0.851) comparable to proprietary models.
>
> Furthermore, we measured judge stability over 3 independent runs with temperature=0 (where applicable).
>
> **Table R2: Test-Retest Reliability (3 runs)** Std Dev measures score variance; Inter-Run Correlation measures sample-level score consistency.
> |Judge Model|Avg. Score ± Std Dev|Inter-Run Correlation|
> |:---|:---:|:---:|
> |Gemini 2.5 Flash|7.38 ± 0.080|0.965|
> |Claude 3.7 Sonnet|6.46 ± 0.300|0.876|
> |Grok 4|8.53 ± 1.097|0.410|
> |Qwen3-VL|7.74 ± 0.079|0.972|
>
> Table R2 demonstrates exceptional stability for Gemini 2.5 Flash (σ<0.1, Correlation>0.96) and validates Qwen3-VL-235B-A22B-Instruct as a highly reliable open-source alternative (ICC=0.851, Correlation=0.972). To ensure strict reproducibility, we will release **the entire codebase, the static dataset, human score logs, and all model generation results**, enabling future researchers to fully reproduce our judging outcomes.

---

> > ### Author Response · Authors · 2025-11-22
> > **Response to Reviewer iTtx (2/2)**
> >
> > > **Q4: Data Source with Bias**
> >
> > A4: We acknowledge the concern that constructing the dataset using a pipeline involving GPT-4o and Claude 3.7 Sonnet might create an evaluation distribution that unfairly favors these specific models due to stylistic overfitting.
> >
> > To investigate this, we utilized **Grok-4** (which was not included in our original pipeline) to construct a completely new dataset of 300 samples (100 for each of the Image to Code, Text to Code, and Code Refinement subtasks). We then evaluated six representative models on this unseen dataset.
> >
> > **Table R3: Results on Grok-4 Constructed Data** Scores are averaged across three tasks. Spearman $\rho$ measures the correlation between aggregated model-level metric scores and human scores.
> > |Model|Code Score|Gemini Visual Score|CLIP Score|DINOv2 Score|Human Score|
> > |:---|:---:|:---:|:---:|:---:|:---:|
> > |Qwen2.5-VL-72B-Instruct|0.32|4.91|0.71|0.51|5.31|
> > |InternVL3-78B|0.29|5.66|0.73|0.50|5.58|
> > |Claude 3.7 Sonnet|0.49|7.88|0.78|0.63|7.76|
> > |Gemini 2.5 Flash|0.44|7.51|0.79|0.66|7.21|
> > |GPT-4o|0.44|6.71|0.75|0.55|5.94|
> > |GPT-5|0.55|7.90|0.83|0.69|7.92|
> > |**Spearman ρ**|0.94|1.00|0.94|0.89|1.00|
> >
> > The results in Table R3 confirm that our benchmark measures fundamental capabilities rather than pipeline-specific artifacts:
> >
> > - **Consistent Performance Hierarchy:** Even on Grok-4 generated data, the model rankings remain consistent with our main benchmark: GPT-5 > Claude 3.7 ≈ Gemini 2.5 > GPT-4o > Open-source Models.
> >
> > - **Robustness:** This consistency confirms the hierarchy is not a synthesis artifact, as our robustness stems from a pipeline that **originates from real-world screenshots**, utilizing models solely for refinement rather than generation from scratch. Furthermore, our Code/Gemini scores maintain high correlation (ρ ＞ 0.94) with human judgement.
> >
> > - **Data Release:** To ensure transparency and facilitate further research, we will release this new Grok-constructed dataset alongside the original FullFront dataset.
> >
> > Thank you again for your valuable time and effort in reviewing our manuscript. We will incorporate these new experiments and analyses into the revised manuscript.

---

### Official Review · Reviewer_JYpf · 2025-10-30

**Soundness:** 3
**Presentation:** 3
**Contribution:** 2
**Rating:** 4
**Confidence:** 4

**Summary:**

This work proposes FullFront, a unified benchmark evaluating the full front-end engineering pipeline: Webpage Design, Webpage Perception QA, and Webpage Code Generation.

**Strengths:**

1. Compared to prior works, FullFront transforms real-world websites into clean, standardized HTML to avoid copyright issues.
2. Many models are comprehensively benchmarked on three subtasks.

**Weaknesses:**

1. While the authors claim they unify multiple components into one cohesive evaluation pipeline, the implementation and results look like three separate benchmarks to me; the analysis of the connection between different parts seems weak.
2. Webpage Design seems to benchmark text-to-image generation capability. This part is kinda less motivated, since why do we want such MLLM to generate a website in image form? What makes it necessary to ask them to generate an image instead of generating code + rendering?
3. Misleading sample size: It seems that a non-trivial amount of experiments is conducted on FullFront-mini, which only contains 10 webpage design data points and 50 webpage code generation data points. The selection process is not justified.

**Questions:**

See Weaknesses

---

> ### Author Response · Authors · 2025-11-22
> **Response to Reviewer JYpf (1/2)**
>
> We appreciate your feedback and have carefully addressed your concerns in the detailed responses below.
>
> > **Q1:Benchmark Cohesion & Inter-Task Connectivity**
>
> A1: To address the concern regarding cohesiveness, we clarify that FullFront is not a mere aggregation of benchmarks, but a unified framework essential for evaluating the full front-end engineering lifecycle.
>
> **1. Uncovering Inter-Capability Dependencies**
>
> We diagnosed the causal link between Perception ($p$) and Code Implementation ($c$) by manually verifying 100 samples from Synthetic QA against their Image to Code outputs, classifying behaviors into four quadrants based on **Perception ($p$)** (QA accuracy) and **Code Implementation ($c$)** (correct implementation of the specific component mentioned in the QA).
>
> **Table R1: Confusion matrix of ($p$) vs. ($c$)**
> |Model|p√c√|p√c×|p×c√|p×c×|
> |:---|:---:|:---:|:---:|:---:|
> |Qwen2.5-VL-72B-Instruct|22|28|10|40|
> |InternVL3-78B|25|23|8|44|
> |GPT-4o|32|13|21|34|
> |Claude 3.7 Sonnet|44|3|36|17|
> |Gemini 2.5 Flash|39|5|25|31|
> |GPT-5|48|6|31|15|
>
> - **"Bottleneck Shift" Phenomenon:** Table R1 reveals distinct failure modes. Open-source models exhibit high $p√c×$ counts, indicating that even with correct perception, **coding capability** is the primary bottleneck. Conversely, advanced proprietary models (e.g., Claude 3.7) show high $p×c√$ counts. This suggests these models often rely on **strong internal code priors** to bypass fine-grained visual perception, essentially generating "correct" code without truly "seeing" the details. This critical insight into Grounding Failure is strictly unique to our aggregated framework.
>
> To explore further, we analyzed generation scores grouped by whether the model answered all perception QA questions for a webpage correctly.
>
> **Table R2: Generation performance grouped by perception accuracy**
> |Model|Perception Group|Code Score|Gemini Visual Score|CLIP Score|DINOv2 Score|
> :---|:---|:---:|:---:|:---:|:---:|
> |Qwen2.5-VL-72B-Instruct|Correct (N=46)|0.386|4.430|0.705|0.409|
> ||Wrong (N=48)|0.405|5.117|0.753|0.529|
> |InternVL3-78B|Correct (N=59)|0.418|4.444|0.736|0.554|
> ||Wrong (N=42)|0.409|4.422|0.715|0.486|
> |GPT-4o|Correct (N=43)|0.336|6.007|0.799|0.672|
> ||Wrong (N=56)|0.342|5.896|0.809|0.697|
> |Claude 3.7 Sonnet|Correct (N=57)|0.646|8.826|0.887|0.805|
> ||Wrong (N=44)|0.649|8.957|0.892|0.790|
> |Gemini 2.5 Flash|Correct(N=42)|0.634|8.790|0.890|0.838|
> ||Wrong (N=72)|0.628|8.596|0.877|0.796|
> |GPT-5|Correct (N=75)|0.568|8.436|0.882|0.808|
> ||Wrong (N=44)|0.535|8.548|0.878|0.832|
>
> - **Decoupling Capabilities:** As shown in Table R2, macroscopic scores across all metrics remain consistent regardless of perception accuracy, implying that fine-grained perceptual errors do not significantly degrade page appearance. This reveals a **decoupling** between perception and generation: models appear to rely on **strong internal priors** rather than precise visual understanding. FullFront uniquely exposes this underlying lack of robust visual grounding, whereas isolated benchmarks would misinterpret strong generation performance as definitive "success."
>
> - **Note:** Qwen2.5-VL-72B-Instruct's lower score in the "Correct" group stems from a disproportionate occurrence of fatal errors (e.g., "Abnormal Image Size," Table 6 in the main paper) in that subset, reflecting generation instability rather than a negative causal link.
>
> **2. Comparative Analysis of Input Modalities**
>
> Aggregation allows direct comparison of input efficacy. Results (Tables 3 & 4 in the main paper) show that Code Refinement outperforms scratch generation. Surprisingly, **Text to Code rivals Image to Code**; notably, Appendix Table 9 confirms that even text-only LLMs achieve highly competitive results when provided with detailed descriptions. This indicates that precise textual specifications can effectively substitute for visual inputs—a critical insight for designing efficient agents.
>
> **3. Simulating the Real-World Workflow & Future-Proofing**
>
> While no single current model can yet master the entire front-end workflow, FullFront is designed to:
> - **Simulate the Real-World Workflow:**  Professional front-end engineering is a sequential pipeline where every stage is indispensable: Conceptualization (creating visual drafts via tools like Figma) $\to$ Comprehension (analyzing designs) $\to$ Implementation (coding). Evaluating specialized models within this framework mirrors the authentic engineering lifecycle.
>
> - **Future-Proofing:** As the field moves toward unified MLLMs, FullFront provides the necessary framework to evaluate future "Generalist Front-End Agents." It serves to benchmark the progress of future agents while pinpointing the specific shortfalls of current models across these essential sub-capabilities.

---

> > ### Author Response · Authors · 2025-11-22
> > **Response to Reviewer JYpf (2/2)**
> >
> > > **Q2: Motivation for the Webpage Design Task**
> >
> > A2: Regarding the motivation for image generation, while we understand the inclination towards code generation, we believe the Webpage Design task (Conceptualization) serves a unique and essential role in the front-end workflow for three key reasons:
> >
> > **1. Simulating Real-World Prototyping**
> >
> > In professional workflows, visual prototypes (e.g., Figma drafts) are **indispensable prerequisites to implementation**. They serve as low-cost environments to resolve requirement ambiguity, iterate on spatial layouts, and validate aesthetics before any engineering work begins. By generating a design image, we simulate this essential "Architect" phase, evaluating whether an agent can effectively plan a coherent visual structure—a fundamental step in real-world product development.
> >
> > **2. Decoupling Design from Implementation**
> >
> > Regarding the approach of 'generating code + rendering,' we have integrated this specific evaluation into our Text to Code task. However, evaluating design solely through code conflates two distinct challenges: visual planning and coding skill. Isolating the Webpage Design task allows us to evaluate the model purely as a "Designer" (assessing spatial reasoning and aesthetics) separate from the "Engineer" (assessing syntax and logic). Just as a product team evaluates a UI designer's mockups differently from a developer's pull request, these tasks assess fundamentally different competencies.
> >
> > **3. "Visual Chain of Thought"**
> >
> > We view the generated design image as a necessary intermediate representation (a form of "Visual Chain of Thought") that grounds abstract textual requirements into concrete visual decisions. While current models may not yet seamlessly execute every stage, **evaluating this specific capability is crucial for future agents** that will likely follow a Text $\to$ Visual Draft $\to$ Refined Code pipeline to handle complex user requests more effectively.
> >
> > > **Q3: Clarification on Sample Size and Mini-Set Validity**
> >
> > A3: We appreciate the opportunity to clarify the experimental scope. We wish to highlight that the **primary conclusions and results** in our paper (specifically Tables 1, 2, and 3) are derived from the Full Dataset (1,800 QA pairs, 200 Design samples, 700 Code problems) for 10 out of the 12 evaluated models .
> >
> > **1. Purpose and Selection of the Mini-Set**
> >
> > The FullFront-mini was designed not as the primary evaluation standard, but as a **low-cost proxy tool** to facilitate rapid iteration and community accessibility. The data points were selected via stratified random sampling to ensure balanced coverage across all subtasks and difficulty levels, ensuring it is statistically representative.
> >
> > **2. Expansion to "FullFront-mini-250"**
> >
> > To definitively address your concern regarding the sample size of the mini-set, **we significantly expanded it to create FullFront-mini-250**, consisting of 50 Webpage Design samples and 200 Code Generation samples (50 per subtask). We conducted **a complete re-evaluation, including a new round of Human Evaluation**.
> >
> > **Table R3: Evaluation of Webpage Design on Expanded Mini-Set (N=50)**
> > |Model|Gemini Visual Score|CLIP Score|DINOv2 Score|
> > |:---|:---:|:---:|:---:|
> > |GPT-4o|4.0960|0.7508|0.5699|
> > |gemini-2.0-flash-exp-image-generation|1.0180|0.6793|0.4771|
> >
> > **Table R4: Evaluation of Webpage Code Generation on Expanded Mini-Set (N=200)** Scores are averaged across four subtasks.
> > |Model|Code Score|Gemini Visual Score|CLIP Score|DINOv2 Score|Human Score|
> > |:---|:---:|:---:|:---:|:---:|:---:|
> > |Qwen2.5-VL-72B-Instruct|0.44|5.61|0.75|0.58|5.96|
> > |InternVL2.5-78B|0.37|4.44|0.72|0.57|4.86|
> > |InternVL3-78B|0.44|4.70|0.72|0.52|5.49|
> > |LLaVA-Onevision-72B|0.23|3.21|0.68|0.44|3.40|
> > |Claude 3.7 Sonnet|0.60|8.63|0.87|0.79|8.35|
> > |Gemini 2.5 Flash|0.62|8.41|0.86|0.78|8.23|
> > |GPT-4o|0.38|6.12|0.78|0.64|6.67|
> > |o4-mini|0.56|8.05|0.84|0.74|7.72|
> > |GPT-4.1|0.58|8.86|0.86|0.80|8.57|
> > |GPT-5|0.56|8.65|0.88|0.79|8.50|
> >
> > The results on the expanded FullFront-mini-250 (Table R3 and R4) show **strong consistency with the Full Dataset results** reported in the main paper (Table 3). The relative rankings (e.g., Claude 3.7/GPT-5 leading, open-source models lagging) remain stable, validating that the random sampling strategy effectively captures the benchmark's distribution.
> >
> > To ensure full transparency and reproducibility, we will:
> > - **Update the final paper** to include these expanded result tables and the corresponding analytical text.
> >
> > - **Open-Source All Resources:** We will release the Full Dataset, the original mini-set, and the new FullFront-mini-250, alongside **all human score logs and model generation artifacts**, enabling the community to fully verify our findings and perform their own analyses.
> >
> > Thank you again for your valuable time and effort in reviewing our manuscript. We will incorporate these new experiments and analyses into the revised manuscript.

---

> ### Comment · Reviewer_JYpf · 2025-11-25
>
> Thanks for the newly added study. I am happy to increase my score.

---

> > ### Author Response · Authors · 2025-11-26
> >
> > Thank you again for your re-evaluation and constructive suggestions. We truly appreciate your positive assessment, which we view as a strong affirmation of our work. We will ensure that the relevant analysis and discussions are incorporated into the final version of the paper. We sincerely appreciate your help in improving our manuscript.

---

### Official Review · Reviewer_gXbA · 2025-11-02

**Soundness:** 3
**Presentation:** 3
**Contribution:** 3
**Rating:** 6
**Confidence:** 4

**Summary:**

This paper introduces FullFront, a benchmark that evaluates MLLMs across the full front-end workflow—design, perception, and code—using reconstructed real-page data and complementary visual/code metrics validated against human judgments. Experiments reveal substantial gaps from human performance, with persistent weaknesses in fine-grained perception, layout fidelity, image handling, and interaction implementation despite strong results from leading proprietary models. The work is timely and likely useful for the community, though its reliance on closed models and the potential metric bias should be weighed carefully.

**Strengths:**

+ The benchmark mirrors real front-end workflows rather than a single slice: it covers conceptualization (Webpage Design), perception (Webpage Perception QA), and implementation (Webpage Code Generation) with concrete task counts across eight subtasks. This end-to-end framing is rare and useful for diagnosing capability gaps.
+ The dataset is grounded in real webpages and reconstructed into standardized, copyright-safe HTML through a two-stage, MLLM-assisted pipeline, addressing common issues of bloated scraped code and oversimplified LLM HTML in prior corpora.
+ The evaluation combines visual and code-level metrics and validates them against human judgments with high Spearman correlations, supporting automated evaluation as a reasonable proxy for human preference.

**Weaknesses:**

- Several construction and scoring steps rely on proprietary models (e.g., GPT-4o/Claude in the pipeline; Gemini-based visual scoring), which can introduce system bias and limit strict reproducibility; data/code release is contingent on acceptance.
- The engineered “Code Score” aggregates DOM and style attributes with fixed design choices; even with strong human correlation, such choices may privilege particular implementation patterns and under-reward acceptable alternatives.
- The results clearly show large model–human gaps (e.g., <60% model accuracy vs. >95% human on perception), but the causal link between perception skill and code quality remains only lightly probed; deeper ablations could clarify what actually transfers across stages.

**Questions:**

1. How reproducible is the dataset and evaluation given reliance on closed models (GPT-4o/Claude for curation; Gemini for visual scoring)? Please quantify any bias toward these systems and provide an open, drop-in alternative or calibration protocol.


2. “Code Score” encodes specific structural/style weights. What sensitivity and failure analyses show that valid alternative implementations aren’t systematically penalized? Include counterexamples where humans deem outputs equivalent but the metric disagrees.

---

> ### Author Response · Authors · 2025-11-22
> **Response to Reviewer gXbA (1/2)**
>
> We appreciate your feedback and have carefully addressed your concerns in the detailed responses below.
> >**Q1: Judge bias and reproducibility concern**
>
> A1: While proprietary models remain necessary for high-quality synthesis, extensive new experiments demonstrate FullFront is **robust, system-agnostic, and reproducible**.
>
> **1. Investigating Construction Bias**
>
> To determine if FullFront reflects fundamental capabilities rather than system bias, we utilized **Grok-4** (absent from original pipeline) to construct a dataset of 300 samples (100 per task: Image/Text/Refine) and evaluated six representative models on this unseen data.
>
> **Table R1: Results on Grok-4 Constructed Data** Scores are averaged across 3 tasks. Spearman ρ measures the correlation between aggregated model-level metric scores and human scores.
> |Model|Code Score|Gemini Visual Score|CLIP Score|DINOv2 Score|Human Score|
> |:---|:---:|:---:|:---:|:---:|:---:|
> |Qwen2.5-VL-72B-Instruct|0.32|4.91|0.71|0.51|5.31|
> |InternVL3-78B|0.29|5.66|0.73|0.50|5.58|
> |Claude 3.7 Sonnet|0.49|7.88|0.78|0.63|7.76|
> |Gemini 2.5 Flash|0.44|7.51|0.79|0.66|7.21|
> |GPT-4o|0.44|6.71|0.75|0.55|5.94|
> |GPT-5|0.55|7.90|0.83|0.69|7.92|
> |**Spearman ρ**|0.94|1.00|0.94|0.89|1.00|
>
> The results confirm our benchmark **measures fundamental capabilities rather than pipeline-specific artifacts**:
> - **Consistent Performance Hierarchy:** Even on Grok-4 generated data, the model rankings remain consistent with our main benchmark: GPT-5 > Claude 3.7 ≈ Gemini 2.5 > GPT-4o > Open-source Models.
>
> - **Robustness:** This consistency confirms the hierarchy is not a synthesis artifact, as our robustness stems from a pipeline that **originates from real-world screenshots**, utilizing models solely for refinement rather than generation from scratch. Furthermore, our Code/Gemini scores maintain high correlation (ρ＞0.94) with human judgement.
>
> - **Data Release:** To ensure transparency, we will release this new Grok-constructed dataset alongside the original FullFront dataset.
>
> **2. Quantifying Judge Bias**
>
> We performed cross-judge validation on 200 samples (50 per Code Generation subtask), comparing Human experts against three proprietary models and Qwen3-VL-235B-A22B-Instruct (**as an open-source alternative**).
>
> **Table R2: Inter-rater Reliability (IRR) and Ranking Consistency.** Format: Avg Score (Bias vs. Human, Rank). Metrics: Spearman ρ and ICC measure sample-level consistency (N=1200); Kendall's τ measures aggregated model ranking alignment (N=6).
> |Model|Human|Gemini 2.5|Claude 3.7|Grok 4|Qwen3-VL|
> |:---|:---:|:---:|:---:|:---:|:---:|
> |GPT-5|8.50 (R1)|8.87 (+0.37, R1)|7.67 (-0.83, R1)|9.02 (+0.52, R1)|8.92 (+0.42, R1)|
> |Claude 3.7 Sonnet|8.35 (R2)|8.73 (+0.39, R2)|7.51 (-0.84, R2)|8.97 (+0.63, R2)|8.83 (+0.49, R2)|
> |Gemini 2.5 Flash|8.23 (R3)|8.61 (+0.38, R3)|7.38 (-0.86, R3)|8.92 (+0.68, R3)|8.70 (+0.46, R3)|
> |GPT-4o|6.67 (R4)|6.72 (+0.05, R4)|5.93 (-0.74, R4)|8.19 (+1.52, R4)|7.34 (+0.67, R4)|
> |Qwen2.5-VL-72B-Instruct|5.96 (R5)|5.98 (+0.02, R5)|5.31 (-0.65, R5)|8.13 (+2.17, R5)|6.51 (+0.55, R5)|
> |InternVL3-78B|5.49 (R6)|5.34 (-0.15, R6)|4.96 (-0.53, R6)|7.98 (+2.50, R6)|6.13 (+0.65, R6)|
> |**Spearman ρ**|1.00|0.769|0.805|0.361|0.764|
> |**ICC**|1.00|0.846|0.873|0.457|0.851|
> |**Kendall's τ**|1.0|1.0|1.0|1.0|1.0|
>
> - **No Self-Preference:** Gemini 2.5 Flash demonstrates strict objectivity. It ranks itself consistently below GPT-5 and Claude 3.7, perfectly matching human ranking (τ = 1.0). Crucially, its score inflation is **systematic** rather than self-serving: the bias toward itself (+0.38) is effectively identical to the bias toward its top competitors, GPT-5 (+0.37) and Claude 3.7 (+0.39). This pattern of uniform bias is corroborated by other judges (e.g., Claude 3.7 Judge subtracts ≈ 0.84 and Qwen3-VL adds ≈ 0.45 across these three models), confirming that Gemini’s scoring reflects a shared, objective standard rather than a specific adjustment to boost its relative standing.
>
> - **Open-Source Alternative:** We identified Qwen3-VL-235B-A22B-Instruct as an excellent drop-in open-source judge, offering reliability **(ICC=0.851)** comparable to proprietary models.
>
> **3. Reproducibility & Test-Retest Reliability** We measured judge stability over 3 independent runs with temperature=0 (where applicable).
>
> **Table R3: Test-Retest Reliability (3 runs)**
> |Judge Model|Avg. Score ± Std Dev|Inter-Run Correlation|
> |:---|:---:|:---:|
> |Gemini 2.5 Flash|7.38 ± 0.080|0.965|
> |Claude 3.7 Sonnet|6.46 ± 0.300|0.876|
> |Grok 4|8.53 ± 1.097|0.410|
> |Qwen3-VL|7.74 ± 0.079|0.972|
>
> Table R3 demonstrates exceptional stability for Gemini 2.5 Flash (σ < 0.1, Correlation > 0.96) and validates Qwen3-VL-235B-A22B-Instruct as a highly reliable open-source alternative (ICC=0.851, Correlation=0.972). To ensure strict reproducibility, we will release **the entire codebase, the static dataset, human score logs, and all model generation results**, enabling future researchers to fully reproduce our judging outcomes.

---

> > ### Author Response · Authors · 2025-11-22
> > **Response to Reviewer gXbA (2/2)**
> >
> > >**Q2: Analysis of Code Score**
> >
> > A2: We appreciate the concern regarding Code Score's potential rigidity. However, its exceptional human correlation (**$\rho=0.9038$**, table 10 in the main paper) validates its reliability. Code Score assesses structural integrity and semantic correctness, forming a robust "Dual-Axis Evaluation" alongside our suite of perceptual visual metrics.
> >
> > To quantify tolerance for valid alternatives, we conducted a **Sensitivity Analysis** on 50 random samples, refactoring Ground Truth code into two groups:
> > - **Exp 1: Syntactic Variation (Valid Alternative):** Rewrote code using different CSS layout techniques (e.g., swapping Flexbox for Grid) while maximizing visual preservation
> >
> > - **Exp 2: Semantic Violation (Broken):** Introduced structural degradations (e.g., removing containers, altering text content) that significantly impacted the visual result.
> >
> > **Table R4: Code Score Sensitivity Analysis.**
> > |Experiment Condition|Gemini Visual Score|Code Score|CLIP Score|DINOv2 Score|
> > |:---|:---|:---|:---|:---|
> > |Original HTML|10.0|1.0|1.0|1.0|
> > |Exp 1|9.266 (-0.734)|0.952 (-0.048)|0.969 (-0.031)|0.942 (-0.058)|
> > |Exp 2|6.838 (-3.162)|0.727 (-0.273)|0.846 (-0.154)|0.760 (-0.240)|
> > |Penalty Ratio ($Δ_2 / Δ_1$)|4.31x|**5.69x**|4.97x|4.14x|
> >
> > **Exceptional Discriminative Ability:** As shown in Table R4, Code Score drops significantly for violations (Δ=0.273) but negligibly for valid variations (Δ=0.048). The high Penalty Ratio (**5.69x**)—the highest among all metrics—demonstrates exceptional discriminative power, confirming the metric effectively **tolerates valid engineering choices while punishing substantive defects**.
> >
> > >**Q3: Causal Analysis of Perception-Code Transfer**
> >
> > **1. Micro-Level Analysis: The "Bottleneck Shift"**
> >
> > We diagnosed the causal link between Perception ($p$) and Code Implementation ($c$) by manually verifying 100 samples from Synthetic QA against their Image to Code outputs, classifying behaviors into four quadrants based on **Perception ($p$)** (QA accuracy) and **Code Implementation ($c$)** (correct implementation of the specific component mentioned in the QA).
> >
> > **Table R5: Confusion matrix of ($p$) vs. ($c$)**
> > |Model|p√c√|p√c×|p×c√|p×c×|
> > |:---|:---:|:---:|:---:|:---:|
> > |Qwen2.5-VL-72B-Instruct|22|28|10|40|
> > |InternVL3-78B|25|23|8|44|
> > |GPT-4o|32|13|21|34|
> > |Claude 3.7 Sonnet|44|3|36|17|
> > |Gemini 2.5 Flash|39|5|25|31|
> > |GPT-5|48|6|31|15|
> >
> > - **"Bottleneck Shift" Phenomenon:** Table R5 reveals distinct failure modes. Open-source models exhibit high $p√c×$ counts, indicating that even with correct perception, **coding capability** is the primary bottleneck. Conversely, advanced proprietary models (e.g., Claude 3.7) show high $p×c√$ counts. This suggests these models often rely on **strong internal code priors** to bypass fine-grained visual perception, essentially generating "correct" code without truly "seeing" the details. This critical insight into Grounding Failure is strictly unique to our aggregated framework.
> >
> > **2. Macro-Level Analysis: Decoupling of Capabilities**
> >
> > To quantify the impact of perception accuracy on overall generation quality, we grouped webpages by whether the model answered all QA questions correctly.
> >
> > **Table R6: Generation performance grouped by perception accuracy**
> > |Model|Perception Group|Code Score|Gemini Visual Score|CLIP Score|DINOv2 Score|
> > :---|:---|:---:|:---:|:---:|:---:|
> > |Qwen2.5-VL-72B-Instruct|Correct (N=46)|0.386|4.430|0.705|0.409|
> > ||Wrong (N=48)|0.405|5.117|0.753|0.529|
> > |InternVL3-78B|Correct (N=59)|0.418|4.444|0.736|0.554|
> > ||Wrong (N=42)|0.409|4.422|0.715|0.486|
> > |GPT-4o|Correct (N=43)|0.336|6.007|0.799|0.672|
> > ||Wrong (N=56)|0.342|5.896|0.809|0.697|
> > |Claude 3.7 Sonnet|Correct (N=57)|0.646|8.826|0.887|0.805|
> > ||Wrong (N=44)|0.649|8.957|0.892|0.790|
> > |Gemini 2.5 Flash|Correct(N=42)|0.634|8.790|0.890|0.838|
> > ||Wrong (N=72)|0.628|8.596|0.877|0.796|
> > |GPT-5|Correct (N=75)|0.568|8.436|0.882|0.808|
> > ||Wrong (N=44)|0.535|8.548|0.878|0.832|
> >
> > - **Decoupling Capabilities:** As shown in Table R6, macroscopic scores across all metrics remain consistent regardless of perception accuracy, implying that fine-grained perceptual errors do not significantly degrade page appearance. This reveals a **decoupling** between perception and generation: models appear to rely on **strong internal priors** rather than precise visual understanding. FullFront uniquely exposes this underlying lack of robust visual grounding, whereas isolated benchmarks would misinterpret strong generation performance as definitive "success."
> >
> > - **Note:** Qwen2.5-VL-72B-Instruct's lower score in the "Correct" group stems from a disproportionate occurrence of fatal errors (e.g., "Abnormal Image Size," Table 6 in the main paper) in that subset, reflecting generation instability rather than a negative causal link.
> >
> > Thank you again for your valuable time and effort in reviewing our manuscript. We will incorporate these new experiments and analyses into the revised manuscript.

---

### Official Review · Reviewer_fHMJ · 2025-11-02

**Soundness:** 2
**Presentation:** 2
**Contribution:** 2
**Rating:** 4
**Confidence:** 4

**Summary:**

FullFront is a new benchmark consisting of three types of tasks related to various aspects of the front-end development cycle, including: Webpage Design (conceptualization phase), Webpage Perception QA (comprehension of visual organization and elements), and Webpage Code Generation (implementation phase).

Webpage Design is an image generation task where the model needs to generate the webpage design based on textual descriptions of synthetic webpages. Webpage perception QA is a QA task based on both real and synthetic webpages, including 75 samples of multi-window QA tasks. The questions are generated by GPT-4o with manual inspection. The webpage code generation task takes visual design as input and expects code generation as output. This task includes image-to-code, text-to-code, interaction authoring, and code refinement.

The authors did benchmarking of various closed and open models on these tasks.

**Strengths:**

- New resource for benchmarking MLLM capability on front-end engineering.

- Relatively thorough coverage of different capabilities and model families.

- Nice to include to human evaluation too.

**Weaknesses:**

- I'm not fully convinced by why we need an aggregate benchmark for front-end development. Some of the capabilities are quite distinct. For example, design (image) generation vs QA vs code generation. You have to use different models for the benchmarking because most models can't do image generation at all. Then what's the point of putting all of these tasks into one benchmark?

- I understand the authors put in effort to curate new data for many of the tasks in the benchmark. But I believe for most of the sub-tasks, there exist prior benchmarks that test the same capability. What's the unique contribution here apart from putting all the result tables into one paper?

**Questions:**

See above.

---

> ### Author Response · Authors · 2025-11-22
> **Response to Reviewer fHMJ (1/2)**
>
> We appreciate your feedback and have carefully addressed your concerns in the detailed responses below.
> >**Q1: Necessity of Aggregate Benchmarking & Model Heterogeneity**
>
> A1: We appreciate your concern regarding the rationale for unifying distinct capabilities into a single benchmark. We firmly believe that FullFront is not merely an aggregation, but a cohesive framework necessary to evaluate the full front-end engineering lifecycle. By unifying these tasks, we can uniquely uncover latent dependencies and "Perception-Implementation Gaps" that isolated benchmarks inevitably miss.
>
> **1. Uncovering Inter-Capability Dependencies**
>
> We diagnosed the causal link between Perception ($p$) and Code Implementation ($c$) by manually verifying 100 samples from Synthetic QA against their Image to Code outputs, classifying behaviors into four quadrants based on **Perception ($p$)** (QA accuracy) and **Code Implementation ($c$)** (correct implementation of the specific component mentioned in the QA).
>
> **Table R1: Confusion matrix of ($p$) vs. ($c$)**
> |Model|p√c√|p√c×|p×c√|p×c×|
> |:---|:---:|:---:|:---:|:---:|
> |Qwen2.5-VL-72B-Instruct|22|28|10|40|
> |InternVL3-78B|25|23|8|44|
> |GPT-4o|32|13|21|34|
> |Claude 3.7 Sonnet|44|3|36|17|
> |Gemini 2.5 Flash|39|5|25|31|
> |GPT-5|48|6|31|15|
>
> - **"Bottleneck Shift" Phenomenon:** Table R1 reveals distinct failure modes. Open-source models exhibit high $p√c×$ counts, indicating that even with correct perception, **coding capability** is the primary bottleneck. Conversely, advanced proprietary models (e.g., Claude 3.7) show high $p×c√$ counts. This suggests these models often rely on **strong internal code priors** to bypass fine-grained visual perception, essentially generating "correct" code without truly "seeing" the details. This critical insight into Grounding Failure is strictly unique to our aggregated framework.
>
> To explore further, we analyzed generation scores grouped by whether the model answered all perception QA questions for a webpage correctly.
>
> **Table R2: Generation performance grouped by perception accuracy**
> |Model|Perception Group|Code Score|Gemini Visual Score|CLIP Score|DINOv2 Score|
> :---|:---|:---:|:---:|:---:|:---:|
> |Qwen2.5-VL-72B-Instruct|Correct (N=46)|0.386|4.430|0.705|0.409|
> ||Wrong (N=48)|0.405|5.117|0.753|0.529|
> |InternVL3-78B|Correct (N=59)|0.418|4.444|0.736|0.554|
> ||Wrong (N=42)|0.409|4.422|0.715|0.486|
> |GPT-4o|Correct (N=43)|0.336|6.007|0.799|0.672|
> ||Wrong (N=56)|0.342|5.896|0.809|0.697|
> |Claude 3.7 Sonnet|Correct (N=57)|0.646|8.826|0.887|0.805|
> ||Wrong (N=44)|0.649|8.957|0.892|0.790|
> |Gemini 2.5 Flash|Correct(N=42)|0.634|8.790|0.890|0.838|
> ||Wrong (N=72)|0.628|8.596|0.877|0.796|
> |GPT-5|Correct (N=75)|0.568|8.436|0.882|0.808|
> ||Wrong (N=44)|0.535|8.548|0.878|0.832|
>
> - **Decoupling Capabilities:** As shown in Table R2, macroscopic scores across all metrics remain consistent regardless of perception accuracy, implying that fine-grained perceptual errors do not significantly degrade page appearance. This reveals a **decoupling** between perception and generation: models appear to rely on **strong internal priors** rather than precise visual understanding. FullFront uniquely exposes this underlying lack of robust visual grounding, whereas isolated benchmarks would misinterpret strong generation performance as definitive "success."
>
> - **Note:** Qwen2.5-VL-72B-Instrut's lower score in the "Correct" group stems from a disproportionate occurrence of fatal errors (e.g., "Abnormal Image Size," Table 6 in the main paper) in that subset, reflecting generation instability rather than a negative causal link.
>
> **2. Comparative Analysis of Input Modalities**
>
> Aggregation allows direct comparison of input efficacy. Results (Tables 3 & 4 in the main paper) show that Code Refinement outperforms scratch generation. Surprisingly, **Text to Code rivals Image to Code**; notably, Appendix Table 9 confirms that even text-only LLMs achieve highly competitive results when provided with detailed descriptions. This indicates that precise textual specifications can effectively substitute for visual inputs—a critical insight for designing efficient agents.
>
> **3. Simulating the Real-World Workflow & Future-Proofing**
>
> While no single current model can yet master the entire front-end workflow, FullFront is designed to:
> - **Simulate the Real-World Workflow:**  Professional front-end engineering is a sequential pipeline where every stage is indispensable: Conceptualization (creating visual drafts via tools like Figma) $\to$ Comprehension (analyzing designs) $\to$ Implementation (coding). Evaluating specialized models within this framework mirrors the authentic engineering lifecycle.
>
> - **Future-Proofing:** As the field moves toward unified MLLMs, FullFront provides the necessary framework to evaluate future "Generalist Front-End Agents." It serves to benchmark the progress of future agents while pinpointing the specific shortfalls of current models across these essential sub-capabilities.

---

> > ### Author Response · Authors · 2025-11-22
> > **Response to Reviewer fHMJ (2/2)**
> >
> > >**Q2: Contribution over existing sub-task benchmarks**
> >
> > A2: While prior works explore individual capabilities like static screenshot-to-code and perception QA, FullFront goes beyond aggregation. We advance the field by introducing novel task formulations, a superior data construction pipeline, and a comprehensive evaluation framework addressing critical limitations of existing benchmarks.
> >
> > **1. Introduction of Novel, High-Value Tasks**
> >
> > FullFront extends beyond standard code generation to cover previously unexplored but essential stages of the front-end workflow:
> > - **Conceptualization (Webpage Design):** The first task evaluating text-to-image models on generating coherent visual designs from text—a "zero-to-one" creative step missing in prior benchmarks.
> >
> > - **Dynamic & Iterative Engineering:** We introduce Interaction Authoring (implementing behaviors like dropdowns) and Code Refinement (optimizing existing code), simulating the complex, iterative nature of real-world development overlooked in static generation tasks.
> >
> > **2. Superior Data Quality and Construction Pipeline**
> >
> > Existing datasets struggle to balance realism with cleanliness; those derived from sources like Common Crawl often contain bloated code, while purely synthetic ones lack visual complexity.
> > - **Hybrid Synthesis Pipeline:** We employ a novel two-stage reconstruction process (Real-World Extraction $\to$ LLM Refinement) to produce standardized, clean, yet visually diverse HTML that mirrors high-quality production code.
> >
> > - **Visual Fidelity:** As shown in Figure 3 in the main paper, our data demonstrates significantly higher visual complexity compared to previous benchmarks.
> >
> > - **Category-Based Image Strategy:** We devised a unique strategy using 15 distinct image categories to resolve copyright and placeholder issues, ensuring consistent, risk-free evaluation of image handling capabilities.
> >
> > **3. Fine-Grained Evaluation and Holistic Analysis**
> > - **Multi-Dimensional Metrics:** We introduce the Code Score for structural/content similarity and the Gemini Visual Score, utilizing a VLM judge with a detailed rubric.
> >
> > - **Holistic Analysis:** The unified framework enables unique cross-task analyses (e.g., the Perception-Implementation Gap) alongside detailed error analysis and human evaluation, which isolated benchmarks cannot support.
> >
> > Thank you again for your valuable time and effort in reviewing our manuscript. We will incorporate these new experiments and analyses into the revised manuscript.

---

> > ### Comment · Reviewer_fHMJ · 2025-11-25
> >
> > This is cool analysis, but from a benchmark perspective, I still don't feel convinced that you need to package this as a new benchmark in order to uncover these findings. For example, you can easily manipulate the vision input of existing Design2Code benchmarks to study whether models actually rely on the image input in code generation (and there are a bunch of existing papers doing this on other tasks).
> >
> > Given the overall response, I maintain my original scores.

---

> > > ### Author Response · Authors · 2025-11-26
> > > **Response to Reviewer fHMJ's Follow-up (1/2)**
> > >
> > > We sincerely appreciate your continued engagement and the opportunity to clarify the unique contributions of our work. While we understand your reservations, we argue that input manipulation on existing benchmarks (e.g., Design2Code) serves a fundamentally different purpose. FullFront is uniquely designed to diagnose the engineering workflow, not just detect performance.
> > >
> > > **1. The "Perception-Implementation Decoupling" Necessitates FullFront**
> > >
> > > Regarding the suggestion to manipulate vision inputs on Design2Code to study reliance, we acknowledge that while this validates that vision is used, it stops short of diagnosing the **causal link between perception and implementation**: specifically, whether correct code stems from accurate visual understanding, or if accurate perception fails to translate into functional code. This distinction is vital because our rebuttal analysis reveals a critical "Perception-Implementation Decoupling" that input manipulation alone cannot diagnose, as illustrated in [Figure R1: Case Study of Perception-Implementation Decoupling](https://anonymous.4open.science/api/repo/anonymous_things-D72C/file/FullFront_files/right_wrong.png?v=ba30b270):
> > >
> > > - **The "Illusion of Competence":** Advanced models often generate correct code despite incorrect visual perception, relying on strong internal priors. As shown in Figure R1, Claude 3.7 Sonnet explicitly fails the QA task (incorrectly stating the input "starts further to the right") yet generates perfectly aligned code. An end-to-end benchmark would falsely credit this as a visual success, masking the model's failure to actually "see" the design fidelity. Crucially, this implies these models are prone to failure in scenarios that deviate from their training priors. For such models, future optimization should prioritize enhancing true perceptual grounding to ensure fidelity in non-standard contexts.
> > >
> > > - **The "Execution Gap":** Conversely, some models perceive correctly but fail to implement. InternVL3-78B correctly identifies the alignment in the QA phase ("left edges are aligned") but produces a broken layout in the code generation (as seen in Figure R1). This isolates the bottleneck to coding capability rather than visual understanding—a distinction impossible to make without decoupled testing. For this group, the path to improvement lies in strengthening code generation capabilities rather than visual understanding.
> > >
> > > **Design2Code is an end-to-end "Black Box":** It only evaluates the final output. If a model generates correct code by guessing (priors) rather than seeing, Design2Code counts it as a success, masking the visual grounding failure. **FullFront is a "Diagnostic Suite":** By explicitly decoupling the Perception QA task from the Code Generation task, FullFront is the unique framework that can expose this decoupling. It allows researchers to pinpoint why a model fails (blindness vs. coding inability) or how it succeeds (true grounding vs. hallucinated priors). This level of granular diagnosis is impossible by simply manipulating inputs in an end-to-end benchmark. For more visual examples and analysis, please refer to **Appendix F.10: DETAILED VISUAL ANALYSIS OF PERCEPTION-CODE QUADRANTS** in the revised paper.

---

> > > > ### Author Response · Authors · 2025-11-26
> > > > **Response to Reviewer fHMJ's Follow-up (2/2)**
> > > >
> > > > **2. Static Translation vs. Dynamic Engineering (Missing Capabilities)**
> > > >
> > > > Front-end engineering extends far beyond static HTML generation. A key motivation of FullFront is to benchmark the **full lifecycle**, which existing benchmarks omit:
> > > > - **Interaction Authoring:** Modern web development requires implementing logic (e.g., dropdowns, hover effects). Our results (Table 8 in the main paper) show that models often fail to attach correct logic even if the static UI is correct. Static benchmarks like Design2Code inherently fail to capture these logic-level deficiencies.
> > > >
> > > > - **Code Refinement:** In real-world development, engineers rarely write from scratch; they refine existing code. Our Code Refinement task simulates this workflow, revealing that some open-source models struggle to follow "edit" instructions despite being able to generate from scratch.
> > > >
> > > > **3. Data Controllability and Granular Image Evaluation**
> > > >
> > > > Existing benchmarks often rely on scraped web data (e.g., Common Crawl), which introduces noise and uncontrolled complexity. FullFront employs a synthesis pipeline derived from real-world screenshots to ensure both high diversity and quality and standardized visual elements. This allows us to rigorously test specific capabilities (e.g., alignment, spacing) without confounding variables. Moreover, this pipeline is inherently scalable, enabling efficient data expansion and even the construction of large-scale, high-quality datasets for future model training.
> > > >
> > > > Furthermore, our unique **image handling strategy** sets FullFront apart. Unlike datasets that ignore image assets or use simple placeholders, we require models to select images from specific categories and manually set their dimensions. This design allowed us to uncover specific deficits—such as the "Abnormal Image Size" errors frequently observed in Qwen2.5-VL-72B-Instruct—providing critical feedback for future MLLM development that other UI-to-code datasets overlook.
> > > >
> > > > **Conclusion**
> > > >
> > > > In summary, we believe FullFront provides unique value by transforming the evaluation from a single "Success/Fail" metric into a comprehensive diagnostic suite. It enables the community to analyze why models fail (Perception vs. Code) and where they struggle across the broader workflow (Conceptualization, Perception, Interaction, and Refinement), paving the way for more robust Generalist Front-End Agents.

---

### Author Response · Authors · 2025-11-30
**Summary Comment**

We sincerely thank the AC and the reviewers for the time and effort devoted to improving our submission. During the rebuttal period, we provided detailed responses to systematically resolve every concern raised by the reviewers, supported by targeted analyses and verification experiments designed to address specific questions. All data and findings have been formally incorporated into the current revision of the paper.

**1. Successful Resolution of Concerns and Score Revision**

Prior to the discussion freeze, Reviewer JYpf explicitly acknowledged the effectiveness of our response and raised their score from 4 to 6, resulting in a score state of `6, 6, 4, 4`.

**2. Systematic Resolution of Concerns & Additional Experiments**

We have addressed all reviewer comments with specific solutions and experimental evidence. Key resolutions include:

* **Data-Pipeline Bias (Reviewers gXbA, 1Ttx)**
    * **Response:** We constructed a **completely new independent dataset of 300 samples using Grok-4** (a model completely absent from our original pipeline). The model performance ranking on this unseen data remained consistent with the results on our original dataset. Furthermore, our Code/Gemini scores achieved a Spearman correlation > 0.94 with human judgment. This proves FullFront evaluates fundamental capabilities rather than pipeline artifacts (see Appendix H).

* **Model-as-Judge Bias (Reviewers gXbA, 1Ttx)**
    * **Response:** We performed a rigorous **Cross-Judge Validation** involving three independent scoring rounds on 200 samples for each of the 6 representative models. The results demonstrated that Gemini 2.5 Flash exhibits zero self-preference (ranking itself objectively below GPT-5 and Claude 3.7 Sonnet) and maintains a perfect rank alignment ($\tau=1.0$) with human experts (see Appendix I).

* **Metric Rigidity of "Code Score" (Reviewer gXbA)**
    * **Response:** We conducted a Sensitivity Analysis on the Code Score. The results show that it penalizes actual errors 5.7x more severely than valid syntactic variations. This demonstrates superior discriminative power while maintaining a high correlation with human judgment ($\rho > 0.9$) (see Appendix J).

* **Necessity of an Aggregated Benchmark (Reviewers fHMJ, JYpf)**
    * **Response:** Through a **Perception-Implementation Gap Analysis** (manual verification of 100 samples for each of the 6 models, totaling 600 samples), we discovered that top-tier models often "fail to perceive but succeed in coding" (relying on priors), while open-source models "perceive correctly but fail to code." This diagnostic insight is impossible to uncover without FullFront's unified framework (see Section 5.2 and Appendix F).

**3. Consensus on Core Contributions**

FullFront is the first comprehensive benchmark designed to evaluate MLLMs across the complete front-end engineering workflow. Reviewers widely acknowledged its value:

- **Simulating the Real-World Workflow:** **Reviewer gXbA** highlighted that our benchmark uniquely "mirrors real front-end workflows rather than a single slice", a comprehensive scope echoed by **Reviewer 1Ttx**, who noted it covers the full pipeline "from concept, perception, and implementation."

- **Superior Data Quality and Construction Pipeline:** **Reviewer gXbA** commended our pipeline for "addressing common issues of bloated scraped code and oversimplified LLM HTML", while **Reviewer JYpf** praised its ability to transform real-world websites into "clean, standardized HTML to avoid copyright issues."

- **Fine-Grained Evaluation and Holistic Analysis:** **Reviewer gXbA** valued how we validated metrics against human judgments with "high Spearman correlations," supporting them as a "reasonable proxy for human preference". **Reviewer 1Ttx** specifically noted the effectiveness of combining "visual similarity with a Code Score to evaluate fine-grained elements", and **Reviewer fHMJ** appreciated the inclusion of "human evaluation". Furthermore, **Reviewers fHMJ**, **JYpf**, and **1Ttx** all acknowledged our "thorough coverage" and "comprehensive benchmarking" of diverse model families.

- **Vital Diagnostic Tool:** As demonstrated in our rebuttal analysis, FullFront uniquely uncovers critical behavioral phenomena—such as the **"Perception-Implementation Gap"**—offering insights into model bottlenecks that are impossible to detect with isolated perception or generation benchmarks.

- **Future-Proofing:** While no single current model has yet mastered the complete workflow, FullFront provides the necessary infrastructure to track this evolution. By pinpointing specific shortfalls in current capabilities, it establishes a foundational standard for evaluating the next generation of "**Generalist Front-End Agents**."

We believe our dedicated efforts and detailed responses have fully addressed the reviewers' concerns and clarified the contributions of our work.

---

### Meta-Review · Area_Chair_bj4j · 2026-01-06

**Summary:**

Apart from clarifications, the two main issues that emerged from the reviews are the following:
1. Rev JYpf and fHMJ raised an important concern about the benchmark cohesion & inter-task connectivity: the three tasks seem quite disconnected, and at first sight, there is no clear reason to consider them in a single benchmark.
2. Rev. 1Ttx and gXbA raised a point about the evaluation and dataset bias: the dataset is generated and evaluated by Gemini 2.5 Flash. Thus, there can be a bias that favors that specific model.

**Reviewer Concerns:**

The concern about dataset bias was solved in the rebuttal. Authors showed through solid experiments that both dataset and evaluation bias are not an issue on the proposed benchmark.
However, in my opinion, the issue about benchmark/sub-tasks cohesion is not solved. Even if the authors presented several experiments showing a connection between tasks (e.g., perception and code implementation), this does not provide a strong reason to consider the different subtasks in a unique benchmark. Furthermore, I agree with rev. JYpf that the webpage design task seems quite disconnected and not very relevant for the overall task of automated web page understanding and generation.
Overall, the proposed datasets and evaluation protocols, even if synthetic, make sense. However, the reason to cluster those tasks together seems quite weak, and the actual novelty of the evaluations seems limited. I thus consider that the proposed paper does not reach the high standard for acceptance to this conference.

**Reviewer Scores:**

- Rev. JYpf: 4 -> 4
I think JYpf maintained their score (4) even after the final messages from the authors as the motivation of building a single benchmark with different sub-tasks is weak.
- Rev. fHMJ: 4 -> 4
I think fHMJ would have maintained their score (4) as the motivation for building a single benchmark with the different sub-tasks was not compelling in my opinion.
- Rev. Gxba: 6 -> 6
Their main issue was about possible biases introduced by the use of Gemini for both generating the dataset and evaluation. Experiments in the rebuttal showed that there is no bias introduced by this. Thus, I think Gxba would have maintained their score (6).
- Rev. 1Ttx: 4 -> 6
The questions about dataset bias were answered well by the authors, thus I think that rev. 1Ttx could have increased their score to 6.

---

### Decision · Program_Chairs · 2026-01-26

Reject